# ACCELERATING DATA GENERATION FOR NEURAL OPERATORS VIA KRYLOV SUBSPACE RECYCLING

**Hong Wang[1,2]\*, Zhongkai Hao[3]\*, Jie Wang[1,2]†, Zijie Geng[1,2], Zhen Wang[1,2], Bin Li[1,2], Feng Wu[1,2]**

[1] CAS Key Laboratory of Technology in GIPAS, University of Science and Technology of China
[2] MoE Key Laboratory of Brain-inspired Intelligent Perception and Cognition, University of Science and Technology of China
{wanghong1700,ustcgzj,wangzhen0518}@mail.ustc.edu.cn,
{jiewangx,binli,fengwu}@ustc.edu.cn
[3] Tsinghua University
hzj21@mails.tsinghua.edu.cn

## ABSTRACT

Learning neural operators for solving partial differential equations (PDEs) has attracted great attention due to its high inference efficiency. However, training such operators requires generating a substantial amount of labeled data, i.e., PDE problems together with their solutions. The data generation process is exceptionally time-consuming, as it involves solving numerous systems of linear equations to obtain numerical solutions to the PDEs. Many existing methods solve these systems independently without considering their inherent similarities, resulting in extremely redundant computations. To tackle this problem, we propose a novel method, namely **S**orting **K**rylov **R**ecycling (**SKR**), to boost the efficiency of solving these systems, thus significantly accelerating data generation for neural operators training. To the best of our knowledge, SKR is the first attempt to address the time-consuming nature of data generation for learning neural operators. The working horse of SKR is Krylov subspace recycling, a powerful technique for solving a series of interrelated systems by leveraging their inherent similarities. Specifically, SKR employs a sorting algorithm to arrange these systems in a sequence, where adjacent systems exhibit high similarities. Then it equips a solver with Krylov subspace recycling to solve the systems sequentially instead of independently, thus effectively enhancing the solving efficiency. Both theoretical analysis and extensive experiments demonstrate that SKR can significantly accelerate neural operator data generation, achieving a remarkable speedup of up to 13.9 times.

## 1 INTRODUCTION

Solving Partial Differential Equations (PDEs) plays a fundamental and crucial role in various scientific domains, including physics, chemistry, and biology (Zachmanoglou & Thoe, 1986). However, traditional PDE solvers like the Finite Element Method (FEM) (Thomas, 2013) often involve substantial computational costs. Recently, data-driven approaches like neural operators (NOs) (Lu et al., 2019) have emerged as promising alternatives for rapidly solving PDEs (Zhang et al., 2023). NOs can be trained on pre-generated datasets as surrogate models. During practical applications, they only require a straightforward forward pass to predict solutions of PDEs which takes only several milliseconds. Such efficient methodologies for solving PDEs hold great potential for real-time predictions and addressing both forward and inverse problems in climate (Pathak et al., 2022), fluid dynamics (Wen et al., 2022), and electromagnetism (Augenstein et al., 2023).

Despite their high inference efficiency, a primary limitation of NOs is the need for a significant volume of labeled data during training. For instance, when training a Fourier Neural Operator

---

\*Equal contribution.
†Corresponding author.

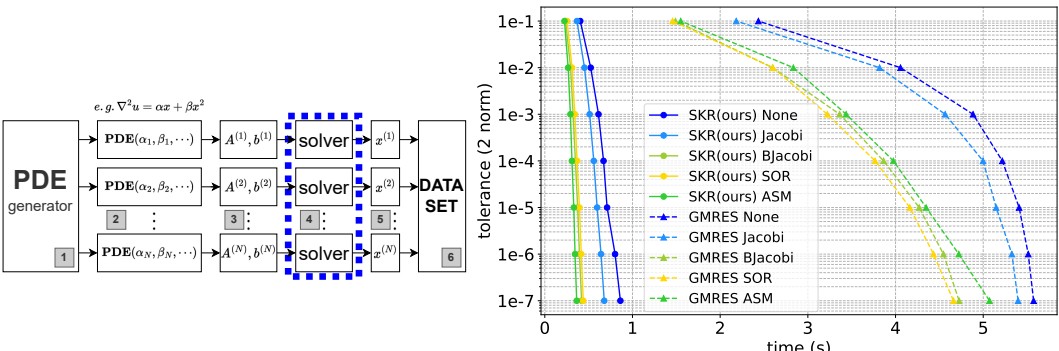

Figure 1: **Left.** Generation process of the NO dataset. 1. Generate a set of random parameters from NO. 2. Export the corresponding PDE based on these parameters. 3. Transform the PDE into a system of linear equations using discretization methods. 4. Invoke linear equation solvers to solve these systems independently. 5. Obtain solutions to the linear systems and convert them into PDE solutions. 6. Assemble into a dataset. **Right.** The accuracy change over time for our SKR algorithm and the baseline GMRES algorithm. As seen, the SKR algorithm can significantly accelerate the solution of the system of linear equations, achieving a speed-up of up to 13.9 times.

(FNO) (Li et al., 2020) for Darcy flow problems, thousands of PDEs and their corresponding solutions under various initial conditions are often necessary. Acquiring this data typically entails running numerous traditional simulations independently, leading to elevated computational expenses. For solvers such as Finite Element Method (FEM) (LeVeque, 2002), accuracy often increases quadratically or even cubically with the mesh grid count. Depending on the desired accuracy, dataset generation can range from several hours to thousands of hours. Furthermore, the field of PDEs is unique compared to other domains, given the significant disparities between equations and the general lack of data generalization. Specifically, for each type of PDE, a dedicated dataset needs recreation when training NOs. The computational cost for generating data for large-scale problems has become a bottleneck hindering the practical usage of NOs (Zhang et al., 2023). While certain methodologies propose integrating physical priors or incorporating physics-informed loss to potentially increase data efficiency, these approaches are still in their primary stages and are not satisfactory for practical problems (Hao et al., 2022). Consequently, designing efficient strategies to generate the data for NOs remains a foundational and critical challenge (Geng et al., 2023).

The creation of the NO dataset is depicted in Figure 1. Notably, the computation in the 4-th step Potentially constitutes approximately 95% of the entire process (Hughes, 2012), underscoring it as a prime candidate for acceleration. While conventional approaches tackle each linear system independently during dataset formulation, we posit that there is an inherent interconnectivity among them. Specifically, as shown in Figure 4 and 9, systems originating from a similar category of PDEs often exhibit comparable matrix structures, eigenvector subspaces, and solution vectors (Parks et al., 2006). Addressing them in isolation leads to considerable computational redundancy.

To address the prevalent issue of computational redundancy in conventional linear system solutions, we introduce a pioneering and efficient algorithm, termed $\underline{S}$orting $\underline{K}$rylov $\underline{R}$ecycling (**SKR**). SKR is ingeniously designed to optimize the process from the ground up. Its initial phase involves a **S**orting algorithm, which serializes the linear systems. This serialization strategy is carefully crafted to augment the correlations between successive systems, setting the stage for enhanced computational synergy. In its subsequent phase, SKR delves into the **K**rylov subspace established during the linear system resolutions. Through the application of a '**R**ecycling' technique, it capitalizes on eigenvectors and invariant subspaces identified from antecedent solutions, thereby accelerating the convergence rate and substantially reducing the number of iterations and associated computation time. Central to its design, SKR discerns and leverages the inherent interrelations within these linear systems. Rather than approaching each system as a discrete entity, SKR adeptly orchestrates their sequential resolution, eliminating considerable redundant calculations that arise from similar structures. This refined methodology not only alleviates the computational demands of linear system solutions but also markedly hastens the creation of training data for NOs. Codes are available at https://github.com/wanghong1700/NO-DataGen-SKR.

## 2 RELATED WORK

### 2.1 DATA-EFFICIENT NEURAL OPERATORS AND LEARNED PDE SOLVERS

Neural operators, such as the Fourier Neural Operator (FNO) (Li et al., 2020) and Deep Operator Network (DeepONet) (Lu et al., 2019), are effective models for solving PDEs. However, their training requires large offline paired parametrized PDE datasets. To improve data efficiency, research has integrated physics-informed loss mechanisms similar to Physics Informed Neural Networks (PINNs) (Raissi et al., 2017). This loss function guides neural operators to align with PDEs, cutting down data needs. Moreover, specific architectures have been developed to maintain symmetries and conservation laws (Brandstetter et al., 2022; Liu et al., 2023), enhancing both generalization and data efficiency. Yet, these improvements largely focus on neural operators without fundamentally revising data generation.

Concurrently, there's a push to create data-driven PDE solvers. For example, Hsieh et al. (2019) suggests a data-optimized parameterized iterative scheme. Additionally, combining neural operators with traditional numerical solvers is an emerging trend. An example is the hybrid iterative numerical transferable solver, merging numerical methods with neural operators for greater accuracy (Zhang et al., 2022). This model incorporates Temporal Stencil Modeling. Notably, numerical iterations can also be used as network layers, aiding in solving PDE-constrained optimization and other challenges.

### 2.2 KRYLOV SUBSPACE RECYCLING

The linear systems produced by the process of generating NOs training data typically exhibit properties of sparsity and large-scale nature (Hao et al., 2022; Zhang et al., 2023). In general, for non-symmetric matrices, Generalized minimal residual method (GMRES) (Saad & Schultz, 1986; Qin & Xu, 2023) of the Krylov algorithm is often employed for generating training data pertinent to NOs.

The concept of recycling in Krylov algorithms has found broad applications across multiple disciplines. For instance, the technique has been leveraged to refine various matrix algorithms (Wang et al., 2007; Mehrmann & Schröder, 2011). Furthermore, it has been deployed for diverse matrix problems (Parks et al., 2006; Gaul, 2014; Soodhalter et al., 2020). Moreover, the recycling methodology has also been adapted to hasten iterative solutions for nonlinear equations (Kelley, 2003; Deuflhard, 2005; Gaul & Schlömer, 2012; Gaul & Schlömer). Additionally, it has been instrumental in accelerating the resolution of time-dependent PDE equations (Meurant, 2001; Bertaccini et al., 2004; Birken et al., 2008). We have designed a sorting algorithm tailored for NO, building upon the original recycling algorithm, making it more suitable for the generation of NO datasets.

## 3 PRELIMINARIES

### 3.1 DISCRETIZATION OF PDEs IN NEURAL OPERATOR TRAINING

We primarily focus on PDE NOs for which the time overhead of generating training data is substantial. As illustrated in Figure 1, the generation of this training data necessitates solving the corresponding PDEs. Due to the inherent complexity of these PDEs and their intricate boundary conditions, discretized numerical algorithms like FDM, FEM, and FVM are commonly employed for their solution (Strikwerda, 2004; Hughes, 2012; Johnson, 2012; LeVeque, 2002; Cheng & Xu, 2023).

These discretized numerical algorithms embed the PDE problem from an infinite-dimensional Hilbert function space into a suitable finite-dimensional space, thereby transforming the PDE issue into a system of linear equations. We provide a straightforward example to elucidate the process under discussion. The detailed procedure for generating the linear equation system is available in the Appendix A. Specifically, we discuss solving a two-dimensional Poisson equation using the FDM to transform it into a system of linear equation:

$$\nabla^2 u(x,y) = f(x,y),$$

using a $2 \times 2$ internal grid (i.e., $N_x = N_y = 2$ and $\Delta x = \Delta y$), the unknowns $u_{i,j}$ can be arranged in row-major order as follows: $u_{1,1}, u_{1,2}, u_{2,1}, u_{2,2}$. For central differencing on a $2 \times 2$ grid, the vector

$\boldsymbol{b}$ will contain the values of $f(x_i, y_j)$ and the linear equation system $\boldsymbol{Ax} = \boldsymbol{b}$ can be expressed as:

$$\begin{bmatrix} -4 & 1 & 1 & 0 \\ 1 & -4 & 0 & 1 \\ 1 & 0 & -4 & 1 \\ 0 & 1 & 1 & -4 \end{bmatrix} \begin{bmatrix} u_{1,1} \\ u_{1,2} \\ u_{2,1} \\ u_{2,2} \end{bmatrix} = \begin{bmatrix} f(x_1, y_1) \\ f(x_1, y_2) \\ f(x_2, y_1) \\ f(x_2, y_2) \end{bmatrix}.$$

By employing various parameters to generate $f$, such as utilizing Gaussian random fields (GRF) or truncated Chebyshev polynomials, we can derive Poisson equations characterized by distinct parameters.

Typically, training a NO requires generating between $10^3$ and $10^5$ numerical solutions for PDEs (Lu et al., 2019). Such a multitude of linear systems, derived from the same distribution of PDEs, naturally exhibit a high degree of similarity. It is precisely this similarity that is key to the effective acceleration of our SKR algorithm's recycle module. (Soodhalter et al., 2020). We can conceptualize this as the task of solving a sequential series of linear equations:

$$\boldsymbol{A}^{(i)}\boldsymbol{x}^{(i)} = \boldsymbol{b}^{(i)} \quad i = 1, 2, \cdots, \tag{1}$$

where the matrix $\boldsymbol{A}^{(i)} \in \mathbb{C}^{n \times n}$ and the vector $\boldsymbol{b}^{(i)} \in \mathbb{C}^n$ vary based on different PDEs.

## 3.2 KRYLOV SUBSPACE METHOD

For solving large-scale sparse linear systems, Krylov subspace methods are typically used (Saad, 2003; Greenbaum, 1997). The core principle behind this technique is leveraging the matrix from the linear equation system to produce a lower-dimensional subspace, which aptly approximates the true space. This approach enables the iterative solutions within this subspace to gravitate towards the actual solution $\boldsymbol{x}$.

Suppose we now solve the $i$-th system and remove the superscript of $\boldsymbol{A}^{(i)}$. The $m$-th Krylov subspace associated with the matrix $\boldsymbol{A}$ and the starting vector $\boldsymbol{r}$ as follows:

$$\mathcal{K}_m(\boldsymbol{A}, \boldsymbol{r}) = \text{span}\{\boldsymbol{r}, \boldsymbol{Ar}, \boldsymbol{A}^2\boldsymbol{r}, \cdots, \boldsymbol{A}^{m-1}\boldsymbol{r}\}. \tag{2}$$

Typically, $\boldsymbol{r}$ is chosen as the initial residual in linear system problem. The Krylov subspace iteration process leads to the Arnoldi relation:

$$\boldsymbol{AV}_m = \boldsymbol{V}_{m+1}\underline{\boldsymbol{H}}_m, \tag{3}$$

where $\boldsymbol{V}_m \in \mathbb{C}^{n \times m}$ and $\underline{\boldsymbol{H}}_m \in \mathbb{C}^{(m+1) \times m}$ is upper Hessenberg. Let $\boldsymbol{H}_m \in \mathbb{C}^{m \times m}$ denote the first $m$ rows of $\underline{\boldsymbol{H}}_m$. By solving the linear equation system related to $\boldsymbol{H}_m$, we obtain the solution within the $\mathcal{K}_m(\boldsymbol{A}, \boldsymbol{r})$, thereby approximating the true solution. If the accuracy threshold is not met, we'll expand the dimension of the Krylov subspace. The process will iteratively transitions from $\mathcal{K}_m(\boldsymbol{A}, \boldsymbol{r})$ to $\mathcal{K}_{m+1}(\boldsymbol{A}, \boldsymbol{r})$, continuing until the desired accuracy is achieved (Arnoldi, 1951; Saad, 2003). For the final Krylov subspace, we omit the subscript $m$ and denote it simply as $\mathcal{K}(\boldsymbol{A}, \boldsymbol{r})$.

For any subspace $S \subseteq \mathbb{C}^n$, $\boldsymbol{y} \in \boldsymbol{S}$ is a Ritz vector of $\boldsymbol{A}$ with Ritz value $\theta$, if

$$\boldsymbol{Ay} - \theta\boldsymbol{y} \perp \boldsymbol{w} \quad \forall \boldsymbol{w} \in S.$$

For iterative methods within the Krylov subspace, we choose $S = \mathcal{K}_m(\boldsymbol{A}, \boldsymbol{r})$ and the eigenvalues of $\boldsymbol{H}_m$ are the Ritz values of $\boldsymbol{A}$.

Various Krylov algorithms exist for different types of matrices. The linear systems generated by the NOs we discuss are typically non-symmetric. The most widely applied and successful algorithm for solving large-scale sparse non-symmetric systems of linear equations is GMRES (Saad & Schultz, 1986; Morgan, 2002), which serves as the baseline for this paper.

## 4 METHOD

As shown in the Figure 2, we aim to accelerate the solution by harnessing the intrinsic correlation among these linear systems. To elaborate, when consecutive linear systems exhibit significant correlation, perhaps due to minor perturbations, it's plausible that certain information from the prior solution, such as Ritz values and Ritz vectors, might not be fully utilized. This untapped information

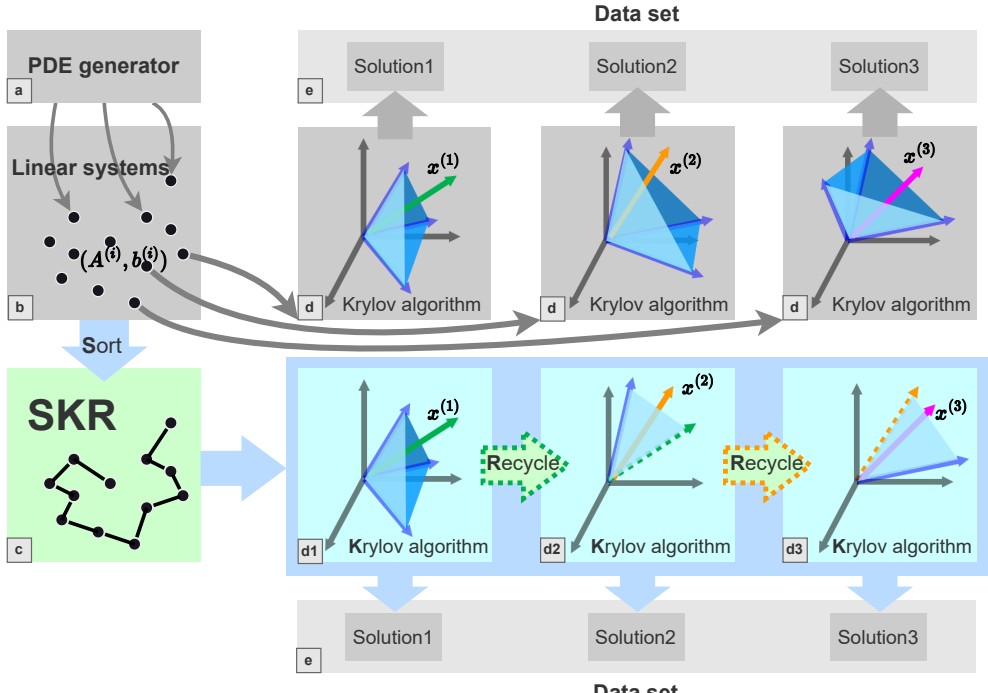

Figure 2: Algorithm Flow Diagram: **a**. Derive the PDEs to be solved from the NO. **b**. Transform these PDEs into a system of linear equations. **c**. Apply the SKR algorithm to sort the linear systems, obtaining a sequence with strong correlations. **d**. Traditional algorithms independently solve the linear systems from step b using Krylov methods. **d1, d2, d3**. The SKR algorithm utilizes 'recycling' to sequentially solve the linear systems, reducing the dimension of the Krylov subspace. **e**. Obtain the solutions and assemble them into a dataset.

could potentially expedite the resolution of subsequent linear systems. This concept, often termed 'recycling', has been well-established in linear system algorithms.

While the notion of utilizing the recycling concept is appealing, the PDEs and linear systems derived from NO parameters are intricate and convoluted. This complexity doesn't guarantee a strong correlation between sequential linear equations, thereby limiting the effectiveness of this approach. To address this issue, we've formulated a sorting algorithm tailored for the PDE solver. Our algorithm SKR, engineered for efficiency, ensures a sequence of linear equations where preceding and subsequent systems manifest pertinent correlations. This organized sequence enables the fruitful application of the recycling concept to expedite the solution process.

### 4.1 THE SORTING ALGORITHM

For linear systems derived from PDE solvers, the generation of these equations is influenced by specific parameters. These parameters and their associated linear equations exhibit continuous variations (Lu et al., 2022; Li et al., 2020). Typically, these parameters are represented as parameter matrices. Our objective is to utilize these parameters to assess the resemblance among various linear systems, and then sequence them to strengthen the coherence between successive systems.

Subsequent to our in-depth Theoretical Analysis 5.2, we arrive at the conclusion that the chosen recycling component is largely resilient to matrix perturbations. It means there is no necessity for excessive computational resources for sorting. As a result, we incorporated a Sorting Algorithm 1 based on the greedy algorithm, serving as the step c in Figure 1.

In typical training scenarios for NOs, the number of data points, represented as $num$, can be quite vast, frequently falling between $10^3$ and $10^5$. To manage this efficiently, one could adopt a cost-effective sorting strategy. Initially, divide the data points into smaller groups, each containing either

---

**Algorithm 1:** The Sorting Algorithm

---

**Input:** Sequence of linear systems to be solved $\boldsymbol{A}^{(i)} \in \mathbb{C}^{n \times n}, \boldsymbol{b}^{(i)} \in \mathbb{C}^n$, corresponding
parameter matrix $\boldsymbol{P}^{(i)} \in \mathbb{C}^{p \times p}$ and $i = 1, 2, \cdots, N$

**Output:** Sequence for solving systems of linear equations $seq_{mat}$

1   Initialize the list with sequence $seq_0 = \{1, 2, \cdots, N\}$, $seq_{mat}$ is an empty list;

2   Set $i_0 = 1$ as the starting point. And remove 1 from $seq_0$ and append 1 to $seq_{mat}$;

3   **for** $i = 1, \cdots, N - 1$ **do**

4      Refresh $dis$ and set it to a large number, e.g., 1000;

5      **for** *each $j$ in $seq_0$* **do**

6          $dis_j = $ the Frobenius norm of the difference between $\boldsymbol{P}^{(i_0)}$ and $\boldsymbol{P}^{(j)}$;

7          **if** $dis_j < dis$ **then**

8              $dis = dis_j$ and $j_{min} = j$;

9      Remove $j_{min}$ from $seq_0$ and append $j_{min}$ to $seq_{mat}$ and set $i_0 = j_{min}$;

10   Get the sequence for solving linear systems $seq_{mat}$.

---

$10^3$ to $10^4$ data points, based on their coordinates. Then, use the greedy algorithm to sort within these groups. Once sorted, these smaller groups can be concatenated.

## 4.2 KRYLOV SUBSPACE RECYCLING

For a sequence of linear systems, there exists an inherent correlation between successive systems. Consequently, we posit that by leveraging information from the solution process of prior systems, we can expedite the iterative convergence of subsequent system, thereby achieving a notable acceleration in computational performance.

For various types of PDE problems, the linear systems generated exhibit matrices with distinct structural characteristics. These unique matrix structures align with specific Krylov recycling algorithms. To validate our algorithm's efficacy, we focus primarily on the most prevalent and general scenario within the domain of NOs, where matrix $\boldsymbol{A}$ is nonsymmetric. Given this context, Generalized Conjugate Residual with Deflated Restarting (GCRO-DR) stands out as the optimal Krylov recycling algorithm. It also forms the core algorithm for steps (d1, d2, d3) in Figure 2.

GCRO-DR employs deflated restarting, building upon the foundational structure of GCRO as illustrated in (de Sturler, 1996). Essentially, GCRO-DR integrates the approximate eigenvector derived from the solution of a preceding linear system into subsequent linear systems. This is achieved through the incorporation of a deflation space preconditioning group solution, leading to a pronounced acceleration effect (Soodhalter et al., 2020; Nicolaides, 1987; Erlangga & Nabben, 2008). The detailed procedure is delineated as follows:

Suppose that we have solved the $i$ th system of (1) with GCRO-DR, and we retain $k$ approximate eigenvectors, $\widetilde{\boldsymbol{Y}}_k = [\widetilde{\boldsymbol{y}}_1, \widetilde{\boldsymbol{y}}_1, \cdots, \widetilde{\boldsymbol{y}}_k]$. Then, GCRO-DR computes matrices $\boldsymbol{U}_k, \boldsymbol{C}_k \in \mathbb{C}^{n \times k}$ from $\widetilde{\boldsymbol{Y}}_k$ and $\boldsymbol{A}^{(i+1)}$ such that $\boldsymbol{A}^{(i+1)} \boldsymbol{U}_k = \boldsymbol{C}_k$ and $\boldsymbol{C}_k^H \boldsymbol{C}_k = \boldsymbol{I}_k$, The specific algorithm can be found in Appendix B.1. By removing the superscript of $\boldsymbol{A}^{(i+1)}$ we can get the Arnoldi relation of GCRO-DR:

$$(\boldsymbol{I} - \boldsymbol{C}_k \boldsymbol{C}_k^H) \boldsymbol{A} \boldsymbol{V}_{m-k} = \boldsymbol{V}_{m-k+1} \underline{\boldsymbol{H}}_{m-k}. \tag{4}$$

Due to the correlation between the two consecutive linear systems, $\boldsymbol{C}_k$ encompasses information from the previous system $\boldsymbol{A}^{(i)}$. This means that when constructing a new Krylov subspace $\mathcal{K}(\boldsymbol{A}^{(i+1)}, \boldsymbol{r}^{(i+1)})$, there is no need to start from scratch; the subspace $\mathrm{span}\{\widetilde{\boldsymbol{Y}}_k\}$ composed of some eigenvectors from $\boldsymbol{A}^{(i)}$ already exists. Building upon this foundation, $\mathcal{K}(\boldsymbol{A}^{(i+1)}, \boldsymbol{r}^{(i+1)})$ can converge more rapidly to the subspace where the solution $\boldsymbol{x}$ lies. This can significantly reduce the dimensionality of the final Krylov subspace, leading to a marked decrease in the number of iterations and resulting in accelerated performance.

Compared to the Arnoldi iteration presented in (3) used by GMRES, the way GCRO-DR leverages solution information from previous systems to accelerate the convergence of a linear sequence becomes clear. GMRES can be intuitively conceptualized as the special case of GCRO-DR where $k$ is initialized at zero (Carvalho et al., 2011; Morgan, 2002; Parks et al., 2006).

The effectiveness of acceleration within the Krylov subspace recycling is highly dependent on selecting $\widetilde{Y}_k$ in a manner that boosts convergence for the following iterations of the linear systems. Proper sequencing can amplify the impact of $\widetilde{Y}_k$, thereby reducing the number of iterations. This underlines the importance of sorting. For a detailed understanding of GCRO-DR, kindly consult the pseudocode provided in the Appendix B.2.

## 5 THEORETICAL ANALYSIS

### 5.1 CONVERGENCE ANALYSIS

When solving a single linear system, GCRO-DR and GMRES-DR are algebraically equivalent. The primary advantage of GCRO-DR is its capability for solving sequences of linear systems (Carvalho et al., 2011; Morgan, 2002; Parks et al., 2006). Stewart (2001); Embree (2022) provide a good framework to analyze the convergence of this type of Krylov algorithm. Here we quote Theorem 3.1 in Parks et al. (2006) to make a detailed analysis of GCRO-DR. We define the $one-sided\ distance$ from the subspace $\mathcal{Q}$ to the subspace $\mathcal{C}$ as

$$\delta(\mathcal{Q}, \mathcal{C}) = \|(I - \Pi_{\mathcal{C}})\Pi_{\mathcal{Q}}\|_2, \tag{5}$$

where $\Pi$ represents the projection operator for the associated space. Its mathematical interpretation corresponds to the largest principal angle between the two subspaces $\mathcal{Q}$ and $\mathcal{C}$ (Beattie et al., 2004), and defineing $P_{\mathcal{Q}}$ as the spectral projector onto $\mathcal{Q}$.

**Theorem 1.** *Given a space $\mathcal{C} = \mathrm{range}(C_k)$, let $\mathcal{V} = \mathrm{range}(V_{m-k+1}\underline{H}_{m-k})$ be the $(m-k)$ dimensional Krylov subspace generated by GCRO-DR as in (4). Let $r_0 \in \mathbb{C}^n$, and let $r_1 = (I - \Pi_{\mathcal{C}})r_0$. Then, for each $\mathcal{Q}$ such that $\delta(\mathcal{Q}, \mathcal{C}) < 1$,*

$$\min_{d_1 \in \mathcal{V} \oplus \mathcal{C}} \|r_0 - d_1\|_2 \leq \min_{d_2 \in (I - P_{\mathcal{Q}})\mathcal{V}} \|(I - P_{\mathcal{Q}})r_1 - d_2\|_2$$
$$+ \frac{\gamma}{1 - \delta}\|P_{\mathcal{Q}}\|_2 \cdot \|(I - \Pi_{\mathcal{V}})r_1\|_2, \tag{6}$$

*where $\gamma = \|(I - \Pi_{\mathcal{C}})P_{\mathcal{Q}}\|_2$.*

In the aforementioned bounds, the left-hand side signifies the residual norm subsequent to $m-k$ iterations of GCRO-DR, employing the recycled subspace $\mathcal{C}$. Contrastingly, on the right-hand side, the initial term epitomizes the convergence of a deflated problem, given that all components within the subspace $\mathcal{Q}$ have been eradicated (Morgan, 2002; Simoncini & Szyld, 2005; Van der Vorst & Vuik, 1993). The subsequent term on the right embodies a constant multiplied by the residual following $m - k$ iterations of GCRO-DR, when solving for $r_1$. Should the recycling space $\mathcal{C}$ encompass an invariant subspace $\mathcal{Q}$, then $\delta = \gamma = 0$ for the given $\mathcal{Q}$, ensuring that the convergence rate of GCRO-DR matches or surpasses that of the deflated problem. In most cases, $\|P_{\mathcal{Q}}\|_2$ is numerically stable and not large, therefore a reduced value of $\delta$ directly correlates with faster convergence in GCRO-DR.

### 5.2 THE RATIONALITY OF SORTING ALGORITHMS

A critical inquiry arises: How much computational effort should be allocated to a sorting algorithm? Given the plethora of sorting algorithms available, is it necessary to expend significant computational resources to identify the optimal sorting outcome?

Drawing from Theorem 3.2 as presented in (Kilmer & De Sturler, 2006): "When the magnitude of the change is smaller than the gap between smallest and large eigenvalues, then the invariant subspace associated with the smallest eigenvalues is not significantly altered." We derive the following insights: The perturbation of invariant subspaces associated with the smallest eigenvalues when the change in the matrix is concentrated in an invariant subspace corresponding to large eigenvalues (Sifuentes et al., 2013; Parks et al., 2006). This effectively implies that the pursuit of optimal sorting isn't imperative. Because we typically recycle invariant subspaces formed by smaller eigenvectors. As long as discernible correlations persist between consecutive linear equations, appreciable acceleration effects are attainable (Gaul, 2014). This rationale underpins our decision to develop a cost-efficient sorting algorithm. Despite its suboptimal nature, the resultant algorithm exhibits commendable performance.

## 6 EXPERIMENT

### 6.1 SET UP

To comprehensively evaluate the performance of **SKR** in comparison to another algorithm, we conducted nearly 3,000 experiments. The detailed data is available in the Appendix D.6. Each experiment utilized a data set crafted to emulate an authentic NO training data set. Our analysis centered on two primary performance metrics viewed through three perspectives. These tests spanned four different datasets, with SKR consistently delivering commendable results. Specifically, the three Perspectives are: 1. Matrix preconditioning techniques, spanning 7 to 10 standard methods. 2. Accuracy criteria for linear system solutions, emphasizing 5 to 8 distinct tolerances. 3. Different matrix sizes, considering 5 to 6 variations. Our primary performance Metrics encompassed: 1. Average computational time overhead. 2. Mean iteration count. For a deeper dive into the specifics, please consult the Appendix D.1.

**Baselines.** As previously alluded to, our focus revolves around a system of linear equations, which consists of large sparse non-symmetric matrices. The GMRES algorithm serves as the predominant solution and sets the benchmark for our study. We utilized the latest version from PETSc 3.19.4 for GMRES.

**Datasets.** To probe the algorithm's adaptability across matrix types, we delved into four distinct linear equation challenges, each rooted in a PDE: 1. Darcy Flow Problem (Li et al., 2020; Rahman et al., 2022; Kovachki et al., 2021; Lu et al., 2022); 2. Thermal Problem (Sharma et al., 2018; Koric & Abueidda, 2023); 3. Poisson Equation (Hsieh et al., 2019; Zhang et al., 2022); 4. Helmholtz Equation (Zhang et al., 2022). For an in-depth exposition of the dataset and its generation, kindly refer to the Appendix D.2. For the runtime environment, refer to Appendix D.4.

### 6.2 QUANTITATIVE RESULTS

Table 1: Comparison of our SKR and GMRES computation time and iterations across datasets, preconditioning, and tolerances. The first column lists datasets with matrix side lengths, the next details tolerances. The data is displayed as 'computation time speed-up ratio/iteration count speed-up ratio'. A GMRES/SKR ratio over 1 denotes better SKR performance.

| Dataset | Time/Iter | None | Jacobi | BJacobi | SOR | ASM | ICC | ILU |
|---|---|---|---|---|---|---|---|---|
| Darcy 6400 | 1e-2 | 2.62/19.2 | 2.88/22.6 | 3.21/23.6 | 2.69/23.3 | 2.23/13.4 | 1.97/9.55 | 1.93/9.44 |
| | 1e-5 | 2.92/21.1 | 3.42/24.5 | 4.07/28.6 | 3.45/27.9 | 3.66/22.2 | 3.18/14.9 | 3.08/14.4 |
| | 1e-8 | 2.70/19.1 | 3.09/22.9 | 4.00/27.5 | 3.54/27.5 | 4.53/25.8 | 4.11/18.9 | 3.70/17.2 |
| Thermal 11063 | 1e-5 | 4.53/20.8 | 3.32/15.0 | 2.38/10.3 | 1.96/8.76 | 2.46/10.3 | 2.40/10.3 | 2.35/10.3 |
| | 1e-8 | 5.35/23.6 | 3.06/13.5 | 2.73/11.1 | 2.34/9.30 | 2.83/11.1 | 2.77/11.1 | 2.69/11.1 |
| | 1e-11 | 5.47/24.9 | 2.93/12.8 | 3.05/11.7 | 2.62/10.2 | 3.14/11.7 | 3.08/11.7 | 3.01/11.7 |
| Poisson 71313 | 1e-5 | 1.27/4.69 | 1.28/4.68 | 1.13/3.87 | 1.19/4.05 | 1.46/3.93 | 0.99/3.92 | 0.98/3.92 |
| | 1e-8 | 1.74/6.29 | 1.75/6.30 | 1.19/3.97 | 1.35/4.45 | 1.94/4.83 | 1.32/4.83 | 1.30/4.87 |
| | 1e-11 | 1.90/6.83 | 1.91/6.82 | 1.19/3.95 | 1.33/4.35 | 2.12/5.11 | 1.42/5.13 | 1.38/5.13 |
| Helmholtz 10000 | 1e-2 | 7.74/17.3 | 8.44/20.3 | 8.83/21.5 | 8.37/22.3 | 10.6/25.4 | 4.77/21.4 | 3.88/17.6 |
| | 1e-5 | 7.61/16.5 | 8.62/20.0 | 11.4/26.5 | 10.5/26.2 | 13.1/29.3 | 6.38/28.3 | 6.13/27.2 |
| | 1e-7 | 6.47/13.68 | 7.96/18.1 | 11.3/26.2 | 10.6/25.9 | 13.9/30.0 | 6.72/29.3 | 6.34/28.0 |

Table 1 showcases selected experimental data. From this table, we can infer several conclusions:

Firstly, across almost all accuracy levels and preconditioning techniques, our SKR method consistently delivers impressive acceleration. It is most noticeable in the Helmholtz dataset. Depending on the convergence accuracy and preconditioning technique, our method reduces wall clock time by factors of 2 to 14 and requires up to 30 times fewer iterations. This suggests that linear systems derived from similar PDEs have inherent redundancies in their resolution processes. Our SKR algorithm effectively minimizes such redundant computations, slashing iteration counts and significantly speeding up the solution process, ultimately accelerating the generation of datasets for neural operators.

Secondly, in most cases, as the demanded solution accuracy intensifies, the acceleration capability of our algorithm becomes more pronounced. For instance, in the Darcy flow problem, the acceleration

for a high accuracy level (1e-8) is 50% to 100% more than for a lower accuracy (1e-2). On one hand, high-precision data demands a larger Krylov subspace and more iterations. SKR's recycling technology leverages information from previously similar linear systems, enabling faster convergence in the Krylov subspace. Thus, its performance is especially superior at higher precisions. Detailed subsequent analyses further corroborate this observation, as seen in Appendix D.5.1, D.5.2. On the other hand, Theoretical Analysis 5.2 indicates that the information recycled in the SKR algorithm possesses strong anti-perturbation capabilities, ensuring maintained precision. Its acceleration effect becomes even more evident at higher accuracy requirements.

Thirdly, we observe that our SKR algorithm can significantly reduce the number of iterations, with reductions up to 30 times, indirectly highlighting the strong algorithmic stability of the SKR method. By analyzing the count of experiments reaching the maximum number of iterations, we can conclude that the stability of the SKR algorithm is superior to that of GMRES. Detailed experimental analyses can be found in the Appendix D.5.3.

Lastly, it is worth noting that outcomes vary based on the preconditioning methods applied. Our algorithm demonstrates remarkable performance when no preconditioner, Jacobian, or Successive Overrelaxation Preconditioner (SOR) methods are used. However, its efficiency becomes less prominent for low-accuracy Poisson equations that employ incomplete Cholesky factorization (ICC) or incomplete lower-upper factorization (ILU) preconditioners. The underlying reason is that algorithms like ICC and ILU naturally operate by omitting smaller matrix elements or those with negligible impact on the preconditioning decomposition, and then proceed with Cholesky or LU decomposition. Such an approach somewhat clashes with the recycling method. While SKR leverages information from prior solutions of similar linear equation systems, ICC and ILU can disrupt this similarity. As a result, when employing ICC or ILU preconditioning, the improvement in SKR's speed is less noticeable at lower accuracy levels. However, at higher precisions, our SKR algorithm still exhibits significant acceleration benefits with ICC and ILU preconditioning.

## 6.3 Ablation Study

To assess the impact of 'sort' on our SKR algorithm, we approached the analysis from both theoretical and experimental perspectives:

Theoretical Perspective: Based on previous Theoretical insights 5.1, the pivotal metric to gauge the convergence speed of the Krylov recycle algorithm is $\delta$. Hence, we examined the variations in the $\delta$ metric before and after sorting. Experimental Perspective: We tested the SKR algorithm both with and without the 'sort' feature, examining the disparities in computation time and iteration count.

Table 2: Comparison of algorithm performance with and without sort in Darcy flow problem, using SOR preconditioning, matrix size $10^4$, and computational tolerance $1e - 8$.

|  | Time(s) | Iter | $\delta$ |
|---|---|---|---|
| SKR(sort) | 0.101 | 183.9 | 0.90 |
| SKR(nosort) | 0.114 | 202.5 | 0.95 |

As illustrated in the Table 2: 1. The $\delta$ metric declined by 5% after employing the 'sort' algorithm. As illustrated in Figure 8, this effectively demonstrates that the 'sort' algorithm can enhance the correlation between consecutive linear equation sets, achieving the initial design goal of 'sort'. 2. Using 'sort' enhances the SKR algorithm's computational speed by roughly 13% and decreases its iterations by 9.2%. This implies that by increasing the coherence among sequential linear equation sets, the number of iterations needed is minimized, thus hastening the system's solution process.

## 7 Limitation and Conclusions

Our paper presents the SKR algorithm as a significant advancement in generating neural operator datasets, especially for linear PDEs, yet recognizes areas for improvement. Future work could explore adapting the SKR for non-linear PDEs, refining the sorting algorithm's distance metrics for better system correlation, and extending the recycling concept to further data generation realms. This pioneering effort not only enhances computational efficiency but also makes neural operators more accessible, indicating a major leap in their development and application.

## REPRODUCIBILITY STATEMENT

For the sake of reproducibility, we have included our codes within the supplementary materials. However, it's worth noting that the current code version lacks structured organization. Should this paper be accepted, we commit to reorganizing the codes for improved clarity. Additionally, in the Appendix D, we provide an in-depth description of our experimental setups and detailed results.

## ACKNOWLEDGEMENTS

The authors would like to thank all the anonymous reviewers for their insightful comments. This work was supported in part by National Key R&D Program of China under contract 2022ZD0119801, National Nature Science Foundations of China grants U23A20388, U19B2026, U19B2044, and 62021001.

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

## A CONVERTING PDEs TO LINEAR SYSTEMS: AN EXAMPLE

### A.1 OVERVIEW

The general approach for solving Partial Differential Equations (PDEs) using techniques like Finite Difference Method (FDM), Finite Element Method (FEM), and Finite Volume Method (FVM) can be broken down into the following key steps (Strikwerda, 2004; Hughes, 2012; Johnson, 2012; LeVeque, 2002):

1. Mesh Generation: Partition the domain over which the PDE is defined into a grid of specific shapes, such as squares or triangles. As illustrated, the Figure 3 shows the FEM finite element mesh for the Poisson equation with a square boundary.

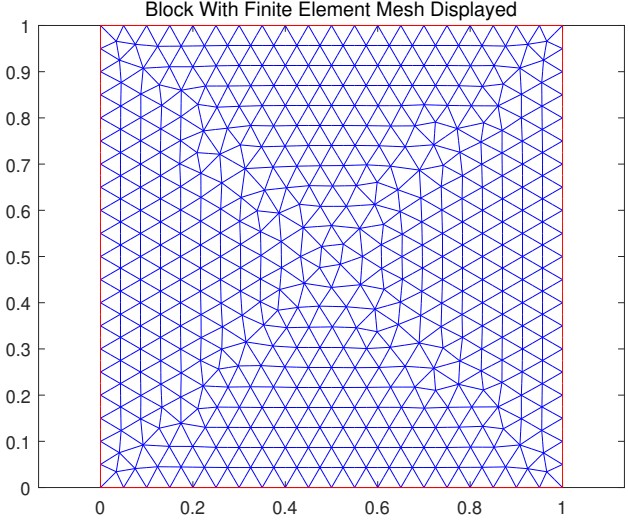

Figure 3: FEM mesh for the Poisson equation

2. Equation Discretization: Utilize the differential form of operators or select basis functions compatible with the grid to transform the PDE into a discrete problem. Essentially, this maps the PDE from an infinite-dimensional Hilbert space to a finite-dimensional representation.

3. Matrix Assembly: If dealing with linear PDEs, the discretized PDE and its boundary conditions can be converted into a system of linear equations in the finite-dimensional space. The system is assembled based on the chosen grid type. For nonlinear PDEs, methods similar to Newton's iteration are typically employed, transforming the problem into a series of linear systems to be solved iteratively (Ciarlet, 2002; Knoll & Keyes, 2004).

4. Applying Boundary Conditions: Discretize the boundary conditions on the grid and incorporate them into the linear systems.

5. Solving the System of Linear Equations: This step is generally the most time-consuming part of the entire algorithm.

6. Obtaining the Numerical Solution: Map the solution of the system of linear equations back onto the domain of the original PDE using the specific grid and corresponding basis functions, thereby obtaining the numerical solution.

### A.2 SPECIFIC EXAMPLES

To illustrate how the FDM can transform the Poisson equation into a system of linear equations, let's consider a concrete and straightforward example (Smith, 1985). Assume we aim to solve the Poisson equation in a two-dimensional space:

$$\nabla^2 u(x, y) = f(x, y).$$

This equation is defined over a square domain $[a, b] \times [c, d]$ with given boundary conditions. The process can be broken down into the following steps:

1. Mesh Generation: In the x-direction, select points $x_i = a + i\Delta x$, where $i = 0, 1, \ldots, N_x$. In the y-direction, select points $y_j = c + j\Delta y$, where $j = 0, 1, \ldots, N_y$.

2. Equation Discretization: Discretize the Poisson equation using a central difference scheme. For a grid point $(x_i, y_j)$, the discretized equation becomes:

$$\frac{u_{i+1,j} - 2u_{i,j} + u_{i-1,j}}{\Delta x^2} + \frac{u_{i,j+1} - 2u_{i,j} + u_{i,j-1}}{\Delta y^2} = f_{i,j}$$

Here, $u_{i,j}$ is an approximation of $u(x_i, y_j)$, and $f_{i,j} = f(x_i, y_j)$. This equation can be rewritten as:

$$-2u_{i,j}\left(\frac{1}{\Delta x^2} + \frac{1}{\Delta y^2}\right) + u_{i+1,j}\frac{1}{\Delta x^2} + u_{i-1,j}\frac{1}{\Delta x^2} + u_{i,j+1}\frac{1}{\Delta y^2} + u_{i,j-1}\frac{1}{\Delta y^2} = f_{i,j}$$

3. Matrix Assembly: Each equation corresponding to an internal point $(i, j)$ contributes one row to the matrix $\boldsymbol{A}$ and the vector $\boldsymbol{b}$. The corresponding row in $\boldsymbol{A}$ contains all the coefficients for $u_{i,j}$, while the element in $\boldsymbol{b}$ is $f_{i,j}$.

4. Applying Boundary Conditions: Implement the given boundary conditions, which may be of Dirichlet type (specifying $\boldsymbol{u}$ values) or Neumann type (specifying the derivative of $\boldsymbol{u}$). Adjust the corresponding elements in the vector $\boldsymbol{b}$ by adding or subtracting terms related to these boundary conditions.

5. Solving the System of Linear Equations $\boldsymbol{Ax} = \boldsymbol{b}$ : Due to the large, sparse nature of the matrix, iterative methods are generally used to solve the system of equations (Saad, 2003; Greenbaum, 1997).

6. Obtaining the Numerical Solution: Based on the difference scheme, map the solution of the system of linear equations from the discrete finite-dimensional space to the function space of the PDE to obtain the numerical solution. In this example, $u_{i,j}$ approximates $u(x_i, y_j)$, so a interpolation step can be used to finalize the solution.

## B  ALGORITHMIC DETAILS

### B.1   COMPUTES MATRICES $\boldsymbol{U}_k$ AND $\boldsymbol{C}_k$

The following computational procedure is adapted from De Sturler (1999); Parks et al. (2006).

GCRO-DR can be modified to solve (1) by carrying over $\boldsymbol{U}_k$ from the $i$-th system to the $(i + 1)$-th system. We have the relation $\boldsymbol{A}^{(i)}\boldsymbol{U}_k = \boldsymbol{C}_k$. We modify $\boldsymbol{U}_k$ and $\boldsymbol{C}_k$ to satisfy

$$\boldsymbol{A}^{(i)}\boldsymbol{U}_k = \boldsymbol{C}_k,$$
$$\boldsymbol{C}_k^H \boldsymbol{C}_k = \boldsymbol{I}_k,$$

with respect to $\boldsymbol{A}^{(i+1)}$ as follows:

$$[\boldsymbol{Q}, \boldsymbol{R}] = \text{reduced QR decomposition of } \boldsymbol{A}^{(i+1)}\boldsymbol{U}_k^{old},$$
$$\boldsymbol{C}_k^{new} = \boldsymbol{Q},$$
$$\boldsymbol{U}_k^{new} = \boldsymbol{U}_k^{old}\boldsymbol{R}^{-1}.$$

### B.2 GCRO-DR

The following computational procedure is adapted from Parks et al. (2006).

---

**Algorithm 2:** GCRO-DR

---

**1** Choose $m$, the maximum size of the subspace, and $k$, the desired number of approximate eigenvectors. Let $tol$ be the convergence tolerance. Choose an initial guess $\boldsymbol{x}_0$. Compute $\boldsymbol{r}_0 = \boldsymbol{b} - \boldsymbol{A}\boldsymbol{x}_0$, and set $i = 1$.

**2 if** $\widetilde{\boldsymbol{Y}}_k$ *is defined (from solving a previous linear system)* **then**

**3**      Let $[\boldsymbol{Q}, \boldsymbol{R}]$ be the reduced QR-factorization of $\boldsymbol{A}\widetilde{\boldsymbol{Y}}_k$.

**4**      $\boldsymbol{C}_k = \boldsymbol{Q}$

**5**      $\boldsymbol{U}_k = \widetilde{\boldsymbol{Y}}_k \boldsymbol{R}^{-1}$

**6**      $\boldsymbol{x}_1 = \boldsymbol{x}_0 + \boldsymbol{U}_k \boldsymbol{C}_k^H \boldsymbol{r}_0$

**7**      $\boldsymbol{r}_1 = \boldsymbol{r}_0 - \boldsymbol{C}_k \boldsymbol{C}_k^H \boldsymbol{r}_0$

**8 else**

**9**      $\boldsymbol{v}_1 = \boldsymbol{r}_0 / \|\boldsymbol{r}_0\|_2$

**10**      $\boldsymbol{c} = \|\boldsymbol{r}_0\|_2 \boldsymbol{e}_1$

**11**      Perform $m$ steps of GMRES, solving $\min \|\boldsymbol{c} - \underline{\boldsymbol{H}}_m \boldsymbol{y}\|_2$ for $\boldsymbol{y}$ and generating $\boldsymbol{V}_{m+1}$ and $\underline{\boldsymbol{H}}_m$.

**12**      $\boldsymbol{x}_1 = \boldsymbol{x}_0 + \boldsymbol{V}_m \boldsymbol{y}$

**13**      $\boldsymbol{r}_1 = \boldsymbol{V}_{m+1}(\boldsymbol{c} - \underline{\boldsymbol{H}}_m \boldsymbol{y})$

**14**      Compute the $k$ eigenvectors $\widetilde{\boldsymbol{z}}_j$ of $(\boldsymbol{H}_m + h_{m+1,m}^2 \boldsymbol{H}_m^{-H} \boldsymbol{e}_m \boldsymbol{e}_m^H) \widetilde{\boldsymbol{z}}_j = \widetilde{\theta}_j \widetilde{\boldsymbol{z}}_j$ associated with the smallest magnitude eigenvalues $\widetilde{\theta}_j$ and store in $\boldsymbol{P}_k$. $h_{m+1,m}$ is the element in row $m+1$ and column $m$ of matrix $\underline{\boldsymbol{H}}_m$.

**15**      $\widetilde{\boldsymbol{Y}}_k = \boldsymbol{V}_m \boldsymbol{P}_k$

**16**      Let $[\boldsymbol{Q}, \boldsymbol{R}]$ be the reduced QR-factorization of $\underline{\boldsymbol{H}}_m \boldsymbol{P}_k$ .

**17**      $\boldsymbol{C}_k = \boldsymbol{V}_{m+1} \boldsymbol{Q}$

**18**      $\boldsymbol{U}_k = \widehat{\boldsymbol{Y}}_k \boldsymbol{R}^{-1}$

**19 while** $\|\boldsymbol{r}_i\|_2 > tol$ **do**

**20**      $i = i + 1$

**21**      Perform $m - k$ Arnoldi steps with the linear operator $(\boldsymbol{I} - \boldsymbol{C}_k \boldsymbol{C}_k^H)\boldsymbol{A}$, letting $\boldsymbol{v}_1 = \boldsymbol{r}_{i-1}/\|\boldsymbol{r}_{i-1}\|_2$ and generating $\boldsymbol{V}_{m-k+1}$, $\underline{\boldsymbol{H}}_{m-k}$ and $\boldsymbol{B}_{m-k}$ .

**22**      Let $\boldsymbol{D}_k$ be a diagonal scaling matrix such that $\widetilde{\boldsymbol{U}}_k = \boldsymbol{U}_k \boldsymbol{D}_k$, where the columns of $\widetilde{\boldsymbol{U}}_k$ have unit norm.

**23**      $\widehat{\boldsymbol{V}}_m = [\widetilde{\boldsymbol{U}}_k \ \boldsymbol{V}_{m-k}]$

**24**      $\widehat{\boldsymbol{W}}_{m+1} = [\boldsymbol{C}_k \ \boldsymbol{V}_{m-k+1}]$

**25**      $\underline{\boldsymbol{G}}_m = \begin{bmatrix} \boldsymbol{D}_k & \boldsymbol{B}_{m-k} \\ 0 & \widetilde{\boldsymbol{H}}_{m-k} \end{bmatrix}$

**26**      Solve $\min \|\widehat{\boldsymbol{W}}_{m+1}^H \boldsymbol{r}_{i-1} - \underline{\boldsymbol{G}}_m \boldsymbol{y}\|_2$ for $\boldsymbol{y}$.

**27**      $\boldsymbol{x}_i = \boldsymbol{x}_{i-1} + \widehat{\boldsymbol{V}}_m \boldsymbol{y}$

**28**      $\boldsymbol{r}_i = \boldsymbol{r}_{i-1} - \widehat{\boldsymbol{W}}_{m+1} \underline{\boldsymbol{G}}_m \boldsymbol{y}$

**29**      Compute the $k$ eigenvectors $\widetilde{\boldsymbol{z}}_i$ of $\underline{\boldsymbol{G}}_m^H \underline{\boldsymbol{G}}_m \widetilde{\boldsymbol{z}}_i = \widetilde{\theta}_i \underline{\boldsymbol{G}}_m^H \widehat{\boldsymbol{W}}_{m+1}^H \widehat{\boldsymbol{V}}_m \widetilde{\boldsymbol{z}}_i$ associated with smallest magnitude eigenvalues $\widetilde{\theta}_i$ and store in $\boldsymbol{P}_k$ .

**30**      $\widetilde{\boldsymbol{Y}}_k = \widehat{\boldsymbol{V}}_m \boldsymbol{P}_k$

**31**      Let $[\boldsymbol{Q}, \boldsymbol{R}]$ be the reduced QR-factorization of $\underline{\boldsymbol{G}}_m \boldsymbol{P}_k$.

**32**      $\boldsymbol{C}_k = \widehat{\boldsymbol{W}}_{m+1} \boldsymbol{Q}$

**33**      $\boldsymbol{U}_k = \widetilde{\boldsymbol{Y}}_k \boldsymbol{R}^{-1}$

**34** Let $\widetilde{\boldsymbol{Y}}_k = \boldsymbol{U}_k$ (for the next system)

---

## C SUPPLEMENTARY THEORETICAL ANALYSIS

### C.1 SUPERLINEAR CONVERGENCE PHENOMENON

The phenomenon of superlinear convergence is frequently observed in the iterative processes of algorithms like GMRES (Van der Vorst & Vuik, 1993; Moret, 1997; Simoncini & Szyld, 2005). The introduction of a deflation space enables these iterative algorithms to swiftly transition into the superlinear convergence phase, thereby expediting their performance (Gaul et al., 2013; Nicolaides, 1987). The principle of recycling essentially embodies this concept, leveraging deflation space for acceleration. This phenomenon has been discerned in algorithms associated with recycling as well (Gaul, 2014).

For the problem under consideration, when a sequence of linear systems manifests inherent correlations in their sequential order, an apt deflation space can be curated. This ensures a rapid entry into the superlinear convergence phase, bypassing the initial slower convergence stages, and thereby enhancing the overall convergence velocity. Empirical data from our experiments underscores this observation. Theoretically, we can further dissect the potential reductions in iterations offered by SKR. The specific experimental phenomenon is illustrated in D.5.1 D.5.2.

## D DETAILS OF EXPERIMENTAL DATA

### D.1 SPECIFIC PARAMETERS OF THE MAIN EXPERIMENT

**Baseline**: To ensure optimal speed, we first generated PDE linear equation systems using either Python or Matlab. Then solve them in the c programming environment. We utilized the latest version from PETSc 3.19.4 for GMRES.

**Three Perspectives**:

1. **Precondition**: When solving large matrices, effective preconditioning can greatly accelerate the resolution process and enhance the stability of the algorithm. Recognizing that different scenarios require distinct preconditioning approaches, we conducted tests with more than ten of the most commonly used methods.

2. **Tolerance**: The tolerance of a solution inherently determines the iteration count and, by extension, the time required for a solution. Distinct algorithms exhibit varied convergence rates, and specific NOs hold unique tolerance standards. To ensure an encompassing comparison, we evaluated computational durations at diverse tolerances, zeroing in on 5-8 optimal error precisions for our tests.

3. **Matrix Dimensionality**: Algorithmic performance tends to fluctuate based on the matrix's dimensionality. Moreover, varying NOs mandate matrices of disparate dimensions. Hence, our study incorporated five distinct matrix dimensions.

**Two Performance Metrics**: 1. **Computational Duration**: This serves as the most unambiguous metric to gauge an algorithm's efficacy. 2. **Iteration Count**: Algorithmic stability is emblematic of its numerical sensitivity, with the iteration count offering direct insights into said stability.

### D.2 DATA SET

1. Darcy Flow

We consider two-dimensional Darcy flows, which can be described by the following equation (Li et al., 2020; Rahman et al., 2022; Kovachki et al., 2021; Lu et al., 2022):

$$-\nabla \cdot (K(x, y)\nabla h(x, y)) = f,$$

where $K$ is the permeability field, $h$ is the pressure, and $f$ is a source term which can be either a constant or a space-dependent function. In our experiment, $K(x, y)$ is derived using the Gaussian Random Field (GRF) method. The parameters inherent to the GRF serve as the foundation for our sort scheme.

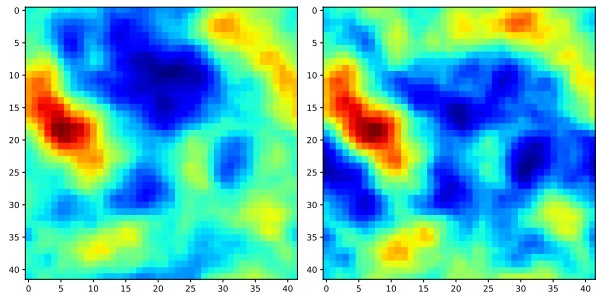

Figure 4: Solutions of Darcy flow equations with close parameters

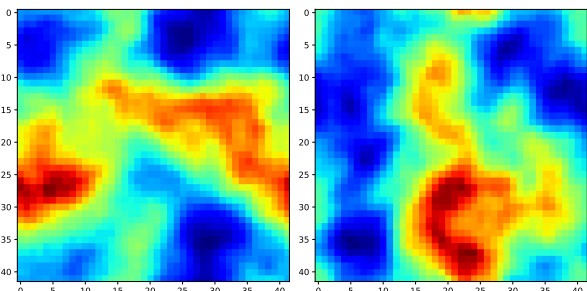

Figure 5: Solutions of Darcy flow equations with divergent parameters

As illustrated, with the matrix two-norm serving as the metric for distance, the first Figure 4 presents the solution for the Darcy Flow problem with two NO parameters that are very close. The subsequent Figure 5 depicts the solution for the Darcy Flow problem where the two NO parameters differ significantly. Clearly, when the parameters are closer, there's a strong correlation between the PDE and its solution, which underpins our sorting algorithm.

2. Thermal Problem

We consider a two-dimensional thermal steady state equation, which can be described by the following equation (Sharma et al., 2018; Koric & Abueidda, 2023):

$$\frac{\partial^2 T}{\partial x^2} + \frac{\partial^2 T}{\partial y^2} = 0,$$

where $T$ is the temperature. We examine the steady-state thermal equation in thermodynamics. The temperatures on the left and right boundaries are determined by random values ranging from -100 to 0 and 0 to 100, respectively. These temperature values are fundamental to our sort approach.

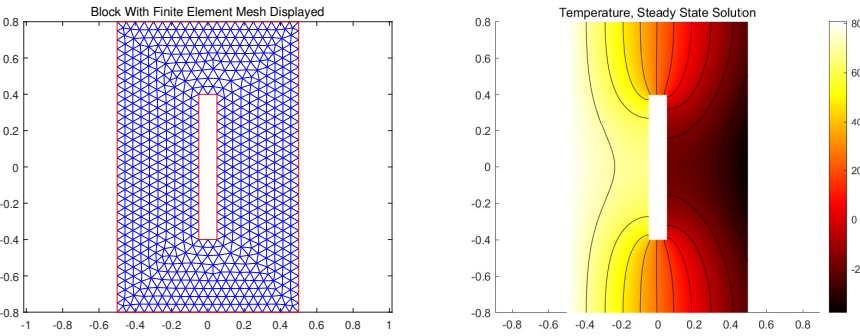

Figure 6: Finite element mesh and solution for the thermal problem

As shown in the Figure 6, for the 'Thermal Problem', we chose an irregular boundary to test the performance of our algorithm. The left image displays the finite element mesh obtained using FEM, an essential Step A in converting the PDE into a system of linear equations, as previously mentioned. The right image showcases the solution of one of the PDEs.

3. Poisson Equation

We consider a two-dimensional Poisson equation, which can be described by the following equation (Hsieh et al., 2019; Zhang et al., 2022):

$$\nabla^2 u = f.$$

Physical Contexts in which the Poisson Equation Appears: 1. Electrostatics; 2. Gravitation; 3. Fluid Dynamics.

We address the Poisson equation within a square domain. The boundary conditions on all four sides, as well as the $f$ value on the left side of the equation, are generated using truncated Chebyshev polynomials. The coefficients of these five Chebyshev polynomials are the basis for our sorting (Driscoll et al., 2014).

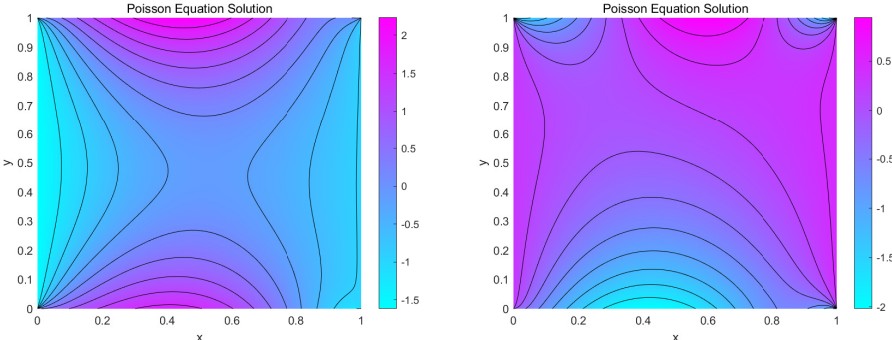

Figure 7: Solution of two PDEs before sorting

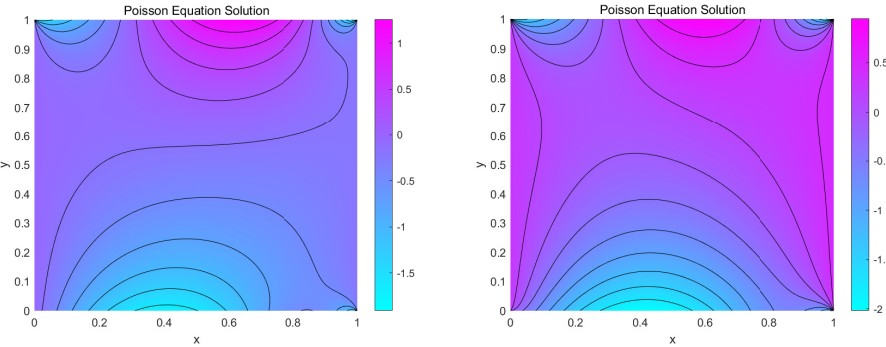

Figure 8: Solution of two PDEs after sorting

As depicted in the figure, with the matrix two-norm serving as the metric for distance, we provide visual representations of the Poisson Equation solutions featured in this study. The first Figure 7 displays the solutions of the two adjacent equations before sorting, while the second Figure 8 shows the solutions of the two adjacent equations after sorting. It is evident that the sorting algorithm has enhanced the correlation between the preceding and following PDEs.

4. Helmholtz Equation We consider a two-dimensional Helmholtz equation, which can be described by the following equation (Zhang et al., 2022):

$$\nabla^2 u + k^2 u = 0,$$

Physical Contexts in which the Helmholtz Equation Appears: 1. Acoustics; 2. Electromagnetism; 3. Quantum Mechanics.

In Helmholtz Equation, $k$ is the wavenumber, related to the frequency of the wave and the properties of the medium in which the wave is propagating. In our experiment, $k$ is derived using the GRF method. The parameters inherent to the GRF serve as the foundation for our sort scheme.

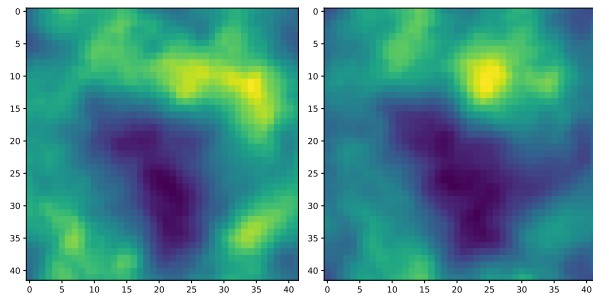

Figure 9: Solutions of Helmholtz equations with close parameters

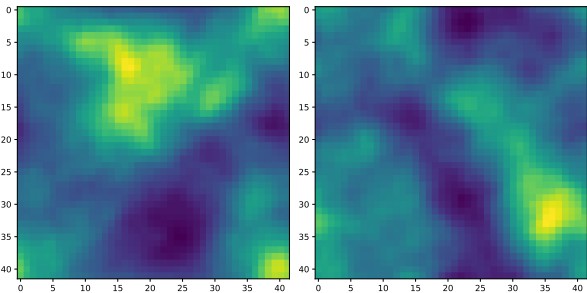

Figure 10: Solutions of Helmholtz equations with divergent parameters

As depicted in the Figure 9 10, with the matrix two-norm serving as the metric for distance, a comparable pattern emerges with the Helmholtz equations. Specifically, when the parameters of the neural operator are closely matched in Helmholtz equations, a pronounced correlation exists between the PDE and its solution. In contrast, this correlation diminishes for equations with significantly differing parameters.

### D.3 PRECONDITION

The experiments in this paper involve the following preconditioning techniques.

1. None: When preconditioning is set to 'none', essentially, no preconditioning operation is performed. This implies that the linear system is iteratively solved as-is, without any prior transformations or modifications.

2. Diagonal Preconditioning (Jacobi) (Saad, 2003): This is a simple preconditioning approach that considers only the diagonal elements of the coefficient matrix. The preconditioner matrix is the inverse of the diagonal elements of the coefficient matrix.

3. Block Jacobi (BJacobi) (Benzi et al., 1996): An extension of the Jacobi preconditioning, where the coefficient matrix is broken down into smaller blocks, each corresponding to a subdomain or subproblem. Each block is preconditioned independently using its diagonal part.

4. Successive Over-relaxation (SOR) (Young, 1954): The SOR method is a variant of the Gauss-Seidel iteration, introducing a relaxation factor to accelerate convergence. The preconditioner transforms the original problem into a weighted new problem, typically aiding in accelerating convergence for some iterative methods.

5. Additive Schwarz Method (ASM) (Toselli & Widlund, 2004): ASM is a domain decomposition method, where the original problem is split into multiple subdomains or subproblems. Problems of

each subdomain are solved independently, and these local solutions are then combined into a global solution.

6. Incomplete Cholesky (ICC) (Lin & Moré, 1999): ICC is a preconditioning method based on the Cholesky decomposition, but drops certain off-diagonal elements during the decomposition, making it "incomplete". It's utilized for symmetric positive definite problems.

7. Incomplete LU (ILU) (Saad, 2003): ILU is based on LU decomposition, but like ICC, drops certain off-diagonal elements during the decomposition. ILU can be applied to nonsymmetric problems.

### D.4 ENVIRONMENT

To ensure consistency in our evaluations, all comparative experiments were conducted under uniform computing environments. Specifically, the environments used are detailed as follows:

1. Environment (Env1):
   - Platform: Docker version 20.10.0
   - Operating System: Ubuntu 22.04.3 LTS
   - Processor: Dual-socket Intel® Xeon® Gold 6154 CPU, clocked at 3.00GHz
2. Environment (Env2):
   - Platform: Windows 11, version 21H2, WSL
   - Operating System: Ubuntu 22.04.3 LTS
   - Processor: 13th Gen Intel® Core™ i7-13700KF, clocked at 3.40 GHz

### D.5 ANALYSIS OF RELEVANT EXPERIMENTAL RESULTS

### D.5.1 CONVERGENCE SPEED ANALYSIS (TIME)

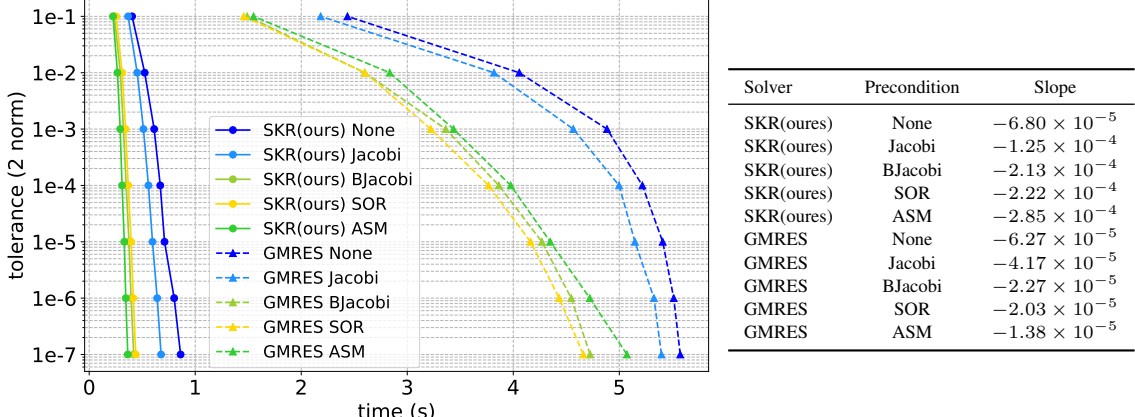

| Solver | Precondition | Slope |
|--------|--------------|-------|
| SKR(ours) | None | $-6.80 \times 10^{-5}$ |
| SKR(ours) | Jacobi | $-1.25 \times 10^{-4}$ |
| SKR(ours) | BJacobi | $-2.13 \times 10^{-4}$ |
| SKR(ours) | SOR | $-2.22 \times 10^{-4}$ |
| SKR(ours) | ASM | $-2.85 \times 10^{-4}$ |
| GMRES | None | $-6.27 \times 10^{-5}$ |
| GMRES | Jacobi | $-4.17 \times 10^{-5}$ |
| GMRES | BJacobi | $-2.27 \times 10^{-5}$ |
| GMRES | SOR | $-2.03 \times 10^{-5}$ |
| GMRES | ASM | $-1.38 \times 10^{-5}$ |

Figure 11: Illustrating the Helmholtz Equation with a matrix size of $10^4$, the graphic delves into the relationship between computational accuracy and average computation time. The left plot displays convergence curves under various preconditions, with the x-axis as average computation time and the y-axis as computational accuracy. The right plot presents linear fits for the three points with the minimum tolerance for each precondition, providing slopes as metrics for their convergence trends.

It is evident that the computational speed of our SKR algorithm far exceeds that of the GMRES algorithm. GMRES and similar Krylov subspace convergence algorithms exhibit a phenomenon known as superlinear convergence phenomenon in Appendix C.1, which means in a log-error versus time or iterations graph, the GMRES convergence curve can be interpreted as a combination of two segments (the first with a smaller absolute slope value, followed by a larger one). Due to the presence of superlinear convergence, an equitable and objective assessment of our SKR and GMRES in the high-precision phase requires discarding the data from the low-precision phase. Therefore,

we selected three high-precision points for each algorithm to perform a linear fit and obtain the convergence slopes for this phase. A comparative analysis revealed that our algorithm's slope has a greater absolute value, indicating that our SKR also converges more quickly than GMRES in the high-precision stage.

### D.5.2 CONVERGENCE SPEED ANALYSIS (ITERATION)

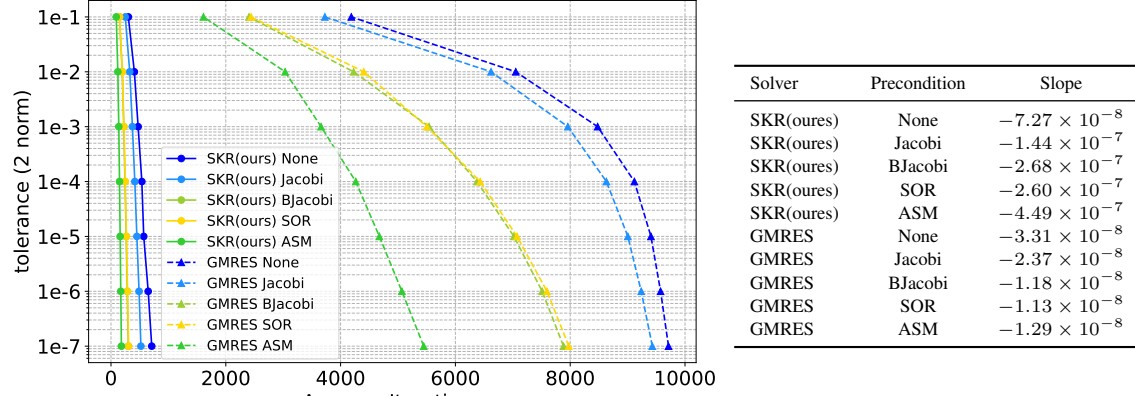

| Solver | Precondition | Slope |
|---|---|---|
| SKR(oures) | None | $-7.27 \times 10^{-8}$ |
| SKR(oures) | Jacobi | $-1.44 \times 10^{-7}$ |
| SKR(oures) | BJacobi | $-2.68 \times 10^{-7}$ |
| SKR(oures) | SOR | $-2.60 \times 10^{-7}$ |
| SKR(oures) | ASM | $-4.49 \times 10^{-7}$ |
| GMRES | None | $-3.31 \times 10^{-8}$ |
| GMRES | Jacobi | $-2.37 \times 10^{-8}$ |
| GMRES | BJacobi | $-1.18 \times 10^{-8}$ |
| GMRES | SOR | $-1.13 \times 10^{-8}$ |
| GMRES | ASM | $-1.29 \times 10^{-8}$ |

Figure 12: Illustrating the Helmholtz Equation with a matrix size of $10^4$, the graphic delves into the relationship between computational accuracy and average iteration count. The left plot displays convergence curves under various preconditions, with the x-axis as average iteration count and the y-axis as computational accuracy. The right plot presents linear fits for the three points with the minimum tolerance for each precondition, providing slopes as metrics for their convergence trends.

It is evident that, at the same level of precision, our SKR algorithm requires significantly fewer iterations than GMRES. This strongly supports the notion that SKR achieves acceleration by expediting the convergence of the Krylov subspace, thereby reducing the number of iterations. As mentioned in Appendix D.5.1, to appraise the convergence rates of both algorithms more equitably during the high-precision phase, we conducted linear fits using three high-precision data points. The comparison reveals that the absolute value of the slope during the high-precision phase is greater for our SKR algorithm than for GMRES, indicating a faster rate of convergence. This robustly validates the superior stability of the SKR algorithm compared to GMRES.

### D.5.3 STABILITY ANALYSIS

Based on the Figure 13, we can draw the following conclusions: 1. The use of the SKR algorithm almost never reaches the maximum iteration count, indicating its excellent iterative stability. 2. GMRES, on the other hand, frequently reaches the maximum iteration count without converging, and the effectiveness of different preconditions varies significantly.

Combining these Results D.5.2 with the previous experiment, it is evident that our SKR algorithm exhibits much greater stability compared to the GMRES algorithm.

### D.6 DETAILED EXPERIMENTAL DATA

We conducted nearly 3,000 experiments. Each experiment employed a dataset specifically designed to mimic genuine NO training data. Below are some actual data from these experiments.

The title of each table below specifies the corresponding experimental dataset, preprocessing method, experimental environment, and details of MPI parallelization. For instance, 'Table 3: Darcy Flow, None, Env1, MPI72' indicates that the experimental results for the Darcy Flow problem are recorded, with no matrix preprocessing algorithm applied, all within computing environment Env1, and all experiments utilizing MPI with 72 parallel threads.

As another example, 'Table 17: Poisson Equation, None, Env1, 7153: MPI10, 11237: MPI10, 20245: MPI20, 45337: MPI5, 71313: MPI5' details that the table records the experimental results

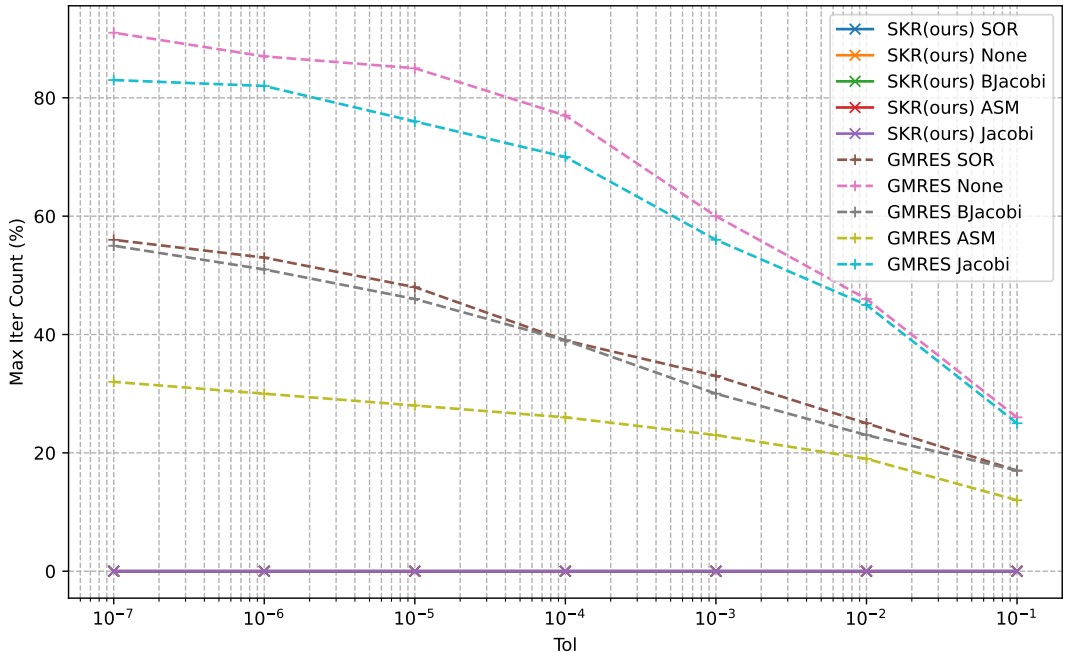

Figure 13: Illustrating the proportion of instances where different algorithms reach the maximum iteration count, the graphic delves into performance under various precisions for the Darcy flow problem with a matrix size of $10^4$ and a maximum iteration count of $10^4$.

for the Poisson Equation, with no matrix preprocessing selected and all within the computing environment Env1. Here, experiments with matrix sizes of 7153 and 11237 were run with MPI using 10 parallel threads, size 20245 with MPI using 20 threads, and sizes 45337 and 71313 with MPI using 5 threads.

These tables are divided into two parts:

- the upper half records the average time to solve each linear system for two different algorithms under various conditions, in seconds, with smaller numbers indicating faster computation.
- The lower half records the average number of iterations needed to solve each linear system for the two algorithms under different conditions, with fewer iterations indicating better algorithm stability and faster convergence.

The first row indicates different required error precisions, and the second column shows the matrix sizes derived from different grid densities. Each row's data is composed of two parts, with the upper half showing the data for the GMRES algorithm—our baseline—and the lower half showing the corresponding data for our SKR algorithm.

Table 3: Darcy Flow, None, Env1, MPI72

| | n | solver\tol | 1.E-01 | 1.E-02 | 1.E-03 | 1.E-04 | 1.E-05 | 1.E-06 | 1.E-07 | 1.E-08 |
|---|---|---|---|---|---|---|---|---|---|---|
| time | 2500 | GMRES | 0.13 | 0.20 | 0.24 | 0.27 | 0.29 | 0.31 | 0.34 | 0.36 |
| | | SKR | 0.08 | 0.09 | 0.10 | 0.11 | 0.11 | 0.12 | 0.13 | 0.13 |
| | 6400 | GMRES | 0.28 | 0.45 | 0.55 | 0.61 | 0.66 | 0.69 | 0.71 | 0.72 |
| | | SKR | 0.14 | 0.17 | 0.19 | 0.21 | 0.23 | 0.24 | 0.25 | 0.27 |
| | 10000 | GMRES | 0.39 | 0.70 | 0.79 | 0.85 | 0.88 | 0.89 | 0.90 | 0.91 |
| | | SKR | 0.21 | 0.28 | 0.32 | 0.36 | 0.41 | 0.45 | 0.53 | 0.52 |
| | 22500 | GMRES | 3.89 | 5.62 | 6.00 | 6.10 | 6.19 | 6.28 | – | – |
| | | SKR | 1.10 | 2.06 | 2.48 | 2.97 | 3.32 | 3.62 | – | – |
| | 40000 | GMRES | 26.28 | 51.20 | 57.63 | 60.89 | 62.88 | 64.83 | – | – |
| | | SKR | 15.19 | 35.80 | 32.60 | 41.01 | 44.22 | 42.63 | – | – |
| iter | 2500 | GMRES | 1,394 | 2,575 | 3,115 | 3,608 | 3,983 | 4,325 | 4641 | 4963 |
| | | SKR | 96 | 131 | 151 | 167 | 183 | 199 | 213 | 227 |
| | 6400 | GMRES | 3,192 | 5,402 | 6,709 | 7,617 | 8,193 | 8,582 | 8847 | 9059 |
| | | SKR | 218 | 281 | 321 | 353 | 387 | 415 | 443 | 473 |
| | 10000 | GMRES | 3,874 | 7,457 | 8,426 | 9,022 | 9,375 | 9,596 | 9713 | 9775 |
| | | SKR | 313 | 446 | 520 | 588 | 682 | 758 | 898 | 881 |
| | 22500 | GMRES | 6,225 | 8,971 | 9,614 | 9,818 | 9,891 | 9,956 | – | – |
| | | SKR | 745 | 1,445 | 1,763 | 2,122 | 2,367 | 2,600 | – | – |
| | 40000 | GMRES | 38,896 | 76,287 | 85,637 | 90,043 | 93,298 | 95,869 | – | – |
| | | SKR | 9,283 | 21,950 | 19,994 | 25,212 | 27,207 | 26,177 | – | – |

Table 4: Darcy Flow, Jacobi, Env1, MPI72

| | n | solver\tol | 1.E-01 | 1.E-02 | 1.E-03 | 1.E-04 | 1.E-05 | 1.E-06 | 1.E-07 | 1.E-08 |
|---|---|---|---|---|---|---|---|---|---|---|
| time | 2500 | GMRES | 0.13 | 0.19 | 0.22 | 0.24 | 0.26 | 0.28 | 0.30 | 0.31 |
| | | SKR | 0.07 | 0.09 | 0.09 | 0.10 | 0.10 | 0.11 | 0.11 | 0.12 |
| | 6400 | GMRES | 0.25 | 0.44 | 0.52 | 0.58 | 0.62 | 0.65 | 0.68 | 0.70 |
| | | SKR | 0.12 | 0.15 | 0.17 | 0.18 | 0.19 | 0.20 | 0.22 | 0.23 |
| | 10000 | GMRES | 0.35 | 0.64 | 0.76 | 0.82 | 0.85 | 0.88 | 0.90 | 0.91 |
| | | SKR | 0.18 | 0.23 | 0.26 | 0.29 | 0.31 | 0.33 | 0.34 | 0.38 |
| | 22500 | GMRES | 0.70 | 1.24 | 1.33 | 1.38 | 1.40 | 1.40 | – | – |
| | | SKR | 0.67 | 0.92 | 1.26 | 1.58 | 1.75 | 1.95 | – | – |
| | 40000 | GMRES | 23.72 | 47.98 | 53.99 | 57.54 | 60.86 | 62.79 | – | – |
| | | SKR | 17.09 | 25.65 | 25.81 | 28.98 | 29.86 | 29.45 | – | – |
| iter | 2500 | GMRES | 1370 | 2357 | 2786 | 3110 | 3407 | 3695 | 3953 | 4192 |
| | | SKR | 81 | 110 | 126 | 140 | 152 | 164 | 176 | 187 |
| | 6400 | GMRES | 2741 | 5235 | 6271 | 7074 | 7621 | 8037 | 8355 | 8622 |
| | | SKR | 173 | 231 | 261 | 286 | 311 | 332 | 356 | 376 |
| | 10000 | GMRES | 3370 | 6642 | 7949 | 8611 | 8987 | 9322 | 9503 | 9610 |
| | | SKR | 252 | 347 | 400 | 440 | 477 | 522 | 549 | 613 |
| | 22500 | GMRES | 4770 | 8741 | 9417 | 9778 | 9900 | 9921 | – | – |
| | | SKR | 812 | 1148 | 1606 | 2024 | 2256 | 2526 | – | – |
| | 40000 | GMRES | 35088 | 70796 | 80000 | 85202 | 89638 | 92399 | – | – |
| | | SKR | 10523 | 15770 | 15762 | 17713 | 18395 | 18073 | – | – |

Table 5: Darcy Flow, BJacobi, Env1, MPI72

| | n | solver\tol | 1.E-01 | 1.E-02 | 1.E-03 | 1.E-04 | 1.E-05 | 1.E-06 | 1.E-07 | 1.E-08 |
|---|---|---|---|---|---|---|---|---|---|---|
| time | 2500 | GMRES | 0.09 | 0.14 | 0.17 | 0.20 | 0.22 | 0.23 | 0.25 | 0.27 |
| | | SKR | 0.07 | 0.08 | 0.08 | 0.08 | 0.09 | 0.09 | 0.10 | 0.10 |
| | 6400 | GMRES | 0.22 | 0.39 | 0.49 | 0.56 | 0.60 | 0.64 | 0.67 | 0.70 |
| | | SKR | 0.10 | 0.12 | 0.13 | 0.14 | 0.15 | 0.16 | 0.16 | 0.17 |
| | 10000 | GMRES | 0.30 | 0.59 | 0.70 | 0.78 | 0.85 | 0.89 | 0.93 | 0.97 |
| | | SKR | 0.13 | 0.16 | 0.18 | 0.19 | 0.21 | 0.22 | 0.23 | 0.24 |
| | 22500 | GMRES | 0.49 | 1.08 | 1.29 | 1.41 | 1.50 | 1.55 | – | – |
| | | SKR | 0.32 | 0.35 | 0.55 | 0.53 | 0.50 | 0.54 | – | – |
| | 40000 | GMRES | 15.43 | 32.62 | 32.00 | 36.61 | 49.46 | 51.52 | – | – |
| | | SKR | 4.70 | 9.14 | 8.35 | 11.76 | 12.55 | 14.58 | – | – |
| iter | 2500 | GMRES | 676 | 1289 | 1675 | 1934 | 2182 | 2411 | 2636 | 2814 |
| | | SKR | 57 | 76 | 87 | 96 | 104 | 113 | 121 | 129 |
| | 6400 | GMRES | 1901 | 3642 | 4779 | 5497 | 5934 | 6310 | 6614 | 6916 |
| | | SKR | 117 | 154 | 175 | 191 | 207 | 222 | 237 | 251 |
| | 10000 | GMRES | 2380 | 5015 | 5926 | 6712 | 7266 | 7705 | 8030 | 8347 |
| | | SKR | 145 | 205 | 232 | 256 | 277 | 297 | 317 | 339 |
| | 22500 | GMRES | 2667 | 6276 | 7520 | 8352 | 8839 | 9135 | – | – |
| | | SKR | 338 | 378 | 630 | 601 | 568 | 614 | – | – |
| | 40000 | GMRES | 21396 | 44977 | 44643 | 51229 | 68445 | 71642 | – | – |
| | | SKR | 2739 | 5382 | 4980 | 7030 | 7450 | 8713 | – | – |

Table 6: Darcy Flow, SOR, Env1, MPI72

| | n | solver\tol | 1.E-01 | 1.E-02 | 1.E-03 | 1.E-04 | 1.E-05 | 1.E-06 | 1.E-07 | 1.E-08 |
|---|---|---|---|---|---|---|---|---|---|---|
| time | 2500 | GMRES | 0.08 | 0.13 | 0.16 | 0.18 | 0.19 | 0.20 | 0.21 | 0.23 |
| | | SKR | 0.06 | 0.08 | 0.08 | 0.08 | 0.09 | 0.09 | 0.09 | 0.10 |
| | 6400 | GMRES | 0.20 | 0.33 | 0.41 | 0.46 | 0.50 | 0.54 | 0.57 | 0.60 |
| | | SKR | 0.10 | 0.12 | 0.13 | 0.14 | 0.15 | 0.15 | 0.16 | 0.17 |
| | 10000 | GMRES | 0.28 | 0.53 | 0.62 | 0.69 | 0.74 | 0.79 | 0.82 | 0.85 |
| | | SKR | 0.13 | 0.16 | 0.18 | 0.19 | 0.20 | 0.22 | 0.22 | 0.24 |
| | 22500 | GMRES | 0.44 | 0.99 | 1.17 | 1.30 | 1.38 | 1.41 | – | – |
| | | SKR | 0.25 | 0.36 | 0.41 | 0.54 | 0.51 | 0.63 | – | – |
| | 40000 | GMRES | 15.72 | 35.73 | 41.95 | 46.75 | 50.32 | 52.57 | – | – |
| | | SKR | 8.63 | 8.68 | 11.27 | 16.21 | 11.08 | 12.65 | – | – |
| iter | 2500 | GMRES | 622 | 1393 | 1822 | 2084 | 2271 | 2483 | 2675 | 2852 |
| | | SKR | 57 | 76 | 87 | 96 | 105 | 113 | 121 | 129 |
| | 6400 | GMRES | 2059 | 3644 | 4688 | 5315 | 5833 | 6238 | 6611 | 6956 |
| | | SKR | 119 | 156 | 175 | 192 | 209 | 224 | 238 | 253 |
| | 10000 | GMRES | 2439 | 5120 | 6028 | 6718 | 7246 | 7778 | 8136 | 8467 |
| | | SKR | 150 | 206 | 236 | 259 | 281 | 303 | 320 | 341 |
| | 22500 | GMRES | 2602 | 6261 | 7536 | 8371 | 8882 | 9150 | – | – |
| | | SKR | 257 | 391 | 460 | 621 | 592 | 739 | – | – |
| | 40000 | GMRES | 22214 | 50690 | 59841 | 66630 | 71761 | 74751 | – | – |
| | | SKR | 5146 | 5136 | 6735 | 9743 | 6668 | 7635 | – | – |

Table 7: Darcy Flow, ASM, Env1, MPI72

|      | n     | solver\tol | 1.E-01 | 1.E-02 | 1.E-03 | 1.E-04 | 1.E-05 | 1.E-06 | 1.E-07 | 1.E-08 |
|------|-------|------------|--------|--------|--------|--------|--------|--------|--------|--------|
| time | 2500  | GMRES      | 0.05   | 0.07   | 0.08   | 0.09   | 0.10   | 0.11   | 0.12   | 0.13   |
|      |       | SKR        | 0.06   | 0.06   | 0.07   | 0.07   | 0.07   | 0.08   | 0.08   | 0.08   |
|      | 6400  | GMRES      | 0.14   | 0.21   | 0.30   | 0.37   | 0.42   | 0.48   | 0.52   | 0.57   |
|      |       | SKR        | 0.09   | 0.10   | 0.10   | 0.11   | 0.12   | 0.12   | 0.12   | 0.13   |
|      | 10000 | GMRES      | 0.29   | 0.56   | 0.67   | 0.75   | 0.82   | 0.89   | 0.95   | 1.01   |
|      |       | SKR        | 0.11   | 0.13   | 0.14   | 0.15   | 0.16   | 0.16   | 0.17   | 0.18   |
|      | 22500 | GMRES      | 0.51   | 1.14   | 1.46   | 1.69   | 1.84   | 1.95   | –      | –      |
|      |       | SKR        | 0.21   | 0.29   | 0.33   | 0.44   | 0.39   | 0.42   | –      | –      |
|      | 40000 | GMRES      | 19.54  | 42.58  | 50.62  | 57.46  | 61.99  | 66.30  | –      | –      |
|      |       | SKR        | 2.55   | 7.57   | 3.64   | 9.26   | 6.42   | 9.41   | –      | –      |
| iter | 2500  | GMRES      | 100    | 233    | 320    | 408    | 500    | 570    | 638    | 726    |
|      |       | SKR        | 32     | 42     | 48     | 53     | 58     | 62     | 67     | 71     |
|      | 6400  | GMRES      | 679    | 1199   | 1761   | 2215   | 2608   | 3003   | 3316   | 3631   |
|      |       | SKR        | 69     | 89     | 99     | 109    | 117    | 126    | 133    | 141    |
|      | 10000 | GMRES      | 1502   | 3144   | 3783   | 4284   | 4681   | 5101   | 5461   | 5820   |
|      |       | SKR        | 92     | 125    | 141    | 153    | 166    | 178    | 189    | 201    |
|      | 22500 | GMRES      | 1982   | 4748   | 6098   | 7088   | 7763   | 8236   | –      | –      |
|      |       | SKR        | 177    | 265    | 306    | 441    | 378    | 414    | –      | –      |
|      | 40000 | GMRES      | 18042  | 39800  | 47228  | 53725  | 57977  | 61608  | –      | –      |
|      |       | SKR        | 1204   | 3664   | 1735   | 4526   | 3112   | 4611   | –      | –      |

Table 8: Darcy Flow, ICC, Env2, No MPI

|      | n     | solver\tol | 1.E-01 | 1.E-02 | 1.E-03 | 1.E-04 | 1.E-05 | 1.E-06 | 1.E-07 | 1.E-08 |
|------|-------|------------|--------|--------|--------|--------|--------|--------|--------|--------|
| time | 2500  | GMRES      | 0.01   | 0.03   | 0.03   | 0.04   | 0.04   | 0.04   | 0.05   | 0.05   |
|      |       | SKR        | 0.02   | 0.03   | 0.04   | 0.04   | 0.04   | 0.05   | 0.05   | 0.05   |
|      | 6400  | GMRES      | 0.11   | 0.29   | 0.38   | 0.47   | 0.60   | 0.71   | 0.81   | 0.91   |
|      |       | SKR        | 0.11   | 0.15   | 0.16   | 0.18   | 0.19   | 0.20   | 0.21   | 0.22   |
|      | 10000 | GMRES      | 0.63   | 1.27   | 1.65   | 1.97   | 2.22   | 2.43   | 2.64   | 2.85   |
|      |       | SKR        | 0.24   | 0.31   | 0.35   | 0.37   | 0.40   | 0.43   | 0.45   | 0.48   |
|      | 22500 | GMRES      | 2.15   | 3.49   | 4.46   | 5.43   | 6.17   | 6.76   | –      | –      |
|      |       | SKR        | 0.56   | 0.77   | 0.87   | 0.95   | 1.03   | 1.10   | –      | –      |
|      | 40000 | GMRES      | 36.38  | 53.80  | 58.95  | 63.86  | 69.77  | 75.75  | –      | –      |
|      |       | SKR        | 1.72   | 2.70   | 3.07   | 14.62  | 3.65   | 3.97   | –      | –      |
| iter | 2500  | GMRES      | 32     | 149    | 172    | 196    | 221    | 245    | 270    | 298    |
|      |       | SKR        | 25     | 32     | 37     | 41     | 44     | 48     | 51     | 54     |
|      | 6400  | GMRES      | 236    | 629    | 825    | 1032   | 1320   | 1551   | 1787   | 2000   |
|      |       | SKR        | 49     | 66     | 74     | 82     | 88     | 94     | 100    | 106    |
|      | 10000 | GMRES      | 878    | 1769   | 2311   | 2752   | 3117   | 3405   | 3684   | 3985   |
|      |       | SKR        | 71     | 96     | 108    | 117    | 126    | 135    | 143    | 151    |
|      | 22500 | GMRES      | 1208   | 1964   | 2506   | 3051   | 3460   | 3797   | –      | –      |
|      |       | SKR        | 72     | 103    | 118    | 130    | 141    | 150    | –      | –      |
|      | 40000 | GMRES      | 11479  | 16958  | 18618  | 20189  | 22018  | 23811  | –      | –      |
|      |       | SKR        | 130    | 210    | 240    | 1184   | 288    | 313    | –      | –      |

Table 9: Darcy Flow, ILU, Env2, No MPI

| | n | solver\tol | 1.E-01 | 1.E-02 | 1.E-03 | 1.E-04 | 1.E-05 | 1.E-06 | 1.E-07 | 1.E-08 |
|---|---|---|---|---|---|---|---|---|---|---|
| time | 2500 | GMRES | 0.01 | 0.03 | 0.03 | 0.03 | 0.04 | 0.04 | 0.04 | 0.05 |
| | | SKR | 0.02 | 0.03 | 0.03 | 0.04 | 0.04 | 0.04 | 0.05 | 0.05 |
| | 6400 | GMRES | 0.10 | 0.27 | 0.37 | 0.47 | 0.56 | 0.64 | 0.72 | 0.80 |
| | | SKR | 0.11 | 0.14 | 0.16 | 0.17 | 0.18 | 0.19 | 0.20 | 0.22 |
| | 10000 | GMRES | 0.63 | 1.19 | 1.52 | 1.88 | 2.13 | 2.36 | 2.55 | 2.75 |
| | | SKR | 0.23 | 0.30 | 0.34 | 0.37 | 0.39 | 0.42 | 0.44 | 0.47 |
| | 22500 | GMRES | 2.38 | 3.96 | 4.86 | 5.90 | 6.69 | 7.42 | – | – |
| | | SKR | 0.66 | 0.89 | 1.01 | 1.10 | 1.19 | 1.28 | – | – |
| | 40000 | GMRES | 30.39 | 53.52 | 61.25 | 68.44 | 72.64 | 76.58 | – | – |
| | | SKR | 1.71 | 2.65 | 2.98 | 3.35 | 3.59 | 3.82 | – | – |
| iter | 2500 | GMRES | 30 | 141 | 157 | 179 | 199 | 219 | 242 | 261 |
| | | SKR | 24 | 30 | 34 | 38 | 42 | 45 | 48 | 51 |
| | 6400 | GMRES | 222 | 599 | 823 | 1034 | 1237 | 1409 | 1594 | 1760 |
| | | SKR | 49 | 63 | 72 | 79 | 85 | 92 | 97 | 103 |
| | 10000 | GMRES | 889 | 1670 | 2133 | 2653 | 2995 | 3328 | 3587 | 3836 |
| | | SKR | 71 | 94 | 105 | 115 | 123 | 132 | 140 | 148 |
| | 22500 | GMRES | 1414 | 2361 | 2896 | 3521 | 3997 | 4419 | – | – |
| | | SKR | 89 | 122 | 140 | 154 | 167 | 180 | – | – |
| | 40000 | GMRES | 9768 | 17195 | 19676 | 21936 | 23358 | 24658 | – | – |
| | | SKR | 127 | 204 | 231 | 256 | 277 | 300 | – | – |

Table 10: Thermal Problem, None, Env1, No MPI

| | n | solver\tol | 1.E-04 | 1.E-05 | 1.E-06 | 1.E-07 | 1.E-08 | 1.E-09 | 1.E-10 | 1.E-11 |
|---|---|---|---|---|---|---|---|---|---|---|
| time | 2755 | GMRES | – | 0.21 | 0.29 | 0.37 | 0.45 | 0.54 | 0.62 | 0.70 |
| | | SKR | – | 0.11 | 0.14 | 0.16 | 0.19 | 0.22 | 0.23 | 0.27 |
| | 7821 | GMRES | – | 1.74 | 2.53 | 3.32 | 4.12 | 4.93 | 5.73 | 6.53 |
| | | SKR | – | 0.44 | 0.58 | 0.71 | 0.85 | 1.01 | 1.13 | 1.26 |
| | 11063 | GMRES | – | 3.44 | 5.08 | 6.71 | 8.34 | 9.99 | 11.60 | 13.21 |
| | | SKR | – | 0.76 | 1.01 | 1.30 | 1.56 | 1.83 | 2.05 | 2.30 |
| | 17593 | GMRES | – | 7.48 | 11.29 | 15.11 | 18.92 | 22.57 | 26.04 | 26.50 |
| | | SKR | – | 1.57 | 2.04 | 2.51 | 3.19 | 3.70 | 4.29 | 4.96 |
| | 31157 | GMRES | 9.06 | 21.84 | 34.60 | 46.55 | – | – | – | – |
| | | SKR | 1.86 | 3.69 | 5.12 | 6.97 | – | – | – | – |
| | 70031 | GMRES | 27.23 | 85.47 | 105.81 | 106.87 | – | – | – | – |
| | | SKR | 5.54 | 13.85 | 20.37 | 25.65 | – | – | – | – |
| iter | 2755 | GMRES | – | 503 | 693 | 882 | 1075 | 1278 | 1476 | 1674 |
| | | SKR | – | 50 | 66 | 76 | 91 | 106 | 118 | 133 |
| | 7821 | GMRES | – | 1468 | 2140 | 2819 | 3500 | 4181 | 4864 | 5547 |
| | | SKR | – | 81 | 104 | 130 | 162 | 190 | 216 | 243 |
| | 11063 | GMRES | – | 2057 | 3035 | 4015 | 4996 | 5977 | 6946 | 7908 |
| | | SKR | – | 99 | 135 | 176 | 211 | 248 | 283 | 318 |
| | 17593 | GMRES | – | 2821 | 4262 | 5704 | 7144 | 8514 | 9830 | 10000 |
| | | SKR | – | 130 | 174 | 217 | 278 | 326 | 375 | 439 |
| | 31157 | GMRES | 1909 | 4592 | 7295 | 9816 | – | – | – | – |
| | | SKR | 85 | 178 | 249 | 345 | – | – | – | – |
| | 70031 | GMRES | 2531 | 7967 | 9888 | 10000 | – | – | – | – |
| | | SKR | 115 | 300 | 442 | 558 | – | – | – | – |

Table 11: Thermal Problem, Jacobi, Env1, No MPI

| | n | solver\tol | 1.E-04 | 1.E-05 | 1.E-06 | 1.E-07 | 1.E-08 | 1.E-09 | 1.E-10 | 1.E-11 |
|---|---|---|---|---|---|---|---|---|---|---|
| time | 2755 | GMRES | – | 0.20 | 0.26 | 0.31 | 0.37 | 0.43 | 0.50 | 0.58 |
| | | SKR | – | 0.11 | 0.15 | 0.16 | 0.19 | 0.20 | 0.23 | 0.26 |
| | 7821 | GMRES | – | 1.37 | 1.78 | 2.14 | 2.46 | 2.73 | 3.04 | 3.36 |
| | | SKR | – | 0.44 | 0.56 | 0.71 | 0.89 | 1.01 | 1.13 | 1.24 |
| | 11063 | GMRES | – | 2.49 | 3.35 | 4.12 | 4.80 | 5.42 | 6.04 | 6.69 |
| | | SKR | – | 0.75 | 1.01 | 1.30 | 1.57 | 1.77 | 2.02 | 2.28 |
| | 17593 | GMRES | – | 3.58 | 5.67 | 7.90 | 9.55 | 10.99 | 12.48 | 14.68 |
| | | SKR | – | 1.59 | 2.03 | 2.50 | 3.02 | 3.67 | 4.30 | 4.90 |
| | 31157 | GMRES | 5.72 | 9.97 | 17.77 | 22.71 | – | – | – | – |
| | | SKR | 1.70 | 3.77 | 5.18 | 6.99 | – | – | – | – |
| | 70031 | GMRES | 15.67 | 34.74 | 68.17 | 101.00 | – | – | – | – |
| | | SKR | 5.88 | 13.55 | 19.76 | 24.83 | – | – | – | – |
| iter | 2755 | GMRES | – | 455 | 606 | 732 | 857 | 994 | 1169 | 1347 |
| | | SKR | – | 48 | 65 | 76 | 91 | 101 | 114 | 129 |
| | 7821 | GMRES | – | 1142 | 1490 | 1788 | 2049 | 2283 | 2530 | 2801 |
| | | SKR | – | 81 | 104 | 131 | 164 | 190 | 214 | 238 |
| | 11063 | GMRES | – | 1471 | 1982 | 2435 | 2841 | 3202 | 3569 | 3960 |
| | | SKR | – | 98 | 134 | 174 | 210 | 242 | 276 | 311 |
| | 17593 | GMRES | – | 1334 | 2111 | 2941 | 3552 | 4090 | 4646 | 5466 |
| | | SKR | – | 132 | 172 | 214 | 262 | 321 | 379 | 428 |
| | 31157 | GMRES | 1186 | 2078 | 3713 | 4737 | – | – | – | – |
| | | SKR | 77 | 184 | 251 | 343 | – | – | – | – |
| | 70031 | GMRES | 1450 | 3216 | 5767 | 8316 | – | – | – | – |
| | | SKR | 122 | 291 | 431 | 541 | – | – | – | – |

Table 12: Thermal Problem, BJacobi, Env1, No MPI

| | n | solver\tol | 1.E-04 | 1.E-05 | 1.E-06 | 1.E-07 | 1.E-08 | 1.E-09 | 1.E-10 | 1.E-11 |
|---|---|---|---|---|---|---|---|---|---|---|
| time | 2755 | GMRES | – | 0.07 | 0.09 | 0.11 | 0.13 | 0.15 | 0.17 | 0.19 |
| | | SKR | – | 0.05 | 0.08 | 0.08 | 0.09 | 0.10 | 0.12 | 0.12 |
| | 7821 | GMRES | – | 0.45 | 0.58 | 0.72 | 0.84 | 0.97 | 1.10 | 1.23 |
| | | SKR | – | 0.23 | 0.31 | 0.33 | 0.36 | 0.43 | 0.46 | 0.53 |
| | 11063 | GMRES | – | 0.85 | 1.13 | 1.41 | 1.69 | 1.95 | 2.21 | 2.46 |
| | | SKR | – | 0.36 | 0.47 | 0.51 | 0.62 | 0.66 | 0.77 | 0.81 |
| | 17593 | GMRES | – | 1.55 | 2.10 | 2.65 | 3.18 | 3.72 | 4.27 | 4.81 |
| | | SKR | – | 0.72 | 0.95 | 1.03 | 1.23 | 1.42 | 1.50 | 1.70 |
| | 31157 | GMRES | 3.19 | 5.49 | 7.33 | 9.11 | – | – | – | – |
| | | SKR | 1.24 | 1.49 | 1.99 | 2.47 | – | – | – | – |
| | 70031 | GMRES | 13.00 | 21.62 | 27.71 | 34.00 | – | – | – | – |
| | | SKR | 2.97 | 5.09 | 7.03 | 8.86 | – | – | – | – |
| iter | 2755 | GMRES | – | 114 | 154 | 188 | 228 | 266 | 306 | 344 |
| | | SKR | – | 21 | 26 | 31 | 35 | 40 | 45 | 49 |
| | 7821 | GMRES | – | 289 | 376 | 466 | 549 | 633 | 718 | 802 |
| | | S SKR | – | 33 | 42 | 50 | 58 | 67 | 74 | 83 |
| | 11063 | GMRES | – | 394 | 525 | 654 | 782 | 908 | 1026 | 1142 |
| | | SKR | – | 38 | 50 | 60 | 71 | 79 | 89 | 98 |
| | 17593 | GMRES | – | 454 | 616 | 778 | 936 | 1094 | 1256 | 1418 |
| | | SKR | – | 49 | 67 | 79 | 93 | 107 | 118 | 132 |
| | 31157 | GMRES | 522 | 902 | 1206 | 1498 | – | – | – | – |
| | | SKR | 46 | 62 | 83 | 103 | – | – | – | – |
| | 70031 | GMRES | 832 | 1383 | 1793 | 2132 | – | – | – | – |
| | | SKR | 50 | 95 | 136 | 173 | – | – | – | – |

Table 13: Thermal Problem, SOR, Env1, No MPI

| | n | solver\tol | 1.E-04 | 1.E-05 | 1.E-06 | 1.E-07 | 1.E-08 | 1.E-09 | 1.E-10 | 1.E-11 |
|---|---|---|---|---|---|---|---|---|---|---|
| time | 2755 | GMRES | – | 0.06 | 0.09 | 0.10 | 0.13 | 0.14 | 0.17 | 0.19 |
| | | SKR | – | 0.07 | 0.08 | 0.09 | 0.10 | 0.12 | 0.12 | 0.13 |
| | 7821 | GMRES | – | 0.58 | 0.79 | 1.00 | 1.20 | 1.41 | 1.61 | 1.79 |
| | | SKR | – | 0.25 | 0.33 | 0.36 | 0.44 | 0.47 | 0.55 | 0.58 |
| | 11063 | GMRES | – | 0.86 | 1.16 | 1.45 | 1.77 | 2.08 | 2.41 | 2.75 |
| | | SKR | – | 0.44 | 0.51 | 0.62 | 0.75 | 0.80 | 0.92 | 1.04 |
| | 17593 | GMRES | – | 2.62 | 3.70 | 4.78 | 5.81 | 6.83 | 7.79 | 8.70 |
| | | SKR | – | 0.76 | 0.96 | 1.20 | 1.43 | 1.55 | 1.74 | 1.95 |
| | 31157 | GMRES | 3.85 | 6.96 | 9.89 | 12.66 | – | – | – | – |
| | | SKR | 1.16 | 1.82 | 2.36 | 2.99 | – | – | – | – |
| | 70031 | GMRES | 12.99 | 24.30 | 31.12 | 37.73 | – | – | – | – |
| | | SKR | 3.91 | 6.36 | 7.21 | 8.83 | – | – | – | – |
| iter | 2755 | GMRES | – | 106 | 151 | 183 | 228 | 255 | 304 | 335 |
| | | SKR | – | 24 | 29 | 36 | 40 | 45 | 50 | 55 |
| | 7821 | GMRES | – | 371 | 505 | 639 | 775 | 907 | 1035 | 1158 |
| | | SKR | – | 39 | 49 | 59 | 70 | 80 | 90 | 100 |
| | 11063 | GMRES | – | 386 | 527 | 660 | 805 | 950 | 1099 | 1256 |
| | | SKR | – | 44 | 58 | 70 | 86 | 98 | 111 | 124 |
| | 17593 | GMRES | – | 751 | 1064 | 1370 | 1670 | 1962 | 2238 | 2500 |
| | | SKR | – | 54 | 69 | 89 | 107 | 122 | 138 | 154 |
| | 31157 | GMRES | 619 | 1123 | 1594 | 2036 | – | – | – | – |
| | | SKR | 45 | 77 | 99 | 129 | – | – | – | – |
| | 70031 | GMRES | 807 | 1600 | 2182 | 2637 | – | – | – | – |
| | | SKR | 49 | 113 | 145 | 200 | – | – | – | – |

Table 14: Thermal Problem, ASM, Env1, No MPI

| | n | solver\tol | 1.E-04 | 1.E-05 | 1.E-06 | 1.E-07 | 1.E-08 | 1.E-09 | 1.E-10 | 1.E-11 |
|---|---|---|---|---|---|---|---|---|---|---|
| time | 2755 | GMRES | – | 0.08 | 0.10 | 0.12 | 0.14 | 0.16 | 0.18 | 0.21 |
| | | SKR | – | 0.05 | 0.08 | 0.09 | 0.09 | 0.10 | 0.12 | 0.13 |
| | 7821 | GMRES | – | 0.48 | 0.62 | 0.77 | 0.90 | 1.03 | 1.17 | 1.31 |
| | | SKR | – | 0.24 | 0.32 | 0.34 | 0.37 | 0.44 | 0.47 | 0.54 |
| | 11063 | GMRES | – | 0.92 | 1.21 | 1.51 | 1.81 | 2.08 | 2.37 | 2.61 |
| | | SKR | – | 0.37 | 0.48 | 0.53 | 0.64 | 0.68 | 0.79 | 0.83 |
| | 17593 | GMRES | – | 1.66 | 2.24 | 2.82 | 3.38 | 3.96 | 4.54 | 5.12 |
| | | SKR | – | 0.75 | 0.97 | 1.06 | 1.26 | 1.46 | 1.54 | 1.73 |
| | 31157 | GMRES | 3.40 | 5.85 | 7.81 | 9.68 | – | – | – | – |
| | | SKR | 1.28 | 1.53 | 2.05 | 2.54 | – | – | – | – |
| | 70031 | GMRES | 12.67 | 21.09 | 29.90 | 35.49 | – | – | – | – |
| | | SKR | 3.06 | 5.23 | 7.20 | 9.06 | – | – | – | – |
| iter | 2755 | GMRES | – | 114 | 154 | 188 | 228 | 266 | 306 | 344 |
| | | SKR | – | 21 | 26 | 31 | 35 | 40 | 45 | 49 |
| | 7821 | GMRES | – | 289 | 376 | 466 | 549 | 633 | 718 | 802 |
| | | SKR | – | 33 | 42 | 50 | 58 | 67 | 74 | 83 |
| | 11063 | GMRES | – | 394 | 525 | 654 | 782 | 908 | 1026 | 1142 |
| | | SKR | – | 38 | 50 | 60 | 71 | 79 | 89 | 98 |
| | 17593 | GMRES | – | 454 | 616 | 778 | 936 | 1094 | 1256 | 1418 |
| | | SKR | – | 49 | 67 | 79 | 93 | 107 | 118 | 132 |
| | 31157 | GMRES | 522 | 902 | 1206 | 1498 | – | – | – | – |
| | | SKR | 46 | 62 | 83 | 103 | – | – | – | – |
| | 70031 | GMRES | 832 | 1383 | 1793 | 2132 | – | – | – | – |
| | | SKR | 50 | 95 | 136 | 173 | – | – | – | – |

Table 15: Thermal Problem, ICC, Env1, No MPI

|  | n | solver\tol | 1.E-04 | 1.E-05 | 1.E-06 | 1.E-07 | 1.E-08 | 1.E-09 | 1.E-10 | 1.E-11 |
|---|---|---|---|---|---|---|---|---|---|---|
| time | 2755 | GMRES | – | 0.07 | 0.09 | 0.11 | 0.13 | 0.15 | 0.17 | 0.19 |
| | | SKR | – | 0.05 | 0.08 | 0.09 | 0.09 | 0.10 | 0.12 | 0.12 |
| | 7821 | GMRES | – | 0.46 | 0.59 | 0.73 | 0.86 | 0.99 | 1.12 | 1.25 |
| | | SKR | – | 0.24 | 0.31 | 0.33 | 0.36 | 0.43 | 0.46 | 0.53 |
| | 11063 | GMRES | – | 0.87 | 1.16 | 1.44 | 1.72 | 2.00 | 2.26 | 2.50 |
| | | SKR | – | 0.36 | 0.47 | 0.52 | 0.62 | 0.66 | 0.77 | 0.81 |
| | 17593 | GMRES | – | 1.59 | 2.15 | 2.71 | 3.26 | 3.81 | 4.37 | 4.93 |
| | | SKR | – | 0.73 | 0.95 | 1.04 | 1.23 | 1.43 | 1.51 | 1.70 |
| | 31157 | GMRES | 3.27 | 5.62 | 7.51 | 9.33 | – | – | – | – |
| | | SKR | 1.24 | 1.51 | 2.02 | 2.50 | – | – | – | – |
| | 70031 | GMRES | 14.18 | 22.57 | 28.99 | 34.54 | – | – | – | – |
| | | SKR | 3.00 | 5.10 | 7.04 | 8.92 | – | – | – | – |
| iter | 2755 | GMRES | – | 114 | 154 | 188 | 228 | 266 | 306 | 344 |
| | | SKR | – | 21 | 26 | 31 | 35 | 40 | 45 | 49 |
| | 7821 | GMRES | – | 289 | 376 | 466 | 549 | 633 | 718 | 802 |
| | | SKR | – | 33 | 42 | 50 | 58 | 67 | 74 | 83 |
| | 11063 | GMRES | – | 394 | 525 | 654 | 782 | 908 | 1026 | 1142 |
| | | SKR | – | 38 | 50 | 60 | 71 | 79 | 89 | 98 |
| | 17593 | GMRES | – | 454 | 616 | 778 | 936 | 1094 | 1256 | 1418 |
| | | SKR | – | 49 | 67 | 79 | 93 | 107 | 118 | 132 |
| | 31157 | GMRES | 522 | 902 | 1206 | 1498 | – | – | – | – |
| | | SKR | 46 | 62 | 83 | 103 | – | – | – | – |
| | 70031 | GMRES | 832 | 1383 | 1793 | 2132 | – | – | – | – |
| | | SKR | 50 | 95 | 136 | 173 | – | – | – | – |

Table 16: Thermal Problem, ILU, Env1, No MPI

|  | n | solver\tol | 1.E-04 | 1.E-05 | 1.E-06 | 1.E-07 | 1.E-08 | 1.E-09 | 1.E-10 | 1.E-11 |
|---|---|---|---|---|---|---|---|---|---|---|
| time | 2755 | GMRES | – | 0.07 | 0.09 | 0.11 | 0.13 | 0.15 | 0.17 | 0.19 |
| | | SKR | – | 0.05 | 0.08 | 0.08 | 0.09 | 0.10 | 0.12 | 0.12 |
| | 7821 | GMRES | – | 0.45 | 0.57 | 0.71 | 0.83 | 0.96 | 1.09 | 1.21 |
| | | SKR | – | 0.23 | 0.31 | 0.33 | 0.36 | 0.43 | 0.45 | 0.53 |
| | 11063 | GMRES | – | 0.85 | 1.12 | 1.40 | 1.67 | 1.94 | 2.18 | 2.43 |
| | | SKR | – | 0.36 | 0.47 | 0.51 | 0.62 | 0.66 | 0.76 | 0.81 |
| | 17593 | GMRES | – | 1.54 | 2.09 | 2.63 | 3.16 | 3.70 | 4.24 | 4.79 |
| | | SKR | – | 0.72 | 0.95 | 1.03 | 1.22 | 1.42 | 1.50 | 1.69 |
| | 31157 | GMRES | 3.17 | 5.46 | 7.28 | 9.03 | – | – | – | – |
| | | SKR | 1.23 | 1.49 | 2.00 | 2.47 | – | – | – | – |
| | 70031 | GMRES | 12.99 | 21.46 | 29.25 | 33.26 | – | – | – | – |
| | | SKR | 2.95 | 5.03 | 6.96 | 8.78 | – | – | – | – |
| iter | 2755 | GMRES | – | 114 | 154 | 188 | 228 | 266 | 306 | 344 |
| | | SKR | – | 21 | 26 | 31 | 35 | 40 | 45 | 49 |
| | 7821 | GMRES | – | 289 | 376 | 466 | 549 | 633 | 718 | 802 |
| | | SKR | – | 33 | 42 | 50 | 58 | 67 | 74 | 83 |
| | 11063 | GMRES | – | 394 | 525 | 654 | 782 | 908 | 1026 | 1142 |
| | | SKR | – | 38 | 50 | 60 | 71 | 79 | 89 | 98 |
| | 17593 | GMRES | – | 454 | 616 | 778 | 936 | 1094 | 1256 | 1418 |
| | | SKR | – | 49 | 67 | 79 | 93 | 107 | 118 | 132 |
| | 31157 | GMRES | 522 | 902 | 1206 | 1498 | – | – | – | – |
| | | SKR | 46 | 62 | 83 | 103 | – | – | – | – |
| | 70031 | GMRES | 832 | 1383 | 1793 | 2132 | – | – | – | – |
| | | SKR | 50 | 95 | 136 | 173 | – | – | – | – |

Table 17: Possion Equation, None, Env1, 7153: MPI10, 11237: MPI10, 20245: MPI20, 45337: MPI5, 71313: MPI5

|  | n | solver\tol | 1.E-05 | 1.E-06 | 1.E-07 | 1.E-08 | 1.E-09 | 1.E-10 | 1.E-11 |
|---|---|---|---|---|---|---|---|---|---|
| time | 7153 | GMRES | 0.07 | 0.09 | 0.11 | 0.13 | 0.15 | 0.17 | 0.19 |
|  |  | SKR | 0.07 | 0.11 | 0.10 | 0.11 | 0.12 | 0.13 | 0.15 |
|  | 11237 | GMRES | 0.24 | 0.33 | 0.41 | 0.50 | 0.59 | 0.66 | 0.75 |
|  |  | SKR | 0.16 | 0.19 | 0.22 | 0.26 | 0.29 | 0.34 | 0.35 |
|  | 20245 | GMRES | 0.41 | 0.57 | 0.74 | 0.90 | 1.06 | 1.24 | 1.39 |
|  |  | SKR | 0.24 | 0.29 | 0.33 | 0.39 | 0.44 | 0.50 | 0.56 |
|  | 45337 | GMRES | 1.81 | 2.74 | 3.69 | 4.64 | 5.60 | 6.56 | 7.51 |
|  |  | SKR | 1.46 | 1.91 | 2.36 | 2.79 | 3.25 | 3.69 | 4.15 |
|  | 71313 | GMRES | 3.89 | 6.13 | 8.39 | 10.74 | 13.02 | 15.33 | 17.68 |
|  |  | SKR | 3.05 | 4.10 | 5.12 | 6.16 | 7.20 | 8.25 | 9.28 |
| iter | 7153 | GMRES | 274 | 374 | 475 | 576 | 676 | 776 | 878 |
|  |  | SKR | 70 | 87 | 104 | 122 | 139 | 158 | 176 |
|  | 11237 | GMRES | 362 | 506 | 651 | 796 | 942 | 1088 | 1234 |
|  |  | SKR | 89 | 112 | 136 | 160 | 185 | 210 | 235 |
|  | 20245 | GMRES | 554 | 804 | 1054 | 1306 | 1558 | 1810 | 2062 |
|  |  | SKR | 128 | 165 | 201 | 239 | 278 | 316 | 355 |
|  | 45337 | GMRES | 958 | 1475 | 1998 | 2526 | 3054 | 3583 | 4112 |
|  |  | SKR | 209 | 279 | 349 | 419 | 489 | 560 | 631 |
|  | 71313 | GMRES | 1359 | 2158 | 2980 | 3810 | 4641 | 5472 | 6303 |
|  |  | SKR | 289 | 397 | 500 | 606 | 710 | 816 | 922 |

Table 18: Possion Equation, Jacobi, Env1, 7153: MPI10, 11237: MPI10, 20245: MPI20, 45337: MPI5, 71313: MPI5

|  | n | solver\tol | 1.E-05 | 1.E-06 | 1.E-07 | 1.E-08 | 1.E-09 | 1.E-10 | 1.E-11 |
|---|---|---|---|---|---|---|---|---|---|
| time | 7153 | GMRES | 0.07 | 0.09 | 0.12 | 0.13 | 0.16 | 0.17 | 0.19 |
|  |  | SKR | 0.08 | 0.09 | 0.10 | 0.12 | 0.12 | 0.14 | 0.15 |
|  | 11237 | GMRES | 0.25 | 0.33 | 0.42 | 0.50 | 0.59 | 0.68 | 0.78 |
|  |  | SKR | 0.18 | 0.21 | 0.24 | 0.27 | 0.31 | 0.34 | 0.37 |
|  | 20245 | GMRES | 0.41 | 0.58 | 0.75 | 0.91 | 1.06 | 1.23 | 1.40 |
|  |  | SKR | 0.25 | 0.30 | 0.35 | 0.40 | 0.46 | 0.51 | 0.56 |
|  | 45337 | GMRES | 1.85 | 2.78 | 3.73 | 4.69 | 5.65 | 6.61 | 7.69 |
|  |  | SKR | 1.48 | 1.91 | 2.37 | 2.80 | 3.25 | 3.69 | 4.13 |
|  | 71313 | GMRES | 3.94 | 6.16 | 8.41 | 10.69 | 12.92 | 15.18 | 17.41 |
|  |  | SSKR | 3.07 | 4.08 | 5.09 | 6.08 | 7.10 | 8.10 | 9.12 |
| iter | 7153 | GMRES | 279 | 379 | 482 | 581 | 681 | 782 | 883 |
|  |  | SKR | 71 | 88 | 106 | 124 | 142 | 160 | 178 |
|  | 11237 | GMRES | 368 | 512 | 658 | 802 | 947 | 1092 | 1236 |
|  |  | SKR | 91 | 114 | 138 | 163 | 188 | 212 | 237 |
|  | 20245 | GMRES | 558 | 804 | 1051 | 1298 | 1546 | 1794 | 2042 |
|  |  | SKR | 129 | 165 | 201 | 238 | 276 | 314 | 352 |
|  | 45337 | GMRES | 970 | 1484 | 2003 | 2526 | 3049 | 3571 | 4094 |
|  |  | SKR | 211 | 279 | 349 | 418 | 486 | 556 | 625 |
|  | 71313 | GMRES | 1362 | 2148 | 2954 | 3762 | 4567 | 5369 | 6167 |
|  |  | SKR | 291 | 394 | 495 | 596 | 699 | 800 | 904 |

Table 19: Possion Equation, BJacobi, Env1, 7153: MPI10, 11237: MPI10, 20245: MPI20, 45337: MPI5, 71313: MPI5

| | n | solver\tol | 1.E-05 | 1.E-06 | 1.E-07 | 1.E-08 | 1.E-09 | 1.E-10 | 1.E-11 |
|---|---|---|---|---|---|---|---|---|---|
| time | 7153 | GMRES | 0.07 | 0.08 | 0.10 | 0.11 | 0.13 | 0.14 | 0.15 |
| | | SKR | 0.07 | 0.08 | 0.09 | 0.10 | 0.11 | 0.12 | 0.13 |
| | 11237 | GMRES | 0.20 | 0.26 | 0.31 | 0.36 | 0.40 | 0.46 | 0.51 |
| | | SKR | 0.14 | 0.17 | 0.20 | 0.22 | 0.25 | 0.28 | 0.30 |
| | 20245 | GMRES | 0.34 | 0.47 | 0.58 | 0.71 | 0.84 | 0.94 | 1.06 |
| | | SKR | 0.21 | 0.26 | 0.29 | 0.33 | 0.38 | 0.42 | 0.46 |
| | 45337 | GMRES | 1.29 | 1.77 | 2.22 | 2.63 | 3.02 | 3.40 | 3.76 |
| | | SKR | 1.15 | 1.45 | 1.78 | 2.09 | 2.39 | 2.70 | 3.01 |
| | 71313 | GMRES | 2.57 | 3.53 | 4.34 | 5.09 | 5.86 | 6.62 | 7.45 |
| | | SKR | 2.26 | 2.97 | 3.63 | 4.27 | 4.94 | 5.56 | 6.24 |
| iter | 7153 | GMRES | 198 | 262 | 327 | 393 | 458 | 521 | 585 |
| | | SKR | 57 | 71 | 85 | 99 | 114 | 129 | 144 |
| | 11237 | GMRES | 262 | 349 | 432 | 513 | 591 | 668 | 743 |
| | | SKR | 70 | 88 | 106 | 125 | 144 | 163 | 182 |
| | 20245 | GMRES | 419 | 595 | 768 | 940 | 1111 | 1279 | 1446 |
| | | SKR | 102 | 132 | 160 | 191 | 220 | 251 | 280 |
| | 45337 | GMRES | 588 | 823 | 1041 | 1241 | 1434 | 1614 | 1795 |
| | | SKR | 152 | 199 | 246 | 294 | 340 | 386 | 434 |
| | 71313 | GMRES | 775 | 1081 | 1337 | 1577 | 1818 | 2061 | 2320 |
| | | SKR | 200 | 269 | 334 | 396 | 462 | 523 | 588 |

Table 20: Possion Equation, SOR, Env1, 7153: MPI10, 11237: MPI10, 20245: MPI20, 45337: MPI5, 71313: MPI5

| | n | solver\tol | 1.E-05 | 1.E-06 | 1.E-07 | 1.E-08 | 1.E-09 | 1.E-10 | 1.E-11 |
|---|---|---|---|---|---|---|---|---|---|
| time | 7153 | GMRES | 0.07 | 0.08 | 0.09 | 0.11 | 0.12 | 0.14 | 0.15 |
| | | S SKR | 0.07 | 0.08 | 0.09 | 0.10 | 0.11 | 0.12 | 0.13 |
| | 11237 | GMRES | 0.20 | 0.26 | 0.32 | 0.37 | 0.42 | 0.47 | 0.51 |
| | | SKR | 0.15 | 0.18 | 0.21 | 0.23 | 0.26 | 0.28 | 0.31 |
| | 20245 | GMRES | 0.33 | 0.46 | 0.58 | 0.70 | 0.82 | 0.94 | 1.08 |
| | | SKR | 0.22 | 0.26 | 0.32 | 0.35 | 0.39 | 0.43 | 0.47 |
| | 45337 | GMRES | 1.43 | 2.04 | 2.62 | 3.17 | 3.68 | 4.15 | 4.60 |
| | | SKR | 1.16 | 1.47 | 1.80 | 2.11 | 2.42 | 2.73 | 3.06 |
| | 71313 | GMRES | 2.76 | 3.94 | 4.98 | 5.90 | 6.78 | 7.64 | 8.49 |
| | | SKR | 2.31 | 3.03 | 3.69 | 4.37 | 5.05 | 5.72 | 6.37 |
| iter | 7153 | GMRES | 203 | 271 | 339 | 408 | 477 | 545 | 613 |
| | | SKR | 57 | 72 | 86 | 99 | 115 | 130 | 144 |
| | 11237 | GMRES | 270 | 362 | 450 | 534 | 614 | 692 | 766 |
| | | SKR | 71 | 88 | 107 | 126 | 145 | 164 | 183 |
| | 20245 | GMRES | 421 | 599 | 775 | 953 | 1129 | 1305 | 1480 |
| | | SKR | 102 | 132 | 160 | 191 | 220 | 251 | 280 |
| | 45337 | GMRES | 647 | 942 | 1221 | 1484 | 1729 | 1960 | 2174 |
| | | SKR | 153 | 202 | 250 | 298 | 344 | 390 | 439 |
| | 71313 | GMRES | 822 | 1192 | 1516 | 1804 | 2076 | 2342 | 2610 |
| | | SKR | 203 | 272 | 339 | 405 | 470 | 535 | 599 |

Table 21: Possion Equation, ASM, Env1, 7153: MPI10, 11237: MPI10, 20245: MPI20, 45337: MPI5, 71313: MPI5

|  | n | solver\tol | 1.E-05 | 1.E-06 | 1.E-07 | 1.E-08 | 1.E-09 | 1.E-10 | 1.E-11 |
|---|---|---|---|---|---|---|---|---|---|
| time | 7153 | GMRES | 0.07 | 0.09 | 0.10 | 0.12 | 0.14 | 0.15 | 0.17 |
|  |  | SKR | 0.07 | 0.08 | 0.10 | 0.10 | 0.11 | 0.13 | 0.14 |
|  | 11237 | GMRES | 0.27 | 0.35 | 0.42 | 0.49 | 0.56 | 0.63 | 0.69 |
|  |  | SKR | 0.17 | 0.21 | 0.23 | 0.27 | 0.30 | 0.33 | 0.36 |
|  | 20245 | GMRES | 0.48 | 0.66 | 0.83 | 1.01 | 1.18 | 1.35 | 1.53 |
|  |  | SKR | 0.25 | 0.31 | 0.36 | 0.41 | 0.46 | 0.52 | 0.57 |
|  | 45337 | GMRES | 1.42 | 1.93 | 2.42 | 2.87 | 3.32 | 3.71 | 4.10 |
|  |  | SKR | 1.20 | 1.51 | 1.85 | 2.16 | 2.47 | 2.79 | 3.12 |
|  | 71313 | GMRES | 2.80 | 3.86 | 4.71 | 5.52 | 6.35 | 7.18 | 8.08 |
|  |  | SKR | 2.35 | 3.09 | 3.76 | 4.42 | 5.11 | 5.75 | 6.44 |
| iter | 7153 | GMRES | 198 | 262 | 327 | 393 | 458 | 521 | 585 |
|  |  | SKR | 57 | 71 | 85 | 99 | 114 | 129 | 144 |
|  | 11237 | GMRES | 262 | 349 | 432 | 513 | 591 | 668 | 743 |
|  |  | SKR | 70 | 88 | 106 | 125 | 144 | 163 | 182 |
|  | 20245 | GMRES | 419 | 595 | 768 | 940 | 1111 | 1279 | 1446 |
|  |  | SKR | 102 | 132 | 160 | 191 | 220 | 251 | 280 |
|  | 45337 | GMRES | 588 | 823 | 1041 | 1241 | 1434 | 1614 | 1795 |
|  |  | SKR | 152 | 199 | 246 | 294 | 340 | 386 | 434 |
|  | 71313 | GMRES | 775 | 1081 | 1337 | 1577 | 1818 | 2061 | 2320 |
|  |  | SKR | 200 | 269 | 334 | 396 | 462 | 523 | 588 |

Table 22: Possion Equation, ICC, Env1, No MPI

|  | n | solver\tol | 1.E-05 | 1.E-06 | 1.E-07 | 1.E-08 | 1.E-09 | 1.E-10 | 1.E-11 |
|---|---|---|---|---|---|---|---|---|---|
| time | 7153 | GMRES | 0.09 | 0.10 | 0.10 | 0.11 | 0.12 | 0.13 | 0.13 |
|  |  | SKR | 0.12 | 0.13 | 0.14 | 0.13 | 0.13 | 0.14 | 0.14 |
|  | 11237 | GMRES | 0.15 | 0.17 | 0.19 | 0.20 | 0.21 | 0.23 | 0.25 |
|  |  | SKR | 0.21 | 0.20 | 0.21 | 0.21 | 0.22 | 0.38 | 0.39 |
|  | 20245 | GMRES | 0.32 | 0.37 | 0.40 | 0.45 | 0.50 | 0.55 | 0.59 |
|  |  | SKR | 0.37 | 0.38 | 0.39 | 0.69 | 0.72 | 0.74 | 0.77 |
|  | 45337 | GGMRES | 0.91 | 1.12 | 1.26 | 1.41 | 1.56 | 1.71 | 1.90 |
|  |  | SKR | 0.88 | 1.58 | 1.66 | 1.75 | 1.83 | 1.93 | 2.27 |
|  | 71313 | GMRES | 2.01 | 2.19 | 2.45 | 2.91 | 3.35 | 3.66 | 3.88 |
|  |  | SKR | 2.50 | 2.67 | 2.86 | 3.11 | 3.62 | 3.81 | 3.97 |
| iter | 7153 | GMRES | 22 | 26 | 30 | 33 | 36 | 39 | 42 |
|  |  | SKR | 15 | 17 | 19 | 16 | 18 | 19 | 20 |
|  | 11237 | GMRES | 26 | 32 | 36 | 40 | 45 | 52 | 59 |
|  |  | SKR | 20 | 17 | 19 | 18 | 20 | 22 | 24 |
|  | 20245 | GMRES | 35 | 42 | 50 | 59 | 67 | 74 | 80 |
|  |  | SKR | 18 | 20 | 19 | 22 | 25 | 28 | 31 |
|  | 45337 | GMRES | 49 | 62 | 75 | 84 | 97 | 110 | 120 |
|  |  | SKR | 19 | 23 | 27 | 31 | 35 | 39 | 43 |
|  | 71313 | GMRES | 73 | 84 | 98 | 118 | 139 | 154 | 167 |
|  |  | SKR | 24 | 29 | 34 | 39 | 44 | 49 | 54 |

Table 23: Possion Equation, ILU, Env1, No MPI

|      | n     | solver\tol | 1.E-05 | 1.E-06 | 1.E-07 | 1.E-08 | 1.E-09 | 1.E-10 | 1.E-11 |
|------|-------|------------|--------|--------|--------|--------|--------|--------|--------|
| time | 7153  | GMRES      | 0.26   | 0.26   | 0.27   | 0.28   | 0.28   | 0.29   | 0.29   |
|      |       | SKR        | 0.29   | 0.30   | 0.32   | 0.31   | 0.32   | 0.32   | 0.33   |
|      | 11237 | GMRES      | 0.41   | 0.43   | 0.44   | 0.45   | 0.46   | 0.49   | 0.49   |
|      |       | SKR        | 0.48   | 0.50   | 0.51   | 0.51   | 0.53   | 0.52   | 0.53   |
|      | 20245 | GMRES      | 0.84   | 0.85   | 0.87   | 0.91   | 0.93   | 0.97   | 0.99   |
|      |       | SKR        | 0.94   | 0.96   | 0.94   | 0.95   | 0.97   | 0.98   | 1.00   |
|      | 45337 | GMRES      | 2.02   | 2.18   | 2.26   | 2.36   | 2.54   | 2.70   | 2.87   |
|      |       | SKR        | 2.15   | 2.24   | 2.22   | 2.29   | 2.98   | 3.02   | 3.13   |
|      | 71313 | MRES       | 3.46   | 3.67   | 3.96   | 4.27   | 4.52   | 4.79   | 4.93   |
|      |       | SKR        | 3.54   | 3.63   | 4.68   | 4.79   | 4.94   | 5.11   | 5.23   |
| iter | 7153  | GMRES      | 15     | 18     | 20     | 22     | 24     | 26     | 27     |
|      |       | SKR        | 10     | 11     | 12     | 13     | 14     | 15     | 16     |
|      | 11237 | GMRES      | 18     | 22     | 24     | 27     | 29     | 31     | 33     |
|      |       | SKR        | 13     | 14     | 14     | 16     | 17     | 18     | 19     |
|      | 20245 | GMRES      | 23     | 27     | 31     | 34     | 38     | 40     | 44     |
|      |       | SKR        | 16     | 18     | 15     | 17     | 18     | 20     | 20     |
|      | 45337 | GMRES      | 32     | 39     | 46     | 53     | 62     | 69     | 76     |
|      |       | SKR        | 17     | 20     | 20     | 21     | 24     | 26     | 29     |
|      | 71313 | GMRES      | 40     | 49     | 60     | 71     | 78     | 86     | 95     |
|      |       | SKR        | 19     | 20     | 23     | 26     | 29     | 33     | 36     |

Table 24: Helmholtz Equation, None, Env1, MPI72

|      | n     | solver\tol | 1.E-01 | 1.E-02 | 1.E-03 | 1.E-04 | 1.E-05 | 1.E-06 | 1.E-07 |
|------|-------|------------|--------|--------|--------|--------|--------|--------|--------|
| time | 2500  | GMRES      | 0.81   | 1.37   | 1.75   | 2.07   | 2.36   | 2.61   | 2.87   |
|      |       | SKR        | 0.17   | 0.21   | 0.22   | 0.23   | 0.25   | 0.27   | 0.29   |
|      | 6400  | GMRES      | 1.74   | 3.09   | 3.72   | 4.14   | 4.49   | 4.77   | 4.93   |
|      |       | SKR        | 0.29   | 0.36   | 0.41   | 0.45   | 0.48   | 0.51   | 0.54   |
|      | 10000 | GMRES      | 2.44   | 4.06   | 4.88   | 5.22   | 5.41   | 5.51   | 5.57   |
|      |       | SKR        | 0.40   | 0.52   | 0.61   | 0.67   | 0.71   | 0.80   | 0.86   |
|      | 22500 | GMRES      | 16.22  | 33.05  | 39.39  | 43.37  | 46.67  | 49.21  | –      |
|      |       | SKR        | 3.60   | 6.73   | 9.31   | 6.13   | 9.04   | 8.54   | –      |
| iter | 2500  | GMRES      | 1,305  | 2,279  | 2,936  | 3,520  | 3,999  | 4,461  | 4882   |
|      |       | SKR        | 98     | 130    | 151    | 168    | 185    | 199    | 214    |
|      | 6400  | GMRES      | 3,036  | 5,436  | 6,604  | 7,357  | 7,978  | 8,421  | 8749   |
|      |       | SKR        | 212    | 277    | 319    | 354    | 388    | 414    | 441    |
|      | 10000 | GMRES      | 4,187  | 7,052  | 8,478  | 9,118  | 9,407  | 9,574  | 9713   |
|      |       | SKR        | 303    | 407    | 473    | 538    | 570    | 648    | 710    |
|      | 22500 | GMRES      | 26,565 | 53,855 | 64,542 | 71,353 | 76,793 | 80,991 | –      |
|      |       | SKR        | 2,647  | 5,014  | 6,937  | 4,557  | 6,743  | 6,394  | –      |

Table 25: Helmholtz Equation, Jacobi, Env1, MPI72

|  | n | solver\tol | 1.E-01 | 1.E-02 | 1.E-03 | 1.E-04 | 1.E-05 | 1.E-06 | 1.E-07 |
|---|---|---|---|---|---|---|---|---|---|
| time | 2500 | GMRES | 0.72 | 1.20 | 1.45 | 1.65 | 1.87 | 2.07 | 2.28 |
|  |  | SKR | 0.16 | 0.19 | 0.21 | 0.23 | 0.24 | 0.25 | 0.27 |
|  | 6400 | GMRES | 1.45 | 2.64 | 3.30 | 3.71 | 4.08 | 4.32 | 4.54 |
|  |  | SKR | 0.27 | 0.33 | 0.37 | 0.39 | 0.42 | 0.45 | 0.47 |
|  | 10000 | GMRES | 2.18 | 3.82 | 4.57 | 5.00 | 5.15 | 5.33 | 5.40 |
|  |  | SKR | 0.37 | 0.45 | 0.51 | 0.56 | 0.60 | 0.64 | 0.68 |
|  | 22500 | GMRES | 16.31 | 29.04 | 37.38 | 41.38 | 43.97 | 45.58 | – |
|  |  | SKR | 4.92 | 4.24 | 6.01 | 6.46 | 5.49 | 5.66 | – |
| iter | 2500 | GMRES | 1,155 | 1,988 | 2,405 | 2,762 | 3,136 | 3,509 | 3849 |
|  |  | SKR | 82 | 110 | 126 | 140 | 153 | 164 | 176 |
|  | 6400 | GMRES | 2,507 | 4,597 | 5,791 | 6,562 | 7,220 | 7,637 | 8028 |
|  |  | SKR | 176 | 228 | 261 | 289 | 315 | 335 | 356 |
|  | 10000 | GMRES | 3,726 | 6,621 | 7,959 | 8,630 | 9,004 | 9,237 | 9430 |
|  |  | SKR | 249 | 327 | 373 | 415 | 451 | 489 | 521 |
|  | 22500 | GMRES | 26,707 | 47,340 | 60,878 | 67,205 | 71,418 | 74,351 | – |
|  |  | SKR | 3,611 | 3,113 | 4,442 | 4,786 | 4,054 | 4,175 | – |

Table 26: Helmholtz Equation, BJacobi, Env1, MPI72

|  | n | solver\tol | 1.E-01 | 1.E-02 | 1.E-03 | 1.E-04 | 1.E-05 | 1.E-06 | 1.E-07 |
|---|---|---|---|---|---|---|---|---|---|
| time | 2500 | GMRES | 0.50 | 0.75 | 0.96 | 1.12 | 1.23 | 1.35 | 1.47 |
|  |  | SKR | 0.12 | 0.14 | 0.15 | 0.16 | 0.18 | 0.18 | 0.19 |
|  | 6400 | GMRES | 1.09 | 2.01 | 2.48 | 2.89 | 3.18 | 3.48 | 3.71 |
|  |  | SKR | 0.19 | 0.23 | 0.26 | 0.28 | 0.29 | 0.31 | 0.33 |
|  | 10000 | GMRES | 1.49 | 2.60 | 3.36 | 3.86 | 4.27 | 4.55 | 4.72 |
|  |  | SKR | 0.24 | 0.29 | 0.32 | 0.35 | 0.37 | 0.39 | 0.42 |
|  | 22500 | GMRES | 13.13 | 19.82 | 25.01 | 28.58 | 30.87 | 32.45 | – |
|  |  | SKR | 2.32 | 3.51 | 0.71 | 0.77 | 0.85 | 3.41 | – |
| iter | 2500 | GMRES | 737 | 1,154 | 1,513 | 1,772 | 1,979 | 2,168 | 2352 |
|  |  | SKR | 58 | 76 | 87 | 96 | 105 | 113 | 122 |
|  | 6400 | GMRES | 1,766 | 3,332 | 4,181 | 4,853 | 5,359 | 5,821 | 6272 |
|  |  | SKR | 116 | 154 | 175 | 193 | 208 | 223 | 238 |
|  | 10000 | GMRES | 2,405 | 4,227 | 5,541 | 6,376 | 7,019 | 7,511 | 7883 |
|  |  | SKR | 147 | 197 | 222 | 244 | 265 | 283 | 301 |
|  | 22500 | GMRES | 20,335 | 30,564 | 38,699 | 44,223 | 47,387 | 50,197 | – |
|  |  | SKR | 1,639 | 2,505 | – | – | – | 2,427 | – |

Table 27: Helmholtz Equation, SOR, Env1, MPI72

| | n | solver\tol | 1.E-01 | 1.E-02 | 1.E-03 | 1.E-04 | 1.E-05 | 1.E-06 | 1.E-07 |
|---|---|---|---|---|---|---|---|---|---|
| time | 2500 | GMRES | 0.48 | 0.77 | 0.93 | 1.10 | 1.19 | 1.30 | 1.38 |
| | | SKR | 0.14 | 0.16 | 0.17 | 0.18 | 0.19 | 0.20 | 0.21 |
| | 6400 | GMRES | 1.17 | 1.98 | 2.40 | 2.77 | 3.04 | 3.30 | 3.59 |
| | | SKR | 0.21 | 0.25 | 0.27 | 0.29 | 0.31 | 0.33 | 0.34 |
| | 10000 | GMRES | 1.46 | 2.60 | 3.22 | 3.77 | 4.16 | 4.43 | 4.66 |
| | | SKR | 0.25 | 0.31 | 0.34 | 0.37 | 0.40 | 0.42 | 0.44 |
| | 22500 | GMRES | 12.28 | 18.00 | 23.99 | 27.31 | 30.17 | 32.18 | – |
| | | SKR | 2.18 | 0.65 | 0.75 | 1.26 | 3.76 | 2.93 | – |
| iter | 2500 | GMRES | 719 | 1,231 | 1,507 | 1,782 | 1,966 | 2,134 | 2290 |
| | | SKR | 57 | 76 | 88 | 96 | 105 | 114 | 122 |
| | 6400 | GMRES | 1,987 | 3,414 | 4,215 | 4,823 | 5,339 | 5,783 | 6262 |
| | | SKR | 118 | 156 | 176 | 194 | 210 | 225 | 239 |
| | 10000 | GMRES | 2,441 | 4,405 | 5,503 | 6,432 | 7,074 | 7,595 | 7976 |
| | | SKR | 149 | 197 | 226 | 248 | 270 | 288 | 308 |
| | 22500 | GMRES | 19,540 | 28,939 | 38,419 | 43,545 | 48,285 | 51,382 | – |
| | | SKR | 1,545 | – | – | – | 2,712 | 2,094 | – |

Table 28: Helmholtz Equation, ASM, Env1, MPI72

| | n | solver\tol | 1.E-01 | 1.E-02 | 1.E-03 | 1.E-04 | 1.E-05 | 1.E-06 | 1.E-07 |
|---|---|---|---|---|---|---|---|---|---|
| time | 2500 | GMRES | 0.12 | 0.34 | 0.42 | 0.49 | 0.55 | 0.63 | 0.69 |
| | | SKR | 0.13 | 0.14 | 0.15 | 0.16 | 0.17 | 0.18 | 0.18 |
| | 6400 | GMRES | 1.01 | 1.61 | 1.95 | 2.23 | 2.59 | 2.84 | 3.06 |
| | | SKR | 0.19 | 0.22 | 0.23 | 0.25 | 0.26 | 0.27 | 0.28 |
| | 10000 | GMRES | 1.55 | 2.83 | 3.44 | 3.98 | 4.35 | 4.72 | 5.07 |
| | | SKR | 0.22 | 0.27 | 0.29 | 0.31 | 0.33 | 0.34 | 0.36 |
| | 22500 | GMRES | 15.36 | 25.21 | 32.21 | 35.62 | 38.32 | 41.75 | – |
| | | SKR | 0.95 | 0.56 | 2.33 | 0.75 | 0.77 | 3.08 | – |
| iter | 2500 | GMRES | 64 | 306 | 384 | 459 | 531 | 601 | 668 |
| | | SKR | 33 | 42 | 48 | 53 | 58 | 63 | 67 |
| | 6400 | GMRES | 1,054 | 1,713 | 2,118 | 2,412 | 2,803 | 3,081 | 3328 |
| | | SKR | 68 | 89 | 100 | 110 | 118 | 126 | 134 |
| | 10000 | GMRES | 1,611 | 3,034 | 3,659 | 4,264 | 4,676 | 5,068 | 5448 |
| | | SKR | 91 | 119 | 136 | 149 | 160 | 170 | 182 |
| | 22500 | GMRES | 15,683 | 25,566 | 32,785 | 36,475 | 38,992 | 42,431 | – |
| | | SKR | 498 | – | 1,310 | – | – | 1,738 | – |

Table 29: Helmholtz Equation, ICC, Env2, NoMPI

| | n | solver\tol | 1.E-01 | 1.E-02 | 1.E-03 | 1.E-04 | 1.E-05 | 1.E-06 | 1.E-07 |
|---|---|---|---|---|---|---|---|---|---|
| time | 2500 | GMRES | 0.01 | 0.02 | 0.03 | 0.04 | 0.05 | 0.06 | 0.07 |
| | | SKR | 0.03 | 0.04 | 0.04 | 0.05 | 0.05 | 0.05 | 0.06 |
| | 6400 | GMRES | 0.22 | 0.71 | 0.93 | 1.15 | 1.29 | 1.43 | 1.55 |
| | | SKR | 0.13 | 0.17 | 0.18 | 0.20 | 0.21 | 0.23 | 0.24 |
| | 10000 | GMRES | 0.65 | 1.58 | 2.12 | 2.50 | 2.80 | 3.10 | 3.32 |
| | | SKR | 0.25 | 0.33 | 0.37 | 0.41 | 0.44 | 0.47 | 0.49 |
| | 22500 | GMRES | 2.03 | 4.77 | 6.08 | 6.93 | 7.66 | 8.34 | – |
| | | SKR | 0.62 | 0.90 | 1.02 | 1.12 | 1.21 | 1.30 | – |
| iter | 2500 | GMRES | 48 | 97 | 160 | 232 | 294 | 362 | 424 |
| | | SKR | 28 | 38 | 43 | 48 | 52 | 55 | 59 |
| | 6400 | GMRES | 492 | 1,578 | 2,048 | 2,499 | 2,866 | 3,160 | 3431 |
| | | SKR | 57 | 78 | 88 | 96 | 103 | 109 | 116 |
| | 10000 | GMRES | 915 | 2,229 | 2,983 | 3,518 | 3,942 | 4,356 | 4665 |
| | | SKR | 76 | 104 | 117 | 129 | 139 | 149 | 159 |
| | 22500 | GMRES | 1,149 | 2,704 | 3,442 | 3,926 | 4,338 | 4,733 | – |
| | | SKR | 81 | 122 | 140 | 155 | 168 | 181 | – |

Table 30: Helmholtz Equation, ILU, Env2, NoMPI

| | n | solver\tol | 1.E-01 | 1.E-02 | 1.E-03 | 1.E-04 | 1.E-05 | 1.E-06 | 1.E-07 |
|---|---|---|---|---|---|---|---|---|---|
| time | 2500 | GMRES | 0.00 | 0.01 | 0.01 | 0.01 | 0.01 | 0.01 | 0.01 |
| | | SKR | 0.01 | 0.02 | 0.03 | 0.03 | 0.03 | 0.03 | 0.04 |
| | 6400 | GMRES | 0.20 | 0.64 | 0.85 | 1.02 | 1.15 | 1.27 | 1.37 |
| | | SKR | 0.13 | 0.16 | 0.18 | 0.19 | 0.21 | 0.22 | 0.23 |
| | 10000 | GMRES | 0.62 | 1.29 | 1.75 | 2.30 | 2.63 | 2.88 | 3.11 |
| | | SKR | 0.25 | 0.33 | 0.37 | 0.40 | 0.43 | 0.46 | 0.49 |
| | 22500 | GMRES | 2.94 | 5.60 | 6.74 | 7.73 | 8.62 | 9.28 | – |
| | | SKR | 0.74 | 1.04 | 1.19 | 1.31 | 1.43 | 1.52 | – |
| iter | 2500 | GMRES | 20 | 28 | 32 | 36 | 40 | 46 | 56 |
| | | SKR | 17 | 21 | 25 | 28 | 30 | 32 | 34 |
| | 6400 | GMRES | 437 | 1,420 | 1,875 | 2,257 | 2,548 | 2,822 | 3056 |
| | | SKR | 57 | 77 | 86 | 93 | 100 | 107 | 114 |
| | 10000 | GMRES | 869 | 1,818 | 2,483 | 3,254 | 3,726 | 4,085 | 4401 |
| | | SKR | 76 | 104 | 115 | 127 | 137 | 147 | 157 |
| | 22500 | GMRES | 1,768 | 3,373 | 4,058 | 4,649 | 5,181 | 5,587 | – |
| | | SKR | 101 | 147 | 169 | 187 | 203 | 219 | – |

# E  ADDITIONAL EXPERIMENTS

## E.1  OTHER APPLICATION SCENARIOS

Neural operators is not a prerequisite for the application of our SKR algorithm. Indeed, our initial use of neural operators revealed the time-consuming issue of dataset generation. After an extensive study of Krylov methods, we developed the SKR algorithm, which has proven to be highly successful. Neural operators are one of the most renowned and efficient data-driven methods for solving PDE problems, and the time-intensive nature of dataset generation has been a significant hurdle in the field, which is why our paper is set in this context. However, the SKR algorithm's utility is not limited to neural operator problems and can be applied to other areas, such as:

1. Data-Driven PDE Algorithms: There are various data-driven PDE solving or parameter optimization algorithms in both traditional heuristics and AI for PDE domains. These algorithms often require the frequent generation of large volumes of PDE data for datasets, a process that our SKR algorithm can accelerate. For instance, Greenfeld et al. (2019) Hsieh et al. (2019).

2. Specific Physical Problem Solving: In certain scenarios, modeling the physical problems at hand translates into solving multiple related linear systems. Our SKR algorithm is adept at efficiently handling such issues. For instance, this is evident in the modeling of fatigue and fracture through finite element analysis (Gullerud & Dodds Jr, 2001), which employs dynamic loading across numerous steps, resulting in a substantial number of interrelated linear systems. Similarly, in the resolution process of lattice quantum field theory (Muroya et al., 2003), the end goal involves solving a large collection of correlated linear systems.

## E.2  PARALLELIZATION STRATEGIES

The parallelism of our SKR algorithm is indeed a critical component. Typically, matrix algorithms employ three parallelization strategies.

### E.2.1  MPI-BASED PARALLELIZATION

Using MPI to parallelize certain computational processes within the matrix algorithm, such as the Krylov subspace iteration in GMRES, which often relies on specific matrix preconditioning methods. Regarding this MPI-based parallel approach, our experiments have conducted numerous comparative tests in Appendix D.6, and our SKR algorithm has consistently achieved excellent results.

### E.2.2  DECOMPOSE COMPUTATIONAL TASKS

Utilizing MPI to decompose the computational task into several independent parts, each solved separately. As you rightly mentioned in your question, this is indeed an excellent approach. We will illustrate the parallelism strategy of our SKR algorithm with a specific example, showcasing its sustained and notable acceleration impact.

- Imagine we have a server CPU with 72 parallel threads. Our parallel SKR algorithm comprises the following parts:

1. Sorting: As with the original SKR, we order these data points using a suitable sorting algorithm into a sequence where consecutive parameters are strongly correlated, meaning the distance between parameter matrices is minimal (we usually use the 1, 2, or infinity norms of matrices as the metric in this Banach space). The choice of sorting algorithm varies with the size of the dataset:

    - Greedy Sorting: Simple to implement and negligible in computational cost for smaller datasets, such as our 7200 data points. However, for larger datasets, say $10^7$ data points, the computational load of the greedy algorithm increases substantially and cannot be parallelized across multiple CPUs.

    - FFT Dimension Reduction + Fractal Division + Greedy Sorting: For very large datasets, we propose a parallelizable method that first reduces dimensionality via FFT to manage the high-dimensional coordinates, then applies a fractal division algorithm based on the Hilbert curve, and finally, performs greedy sorting within each divided section.

2. Solving Linear Systems: Assuming the task at hand is to generate a dataset comprising 7200 data points, upon sequencing the data to establish a strong correlation, we subsequently partition these 7200 data points into 72 batches, each containing 100 points, distributing the tasks of solving the corresponding linear systems to different CPU threads. Each thread employs our SKR algorithm for solving, achieving parallelism in the process.

- To demonstrate the feasibility of our parallel approach and its effective acceleration, we have designed the following experiments.

    – An experimental simulation was conducted to generate a dataset for the Helmholtz equation, comprising 7200 data points. Each corresponding matrix has a dimension of 10000, with Successive Over-Relaxation (SOR) used uniformly for matrix preconditioning. Both our SKR algorithm and the baseline GMRES algorithm were implemented to distribute the solving of 7200 tasks across 72 threads, with each thread solving 100 linear systems. Due to the potential variance in computation completion times across threads, the reported computation times and iteration counts have been averaged for consistency.

    – The upper half of the table presents the average computation time taken by both algorithms to solve each linear system, measured in seconds. The lower half details the average number of iterations required to solve each linear system. The first row of the table specifies the precision requirements for solving linear systems.

Table 31: Comparative Experimets of Parallel Versions

|  |  | 1E-03 | 1E-05 | 1E-07 |
|---|---|---|---|---|
| time(s) | Parallel GMRES | 0.08 | 0.105 | 0.122 |
| time(s) | Parallel SKR(ours) | 0.011 | 0.013 | 0.015 |
| iter | Parallel GMRES | 3734 | 4906 | 5715 |
| iter | Parallel SKR(ours) | 126 | 148 | 167 |

    – The experimental results reveal that our SKR algorithm achieves a 6.7-8.0 fold acceleration in computation time and a 30-34 fold reduction in the number of iterations compared to the baseline.

### E.2.3 BLOCK PARALLE

The block concept is a strategy for parallelizing large matrices and reducing memory overhead. Drawing from the idea of Krylov blocks, we have redesigned a block version of our SKR algorithm to facilitate parallel processing. The fundamental principle involves transforming the conventional matrix algorithm into a block matrix variant, where each block is computed on a distinct CPU, thereby achieving parallel execution of the matrix algorithm. This parallel approach significantly reduces memory usage, making the algorithm suitable for extremely large matrix computations. We will now demonstrate the remarkable acceleration achieved by our block version of the SKR algorithm through experimental results:

- An experimental simulation was conducted to generate a dataset for the Helmholtz equation. Each corresponding matrix has a dimension of 10000, with Successive Over-Relaxation (SOR) used uniformly for matrix preconditioning, running on 72 parallel MPI threads.

- The upper half of the table presents the average computation time taken by both algorithms to solve each linear system, measured in seconds. The lower half details the average number of iterations required to solve each linear system. The first row of the table specifies the precision requirements for solving linear systems.

- The experimental results reveal that our SKR algorithm achieves a 150-200 fold acceleration in computation time and a 24-26 fold reduction in the number of iterations compared to the baseline. The significant improvement in computation time is mainly attributed to the effects of MPI parallelization.

Table 32: Comparative Experimets of Block Versions

|       |                 | 1E-03 | 1E-05 | 1E-07 |
|-------|-----------------|-------|-------|-------|
| time(s) | GMRES         | 3.22  | 4.16  | 4.66  |
| time(s) | Block SKR(ours) | 0.015 | 0.025 | 0.03  |
| iter  | GMRES           | 5503  | 7074  | 7976  |
| iter  | Block SKR(ours) | 224   | 269   | 305   |

### E.3 DATASET VALIDITY

we will explain from both theoretical and experimental perspectives that the data set generated using our SKR algorithm will not affect the training of neural operators.

#### E.3.1 THEORETICAL PERSPECTIVE

From the theoretical perspective, under specific error tolerances, our SKR algorithm does not alter the outcome of the generated dataset. As detailed in Section 5.1, our SKR algorithm is backed by rigorous convergence analysis, ensuring accurate convergence to the true solution. Especially for linear systems with high correlation, our SKR algorithm demonstrates a superior theoretical convergence rate over GMRES.

In the field of numerical solutions for large matrices, there are generally no true solutions without error, only numerical solutions with errors smaller than an acceptable threshold. We consider the numerical solution satisfactory when the error of the computed linear system's solution is below the acceptable threshold, which we believe accurately represents the true solution.

#### E.3.2 EXPERIMENTAL PERSPECTIVE

Experimentally, we found that our SKR algorithm does not affect the effectiveness of the generated dataset when training neural operators.

For the Darcy flow dataset, we simultaneously generated 5256 data points with a precision of $10^{-8}$ using both our SKR algorithm and GMRES, of which 5000 were designated for the training set and 256 for the test set, with the matrix size being 2500. The data in the table represent the relative error under the two-norm during the training process, where the first row indicates the number of training epochs and the rightmost column shows the final convergence error.

Table 33: Comparative Efficacy of Neural Operators Trained on Datasets Generated by Different Algorithms

|            | 0     | 100   | 200   | 300   | 400   | final |
|------------|-------|-------|-------|-------|-------|-------|
| GMRES 1    | 0.912 | 0.127 | 0.096 | 0.055 | 0.025 | 0.02  |
| GMRES 2    | 0.927 | 0.144 | 0.094 | 0.054 | 0.026 | 0.02  |
| SKR(ours) 1 | 0.886 | 0.132 | 0.085 | 0.055 | 0.026 | 0.02  |
| SKR(ours) 2 | 0.912 | 0.145 | 0.088 | 0.051 | 0.028 | 0.02  |

To mitigate the impact of randomness during the training process, we trained the neural network models twice using datasets generated by GMRES and SKR respectively to compare their training dynamics. The results of this experiment demonstrate that the datasets generated by our SKR algorithm and GMRES yield identical outcomes when used to train neural operators.

Based on the aforementioned theoretical analysis and experimental validation, we can confidently state that the datasets generated by our SKR algorithm and the baseline GMRES algorithm have no impact on the training of neural operators. That is, the performance of the datasets they generate is equivalent.

