# OpenReview forum: "Accelerating Data Generation for Neural Operators via Krylov Subspace Recycling"
_ICLR.cc/2024/Conference — ICLR 2024 spotlight_

### Official Review · Reviewer_A7dr · 2023-10-30

**Soundness:** 3 good
**Presentation:** 2 fair
**Contribution:** 3 good
**Rating:** 8
**Confidence:** 3

**Summary:**

The paper addresses lengthy workload required for generating data for neural operators, which are one class of typical machine learning models used as surrogate models for solving partial different equations. The key idea of the paper is the use of Krylov subspace recycling method, a technique for solving a series of interrelated systems. Together with the sorting algorithm based on inherent similarities, the proposed method greatly accelerates the data generation process.

**Strengths:**

The main advantage of the paper is an oriented way of generating dataset based on Krylov subspace recycling method, whose validity is supported by rigorous theoretical analysis. The paper reveals common features of (generating process of) NO datasets. The proposed method is evaluated with a wide range of its variants and the reduction in the computational cost is shown significant for most of the cases.

**Weaknesses:**

Although the reduction in the computational workload of SKR is significant, it is still not clear to me that SKR is a better choice as a generator for NO dataset, since the paper lacks evaluation of the dataset generated by SKR in the context of training NOs. One of my questions in this regard is:
How does the performance of NO instances (e.g. FNO, DeepONet) trained on datasets generated by SKR compare to those trained on their original datasets?
Speaking from the practical aspect, it might be also beneficial for readers if the authors could give run-time comparison of SKR to parallelized solvers.

The presentation of the paper does not look clear to me. The followings are a couple of examples that made it difficult for me to understand the paper
* In Section 3.1 ”The data points for these PDEs in the parameter space of the NO inputs are relatively dense” : How did the authors withdraw this conclusion? Compared to what PDEs are the inputs dense?
* Table 5 (and its caption) is entirely confusing and hard to understand.
* Caption in Figure 9 “Specifically, when the parameters of the neural operator are closely matched in Helmholtz equations” : how do you compare the parameters of NO to Helmonltz equations?

**Questions:**

See the weeknesses.

**Details Of Ethics Concerns:**

No ethical concerns.

---

> ### Author Response · Authors · 2023-11-19
> **Response to Reviewer A7dr (1/4)**
>
> We thank the reviewer for the insightful and valuable comments. We respond to your comments as follows and sincerely hope that our rebuttal could properly address your concerns. If so, we would deeply appreciate it if you could raise your score and your confidence. If not, please let us know your further concerns, and we will continue actively responding to your comments and improving our work.
>
> ## **Weakness 1**
>
> > Although the reduction in the computational workload of SKR is significant, it is still not clear to me that SKR is a better choice as a generator for NO dataset, since the paper lacks evaluation of the dataset generated by SKR in the context of training NOs. One of my questions in this regard is: How does the performance of NO instances (e.g. FNO, DeepONet) trained on datasets generated by SKR compare to those trained on their original datasets?
>
> - Thank you for your insightful question. Assessing the performance of a dataset generated by an algorithm is indeed very important. Thank you for raising this issue, and **we will explain from both theoretical and experimental perspectives that the data set generated using our SKR algorithm will not affect the training of neural operators**.
>
> 1. **From the theoretical perspective, under specific error tolerances, our SKR algorithm does not alter the outcome of the generated dataset.** As detailed in **Section 5.1**, our SKR algorithm is backed by rigorous convergence analysis, ensuring accurate convergence to the true solution. Especially for linear systems with high correlation, our SKR algorithm demonstrates a superior theoretical convergence rate over GMRES.
>
>    1.  In the field of numerical solutions for large matrices, there are generally no true solutions without error, only numerical solutions with errors smaller than an acceptable threshold. We consider the numerical solution satisfactory when the error of the computed linear system's solution is below the acceptable threshold, which we believe accurately represents the true solution.
>
> 2. **Experimentally, we found that our SKR algorithm does not affect the effectiveness of the generated dataset when training neural operators.**
>
>    1.  For the Darcy flow dataset, we simultaneously generated 5256 data points with a precision of \(10^{-8}\) using both our SKR algorithm and GMRES, of which 5000 were designated for the training set and 256 for the test set, with the matrix size being 2500. The data in the table represent the relative error under the two-norm during the training process, where the first row indicates the number of training epochs and the rightmost column shows the final convergence error.
>
>    2. |             | 0     | 100   | 200   | 300   | 400   | final |
>       | ----------- | ----- | ----- | ----- | ----- | ----- | ----- |
>       | GMRES 1     | 0.912 | 0.127 | 0.096 | 0.055 | 0.025 | 0.020 |
>       | GMRES 2     | 0.927 | 0.144 | 0.094 | 0.054 | 0.026 | 0.020 |
>       | SKR(ours) 1 | 0.886 | 0.132 | 0.085 | 0.055 | 0.026 | 0.020 |
>       | SKR(ours) 2 | 0.912 | 0.145 | 0.088 | 0.051 | 0.028 | 0.020 |
>
>    3.  To mitigate the impact of randomness during the training process, we trained the neural network models twice using datasets generated by GMRES and SKR respectively to compare their training dynamics. **The results of this experiment demonstrate that the datasets generated by our SKR algorithm and GMRES yield identical outcomes when used to train neural operators.**
>
> - Based on the aforementioned theoretical analysis and experimental validation, we can confidently state that the datasets generated by our SKR algorithm and the baseline GMRES algorithm have no impact on the training of neural operators. That is, **the performance of the datasets they generate is equivalent.**

---

> ### Author Response · Authors · 2023-11-19
> **Response to Reviewer A7dr (2/4)**
>
> ## **Weakness 2**
>
> > Speaking from the practical aspect, it might be also beneficial for readers if the authors could give run-time comparison of SKR to parallelized solvers.
>
> - Thank you for your question. The parallelism of our SKR algorithm is indeed a critical component. Typically, matrix algorithms employ three parallelization strategies:
>
> **The First Strategy**： Using MPI to parallelize certain computational processes within the matrix algorithm, such as the Krylov subspace iteration in GMRES, which often relies on specific matrix preconditioning methods. Regarding this MPI-based parallel approach, our experiments have conducted numerous comparative tests in Appendix D.6, and **our SKR algorithm has consistently achieved excellent results.**
>
> **The Second Strategy**： Utilizing MPI to decompose the computational task into several independent parts, each solved separately. As you rightly mentioned in your question, this is indeed an excellent approach. **We will illustrate the** **parallelism** **strategy of our SKR algorithm with a specific example, showcasing its sustained and notable acceleration impact.**
>
> 1. Imagine we have a server CPU with 72 parallel threads. Our parallel SKR algorithm comprises the following parts:
>
>    - Sorting: As with the original SKR, we order these data points using a suitable sorting algorithm into a sequence where consecutive parameters are strongly correlated, meaning the distance between parameter matrices is minimal (we usually use the 1, 2, or infinity norms of matrices as the metric in this Banach space). The choice of sorting algorithm varies with the size of the dataset:
>      1. Greedy Sorting: Simple to implement and negligible in computational cost for smaller datasets, such as our 7200 data points. However, for larger datasets, say $10^7$ data points, the computational load of the greedy algorithm increases substantially and cannot be parallelized across multiple CPUs.
>      2. FFT Dimension Reduction + Fractal Division + Greedy Sorting: For very large datasets, we propose a parallelizable method that first reduces dimensionality via FFT to manage the high-dimensional coordinates, then applies a fractal division algorithm based on the Hilbert curve, and finally, performs greedy sorting within each divided section.
>    - Solving Linear Systems: Assuming the task at hand is to generate a dataset comprising 7200 data points, upon sequencing the data to establish a strong correlation, we subsequently partition these 7200 data points into 72 batches, each containing 100 points, distributing the tasks of solving the corresponding linear systems to different CPU threads. Each thread employs our SKR algorithm for solving, achieving parallelism in the process.
>
> 2.  To demonstrate the feasibility of our parallel approach and its effective acceleration, we have designed the following experiments.
>
>    - An experimental simulation was conducted to generate a dataset for the Helmholtz equation, comprising 7200 data points. Each corresponding matrix has a dimension of 10000, with Successive Over-Relaxation (SOR) used uniformly for matrix preconditioning. Both our SKR algorithm and the baseline GMRES algorithm were implemented to distribute the solving of 7200 tasks across 72 threads, with each thread solving 100 linear systems. Due to the potential variance in computation completion times across threads, the reported computation times and iteration counts have been averaged for consistency.
>
>    - The upper half of the table presents the average computation time taken by both algorithms to solve each linear system, measured in seconds. The lower half details the average number of iterations required to solve each linear system. The first row of the table specifies the precision requirements for solving linear systems.
>
>      - |         |                    | 1E-03 | 1E-05 | 1E-07 |
>        | ------- | ------------------ | ----- | ----- | ----- |
>        | time(s) | Parallel GMRES     | 0.080 | 0.105 | 0.122 |
>        | time(s) | Parallel SKR(ours) | 0.011 | 0.013 | 0.015 |
>        | iter    | Parallel GMRES     | 3734  | 4906  | 5715  |
>        | iter    | Parallel SKR(ours) | 126   | 148   | 167   |
>
>    - **The experimental results reveal that our SKR algorithm achieves a 6.7-8.0 fold acceleration in computation time and a 30-34 fold reduction in the number of iterations compared to the baseline.**

---

> ### Author Response · Authors · 2023-11-19
> **Response to Reviewer A7dr (3/4)**
>
> **The Third Strategy**： **Block Parallel Version**: The block concept is a strategy for parallelizing large matrices and reducing memory overhead. Drawing from the idea of Krylov blocks, we have redesigned a block version of our SKR algorithm to facilitate parallel processing. The fundamental principle involves transforming the conventional matrix algorithm into a block matrix variant, where each block is computed on a distinct CPU, thereby achieving parallel execution of the matrix algorithm. This parallel approach significantly reduces memory usage, making the algorithm suitable for extremely large matrix computations. **We will now demonstrate the remarkable acceleration achieved by our block version of the SKR algorithm through experimental results**:
>
> 1. An experimental simulation was conducted to generate a dataset for the Helmholtz equation. Each corresponding matrix has a dimension of 10000, with Successive Over-Relaxation (SOR) used uniformly for matrix preconditioning, running on 72 parallel MPI threads.
>
> 2. The upper half of the table presents the average computation time taken by both algorithms to solve each linear system, measured in seconds. The lower half details the average number of iterations required to solve each linear system. The first row of the table specifies the precision requirements for solving linear systems.
>
>    - |         |                 | 1E-03 | 1E-05 | 1E-07 |
>      | ------- | --------------- | ----- | ----- | ----- |
>      | time(s) | GMRES           | 3.22  | 4.16  | 4.66  |
>      | time(s) | Block SKR(ours) | 0.015 | 0.025 | 0.030 |
>      | iter    | GMRES           | 5503  | 7074  | 7976  |
>      | iter    | Block SKR(ours) | 224   | 269   | 305   |
>
> 3. **The experimental results reveal that our SKR algorithm achieves a 150-200 fold acceleration in computation time and a 24-26 fold reduction in the number of iterations compared to the baseline.** The significant improvement in computation time is mainly attributed to the effects of MPI parallelization.
>
> - **In summary, our SKR algorithm can be parallelized, and the parallel version maintains a significant acceleration effect**.
>
> ## **Weakness 3**
>
> > In Section 3.1 ”The data points for these PDEs in the parameter space of the NO inputs are relatively dense” : How did the authors withdraw this conclusion? Compared to what PDEs are the inputs dense?
>
> Thank you for your question. We greatly appreciate your attentive reading of my paper and your question, which indicates that my expression was not clear enough. This suggestion is very important to me, and we have revised the wording in **Section 3.1**.
>
> In that statement, we intended to convey that training a neural operator typically necessitates the generation of a large number of numerical solutions for PDEs, ranging from 10^3 to 10^5. **Such a multitude of linear systems, derived from the same distribution of PDEs, naturally exhibit a high degree of similarity. It is precisely this similarity that is key to the effective acceleration of our SKR algorithm's recycle module.**

---

> ### Author Response · Authors · 2023-11-19
> **Response to Reviewer A7dr (4/4)**
>
> ## **Weakness 4**
>
> > Table 5 (and its caption) is entirely confusing and hard to understand.
>
> - Thank you for your question. Table 5 records the comparative data between our SKR algorithm and the baseline GMRES algorithm in Appendix D.6. **The caption indicates that the experiment pertains to the generation of datasets for the Darcy flow problem, using BJacobi as matrix preconditioner, within computing environment env1, and utilizing 72 parallel** **MPI** **threads**.
>   - "env1" refers to the following computing environment:
>     - Platform: Docker version 20.10.0
>     - Operating System: Ubuntu 22.04.3 LTS
>     - Processor: Dual-socket Intel XeonGold 6154 CPU, clocked at 3.00GHz
> - This table is divided into two parts:
>   - the upper half records the average time to solve each linear system for two different algorithms under various conditions, in seconds, with smaller numbers indicating faster computation.
>   - The lower half records the average number of iterations needed to solve each linear system for the two algorithms under different conditions, with fewer iterations indicating better algorithm stability and faster convergence.
> - The first row indicates different required error precisions, and the second column shows the matrix sizes derived from different grid densities. Each row's data is composed of two parts, with the upper half showing the data for the GMRES algorithm—our baseline—and the lower half showing the corresponding data for our SKR algorithm.
> - **The data recorded in Table 5 demonstrates that under the present conditions, our SKR algorithm achieves up to a 4-fold increase in computational speed and up to a 29-fold reduction in the number of iterations compared to GMRES.**
> - Due to space limitations in the main text, we placed this experimental data in the appendix. We conducted nearly 3000 sets of experiments for this project, each simulating the generation process of a real neural operator dataset. The purpose of so many experiments is to demonstrate that our SKR algorithm can effectively accelerate the dataset generation process for neural operators under various conditions.
> - I am very grateful for your careful reading of my paper. The confusion caused by the overly succinct titles was my oversight, and **I have thoroughly checked for similar issues throughout and added detailed annotations in Appendix D.6**. Thank you for your suggestion; it is very important to me.
>
> ## **Weakness 5**
>
> > Caption in Figure 9 “Specifically, when the parameters of the neural operator are closely matched in Helmholtz equations” : how do you compare the parameters of NO to Helmonltz equations?
>
> - Thank you for your question. In Appendix D.2 Figure 9, **I aim to qualitatively convey that when the corresponding parameters of a** **PDE** **are closely aligned, their solutions are intuitively similar as well**. Typically, parameters are represented in tensor form. For the two-dimensional Helmholtz equation, we use parameter matrices to represent different instances of PDEs and **commonly employ the matrix two-norm to measure the distance between these parameter matrices**. Quantitatively speaking, in Figure 9, the two-norm distance between the two parameter matrices is 8, and the distance between their solution matrices, viewed as another matrix, is 12. In contrast, Figure 10 shows parameter matrices with a two-norm distance of 28, and their solution matrices have a distance of 23.
> - From another perspective, neural operators generally handle problems where the mapping from PDE parameters to their corresponding solutions is continuous. That is, when there is a minimal discrepancy in parameters, the differences between PDE solutions are also minor. If the mapping from parameters to solutions is discontinuous, such as with abrupt changes or discontinuities, then neural operators perform poorly. **This continuity is a prerequisite for using neural operators**, and our SKR algorithm similarly requires continuity in the mapping from parameters to solutions.
> - I greatly appreciate your thorough reading of my paper. The issue you've raised has pointed out a lapse in the clarity of my explanation. Your suggestion is invaluable to me, and **I have revised the expression in Appendix D.2 to be more clear and explicit**.

---

> ### Comment · Reviewer_A7dr · 2023-11-21
> **Strong response from the authors.**
>
> I appreciate for the authors’ answers. Now that all my concerns including my major concern about the quality of data are resolved, I believe the paper addresses very interesting problem and its contribution is significant. Thus, given the contributions of the paper as well as the authors’ willingness to provide explanations and a considerable amount of additional experimental results, I would recommend the paper for acceptance and am raising my score to 8.

---

### Official Review · Reviewer_7U2z · 2023-11-03

**Soundness:** 2 fair
**Presentation:** 3 good
**Contribution:** 3 good
**Rating:** 6
**Confidence:** 3

**Summary:**

This paper proposes a new method, SKR, to speed up the process of repeatedly solving a given PDE with different parameters (e.g. initial conditions, boundary conditions, right-hand side, etc.). This scenario arises when generating data to train neural operators, as this process effectively entails solving the same PDE multiple times with different input parameters. Solving a PDE often involves a potentially very large linear system, which can be solved using Krylov methods. SKR leverages the fact that the same PDE is solved across samples with different parameters to accelerate the data generation process, thanks to Krylov subspace recycling and a specific sorting algorithm.

**Strengths:**

This work addresses the important problem of data generation for NO training. The idea relies on the well-known Krylov subspace recycling method and, to the best of my knowledge, has not been previously proposed.

**Weaknesses:**

The paper does not discuss the aspect of parallelism, which raises concerns about the fairness of the benchmark. More specifically, in the traditional setting, PDE solves are independent of each other. Consequently, data generation can be achieved by splitting the PDE systems to solve across different MPI processes to reduce the cost and achieve faster data generation. However, in the SKR setting, the PDE solves are no longer independent, which undermines the claimed advantage of SKR over the classical approach.

**Questions:**

Is there a plan to add support for widely used PDE software such as Firedrake, FEniCS, etc.? This would enable data generation for neural operator training with just a few lines of code using SKR in high-level languages like FEniCS or Firedrake, enhancing the method's dissemination.

---

> ### Author Response · Authors · 2023-11-19
> **Response to Reviewer 7U2z （1/3）**
>
> We thank the reviewer for the insightful and valuable comments. We respond to your comments as follows and sincerely hope that our rebuttal could properly address your concerns. If so, we would deeply appreciate it if you could raise your score and your confidence. If not, please let us know your further concerns, and we will continue actively responding to your comments and improving our work.
>
> ## **Weakness**
>
> > The paper does not discuss the aspect of parallelism, which raises concerns about the fairness of the benchmark. More specifically, in the traditional setting, PDE solves are independent of each other. Consequently, data generation can be achieved by splitting the PDE systems to solve across different MPI processes to reduce the cost and achieve faster data generation. However, in the SKR setting, the PDE solves are no longer independent, which undermines the claimed advantage of SKR over the classical approach.
>
> - Thank you for raising this essential aspect of our study. The parallelism of our SKR algorithm is indeed a critical component. Typically, matrix algorithms employ three parallelization strategies:
>
> **The First Strategy**： Using MPI to parallelize certain computational processes within the matrix algorithm, such as the Krylov subspace iteration in GMRES, which often relies on specific matrix preconditioning methods. Regarding this MPI-based parallel approach, our experiments have conducted numerous comparative tests in Appendix D.6, and **our SKR algorithm has consistently achieved excellent results.**

---

> ### Author Response · Authors · 2023-11-19
> **Response to Reviewer 7U2z （2/3）**
>
> **The Second Strategy**： Utilizing MPI to decompose the computational task into several independent parts, each solved separately. As you rightly mentioned in your question, this is indeed an excellent approach. **We will illustrate the** **parallelism** **strategy of our SKR algorithm with a specific example, showcasing its sustained and notable acceleration impact.**
>
> 1. Imagine we have a server CPU with 72 parallel threads. Our parallel SKR algorithm comprises the following parts:
>
>    - Sorting: As with the original SKR, we order these data points using a suitable sorting algorithm into a sequence where consecutive parameters are strongly correlated, meaning the distance between parameter matrices is minimal (we usually use the 1, 2, or infinity norms of matrices as the metric in this Banach space). The choice of sorting algorithm varies with the size of the dataset:
>      1. Greedy Sorting: Simple to implement and negligible in computational cost for smaller datasets, such as our 7200 data points. However, for larger datasets, say $10^7$ data points, the computational load of the greedy algorithm increases substantially and cannot be parallelized across multiple CPUs.
>      2. FFT Dimension Reduction + Fractal Division + Greedy Sorting: For very large datasets, we propose a parallelizable method that first reduces dimensionality via FFT to manage the high-dimensional coordinates, then applies a fractal division algorithm based on the Hilbert curve, and finally, performs greedy sorting within each divided section.
>    - Solving Linear Systems: Assuming the task at hand is to generate a dataset comprising 7200 data points, upon sequencing the data to establish a strong correlation, we subsequently partition these 7200 data points into 72 batches, each containing 100 points, distributing the tasks of solving the corresponding linear systems to different CPU threads. Each thread employs our SKR algorithm for solving, achieving parallelism in the process.
>
> 2.  To demonstrate the feasibility of our parallel approach and its effective acceleration, we have designed the following experiments.
>
>    - An experimental simulation was conducted to generate a dataset for the Helmholtz equation, comprising 7200 data points. Each corresponding matrix has a dimension of 10000, with Successive Over-Relaxation (SOR) used uniformly for matrix preconditioning. Both our SKR algorithm and the baseline GMRES algorithm were implemented to distribute the solving of 7200 tasks across 72 threads, with each thread solving 100 linear systems. Due to the potential variance in computation completion times across threads, the reported computation times and iteration counts have been averaged for consistency.
>
>    - The upper half of the table presents the average computation time taken by both algorithms to solve each linear system, measured in seconds. The lower half details the average number of iterations required to solve each linear system. The first row of the table specifies the precision requirements for solving linear systems.
>
>      - |         |                    | 1E-03 | 1E-05 | 1E-07 |
>        | ------- | ------------------ | ----- | ----- | ----- |
>        | time(s) | Parallel GMRES     | 0.080 | 0.105 | 0.122 |
>        | time(s) | Parallel SKR(ours) | 0.011 | 0.013 | 0.015 |
>        | iter    | Parallel GMRES     | 3734  | 4906  | 5715  |
>        | iter    | Parallel SKR(ours) | 126   | 148   | 167   |
>
>    - **The experimental results reveal that our SKR algorithm achieves a 6.7-8.0 fold acceleration in computation time and a 30-34 fold reduction in the number of iterations compared to the baseline.**

---

> ### Author Response · Authors · 2023-11-19
> **Response to Reviewer 7U2z （3/3）**
>
> **The Third Strategy**： **Block Parallel Version**: The block concept is a strategy for parallelizing large matrices and reducing memory overhead. Drawing from the idea of Krylov blocks, we have redesigned a block version of our SKR algorithm to facilitate parallel processing. The fundamental principle involves transforming the conventional matrix algorithm into a block matrix variant, where each block is computed on a distinct CPU, thereby achieving parallel execution of the matrix algorithm. This parallel approach significantly reduces memory usage, making the algorithm suitable for extremely large matrix computations. **We will now demonstrate the remarkable acceleration achieved by our block version of the SKR algorithm through experimental results**:
>
> 1. An experimental simulation was conducted to generate a dataset for the Helmholtz equation. Each corresponding matrix has a dimension of 10000, with Successive Over-Relaxation (SOR) used uniformly for matrix preconditioning, running on 72 parallel MPI threads.
>
> 2. The upper half of the table presents the average computation time taken by both algorithms to solve each linear system, measured in seconds. The lower half details the average number of iterations required to solve each linear system. The first row of the table specifies the precision requirements for solving linear systems.
>
>    - |         |                 | 1E-03 | 1E-05 | 1E-07 |
>      | ------- | --------------- | ----- | ----- | ----- |
>      | time(s) | GMRES           | 3.22  | 4.16  | 4.66  |
>      | time(s) | Block SKR(ours) | 0.015 | 0.025 | 0.030 |
>      | iter    | GMRES           | 5503  | 7074  | 7976  |
>      | iter    | Block SKR(ours) | 224   | 269   | 305   |
>
> 3. **The experimental results reveal that our SKR algorithm achieves a 150-200 fold acceleration in computation time and a 24-26 fold reduction in the number of iterations compared to the baseline.** The significant improvement in computation time is mainly attributed to the effects of MPI parallelization.
>
> - **In summary, our SKR algorithm can be parallelized, and the parallel version maintains a significant acceleration effect**.
>
> ## **Question**
>
> > Is there a plan to add support for widely used PDE software such as Firedrake, FEniCS, etc.? This would enable data generation for neural operator training with just a few lines of code using SKR in high-level languages like FEniCS or Firedrake, enhancing the method's dissemination.
>
> - Your suggestion is indeed valuable and aligns with the objectives we are actively pursuing. As you have pointed out, **we are planning to integrate our algorithm into FEniCSx as a** **library function**. This integration will allow users to employ our algorithm simply by calling this function, akin to selecting any other existing solver for linear systems.
> - In fact, we had initiated plans for this integration some time ago. The initial implementation of our SKR algorithm was developed within the Python scipy environment, with the baseline comparison made against scipy's gmres. FEniCSx, being a frequently used Python finite element library by our team, which I am aware embeds the PETSc and MPI libraries. PETSc is notable for its cutting-edge Krylov subspace iteration implementations. To enhance the speed of our algorithm and to facilitate an objective comparison with the latest C-version of gmres (our baseline), **as well as to** **streamline** **future integration into the FEniCSx finite element library, we have rewritten our core code to rely on the PETSc library.**
> - The code used for the experiments in our paper, now modularized into several components, can be easily utilized in any Linux environment with the C and Python versions of the PETSc and MPI libraries installed. Moving forward, **we plan to use the PETSc library interface within FEniCSx to craft it into a** **library function** **for FEniCSx, simplifying the process for users to invoke it effortlessly. We will be updating the code accordingly.**

---

> ### Author Response · Authors · 2023-11-22
> **We are looking forward to your further comments and/or questions.**
>
> Dear Reviewer 7U2z，
>
> Thanks again for your valuable comments and constructive suggestions, which are of great help to improve the quality of our work. We are looking forward to your further comments and/or questions.
>
> We sincerely hope that our rebuttal has properly addressed your concerns. If so, we would deeply appreciate it if you could raise your score or confidence. If not, please let us know your further concerns, and we will continue actively responding to your comments and/or questions.
>
> Best,
>
> Authors

---

### Official Review · Reviewer_ZpMg · 2023-11-04

**Soundness:** 2 fair
**Presentation:** 2 fair
**Contribution:** 2 fair
**Rating:** 6
**Confidence:** 2

**Summary:**

The paper addresses a fundamental challenge in the training of Neural Operators (NOs) for solving Partial Differential Equations (PDEs) by introducing a novel algorithm called Sorting Krylov Recycling (SKR). The primary challenge is the time-consuming and computationally expensive process of generating labeled data for training NOs, which involves solving numerous systems of linear equations. The proposed SKR algorithm is designed to improve the efficiency of solving these systems and significantly accelerate data generation for NOs training.

**Strengths:**

Innovative Algorithm: The SKR algorithm is a novel and innovative approach to accelerate the generation of training data for NOs, focusing on the efficient resolution of linear systems. It addresses a fundamental challenge in the field of data-driven PDE solvers.

Efficiency Improvement: The paper demonstrates that SKR can achieve a remarkable speedup in the generation of training data, with a potential acceleration of up to 13.9 times. This significant efficiency improvement has the potential to reduce computational costs in the training of NOs.

Real-World Relevance: The need for efficient data generation for NOs is of practical importance, especially in scientific domains where PDEs play a crucial role, such as climate modeling, fluid dynamics, and electromagnetism.

**Weaknesses:**

Complexity: While SKR is a promising algorithm, the paper does not discuss its potential complexities or challenges in practical implementation. It is essential to evaluate the algorithm's feasibility and usability in real-world scenarios.

Application Scope: The paper primarily focuses on the problem of generating training data for NOs. It would be beneficial to discuss broader applications or scenarios where SKR could be applied beyond this specific context.

Generalization: The paper does not extensively discuss the generalizability of the SKR algorithm to different types of PDEs or scenarios. It is important to assess whether SKR is applicable in a wide range of PDE-related problems.

**Questions:**

Could you provide more details about the specific types of PDEs or problems where the SKR algorithm is expected to have the most significant impact in terms of efficiency improvement?

The paper mentions that SKR can achieve a remarkable speedup. Are there specific parameters or settings that are critical for achieving this speedup, and are there scenarios where the speedup might be less pronounced?

Have you considered potential challenges or limitations in implementing the SKR algorithm in practical applications, and are there strategies to address these challenges?

Beyond the context of generating training data for NOs, are there other domains or problems where SKR's approach of optimizing linear system solutions could be applied effectively?

How does the SKR algorithm handle variations in the complexity of PDEs, and does it exhibit consistent efficiency improvements across different levels of complexity?

---

> ### Author Response · Authors · 2023-11-19
> **Response to Reviewer ZpMg (1/5)**
>
> We thank the reviewer for the insightful and valuable comments. We respond to your comments as follows and sincerely hope that our rebuttal could properly address your concerns. If so, we would deeply appreciate it if you could raise your score and your confidence. If not, please let us know your further concerns, and we will continue actively responding to your comments and improving our work.
>
> ## **Weakness 1 & Question 3**
>
> > Complexity: While SKR is a promising algorithm, the paper does not discuss its potential complexities or challenges in practical implementation. It is essential to evaluate the algorithm's feasibility and usability in real-world scenarios.
>
> > Have you considered potential challenges or limitations in implementing the SKR algorithm in practical applications, and are there strategies to address these challenges?
>
> - Thank you for your question. Here are the potential challenges and limitations that may arise when applying the SKR algorithm in practice, as well as the strategies we have devised to address these issues:
>
> 1. **Practicality and usability** are key metrics for assessing an algorithm's value. Our SKR algorithm has been compartmentalized into independent modules, allowing for dataset generation by simply installing the necessary base libraries on a Linux system according to user needs. However, this is not the epitome of convenience. Therefore, we have reconstructed our SKR algorithm based on the PETSc library, and all experiments within this paper utilize the revised code. Considering that large Python finite element libraries like FEniCSx have integrated PETSc, our next step is to integrate our code into the FEniCSx library as one of its functions. **This integration will enable users to deploy our algorithm effortlessly with just a single line of code.**
>
> 2. As discussed in Section 7 Limitation of the paper, **optimizing for specific PDE matrix structures presents a potential challenge**. Our algorithm in the paper is designed for the most general and common scenarios—non-symmetric matrices. In practical applications, one may encounter a variety of matrix structures, such as block matrices or symmetric or symmetric positive-definite matrices, necessitating tailored optimizations of our SKR algorithm for enhanced performance. We have accomplished the implementation of the block parallel version of our SKR algorithm, and the designs for the versions tailored to symmetric and symmetric positive-definite matrices are fully developed. Nevertheless, given the brief period allocated for this rebuttal, we have not yet translated these designs into their respective coding implementations. **In the future, we plan to actualize these specialized versions and integrate them into the FEniCSx library.**
>
>    1. **Block parallel version**: The block concept is an approach to parallelize large matrices and reduce memory overhead. Inspired by the block Krylov idea, we have redesigned a block version of our SKR algorithm and conducted preliminary experiments:
>
>       1. An experimental simulation was conducted to generate a dataset for the Helmholtz equation. Each corresponding matrix has a dimension of 10000, with SOR matrix preconditioning, running on 72 parallel MPI threads.
>
>       2. The upper half of the table presents the average computation time taken by both algorithms to solve each linear system, measured in seconds. The lower half details the average number of iterations required to solve each linear system. The first row of the table specifies the precision requirements for solving linear systems.
>
>          - |         |                 | 1E-03 | 1E-05 | 1E-07 |
>            | ------- | --------------- | ----- | ----- | ----- |
>            | time(s) | GMRES           | 3.22  | 4.16  | 4.66  |
>            | time(s) | Block SKR(ours) | 0.015 | 0.025 | 0.030 |
>            | iter    | GMRES           | 5503  | 7074  | 7976  |
>            | iter    | Block SKR(ours) | 224   | 269   | 305   |
>
>       3. **The experimental results reveal that our SKR algorithm achieves a 150-200 fold acceleration in computation time and a 24-26 fold reduction in the number of iterations compared to the baseline. The significant improvement in computation time is mainly attributed to the effects of** **MPI** **parallelization.**
>
> 3. The SKR algorithm in our paper achieves acceleration by eliminating redundant computations in the neural operator dataset generation process. As discussed in Section 7 Limitation of the paper, although our SKR algorithm is highly effective, it operates at the level of solving linear systems without considering the influence of the PDE forms themselves. For real-world applications, PDEs generally serve as explicit prior information. Moving forward, **we plan to develop SKR variants tailored to specific PDEs, starting from the geometric-algebraic properties of** **PDE** **operators and their characteristic functions, to achieve even better acceleration results.**

---

> ### Author Response · Authors · 2023-11-19
> **Response to Reviewer ZpMg (2/5)**
>
> ## **Weakness 2 & Question 4**
>
> > Application Scope: The paper primarily focuses on the problem of generating training data for NOs. It would be beneficial to discuss broader applications or scenarios where SKR could be applied beyond this specific context.
>
> > Beyond the context of generating training data for NOs, are there other domains or problems where SKR's approach of optimizing linear system solutions could be applied effectively?
>
> - Thank you for posing this insightful question. Initially, when utilizing neural operators, We encountered serious time-consuming problem in dataset generation. After in-depth research into Krylov methods, we developed the SKR algorithm, which has proven to be highly effective. Neural operators are among the most renowned and efficient data-driven approaches for solving PDE problems, and the time-consuming problem of dataset generation has become a substantial impediment in this field. Hence, our paper focused on this context.
> - However, **the applicability of our SKR algorithm extends beyond neural operator problems and can be effectively employed in other domains** such as:
>   - **Data-Driven PDE Algorithms**: There are various data-driven PDE solving or parameter optimization algorithms in both traditional heuristic and AI for PDE domains. These algorithms often require the frequent generation of large volumes of PDE data for datasets, a process that our SKR algorithm can accelerate. For instance, [1] [2].
>   - **Specific Physical Problem Solving**: In certain scenarios, modeling the physical problems at hand translates into solving multiple related linear systems. Our SKR algorithm is adept at efficiently handling such issues. For instance, this is evident in the modeling of fatigue and fracture through finite element analysis [3], which employs dynamic loading across numerous steps, resulting in a substantial number of interrelated linear systems. Similarly, in the resolution process of lattice quantum field theory [4], the end goal involves solving a large collection of correlated linear systems.
> - Reference
>   - [1] Daniel Greenfeld, Meirav Galun, Ron Kimmel, Irad Yavneh, Ronen Basri. "Learning to Optimize Multigrid PDE Solvers."  International Conference on Machine Learning (ICML). 2019.
>   - [2] Jun-Ting Hsieh, Shengjia Zhao, Stephan Eismann, Lucia Mirabella, Stefano Ermon. "Learning Neural PDE Solvers with Convergence Guarantees." International Conference on Learning Representation (ICLR). 2019.
>   - [3] A. Gullerud, R. H. Dodds. "MPI-based implementation of a PCG solver using an EBE architecture and preconditioner for implicit, 3-D finite element analyses." Comput. & Structures, 79 (2001), pp. 553–575.
>   - [4] Shin Muroya, Atsushi Nakamura, Chiho Nonaka, Tetsuya Takaishi. "Lattice QCD at Finite Density: An Introductory Review." Progress of Theoretical Physics, Volume 110, Issue 4, October 2003, Pages 615–668.

---

> ### Author Response · Authors · 2023-11-19
> **Response to Reviewer ZpMg (3/5)**
>
> ## **Weakness 3 & Question 5**
>
> > Generalization: The paper does not extensively discuss the generalizability of the SKR algorithm to different types of PDEs or scenarios. It is important to assess whether SKR is applicable in a wide range of PDE-related problems.
>
> > How does the SKR algorithm handle variations in the complexity of PDEs, and does it exhibit consistent efficiency improvements across different levels of complexity?
>
> - Thank you for your question. Generalizability is a critical criterion for the assessment of a foundational linear systems solver like SKR and its capability to expedite the generation of neural operator datasets.
> - To demonstrate the generalizability of our SKR algorithm, our experimental design incorporated diverse scenarios and complexities. Due to space constraints in the main text, we have detailed the relevant experimental data in Appendix D.6. We designed different problems from several perspectives: 1. Data set (types of PDEs), 2. Precondition, 3. Tolerance, and 4. Matrix Dimensionality (grid density). Different types of PDEs and matrix sizes represent varying complexities of PDEs. Let us explain these aspects individually:
>   - **Data set (types of PDEs)**: with detailed descriptions available in **Appendix D.2**.
>     1.  We chose four different PDEs from various physical problems commonly addressed in the neural operator field, to demonstrate our SKR algorithm's efficacy across PDE types. These include:
>         -  Darcy flow problems from fluid dynamics.
>         -  Heat conduction equations from thermal problems.
>         -  Poisson equations often involved in gravitational and field computations.
>         -  Helmholtz equations relevant to wave propagation and electromagnetic field simulations.
>     2. Each of these has different physical backgrounds and PDE forms. To showcase our algorithm's generalizability, we employed several generation methods for the specific PDE problems:
>        -  Gaussian random fields were used to simulate real physical scenarios in Darcy flow and Helmholtz equations.
>        -  A range of random numbers for generating boundary temperatures in thermal problems.
>        -  Truncated Chebyshev polynomials to generate boundary condition functions for Poisson's equations.
>   - **Precondition**: In real numerical PDE solving, professionals often select appropriate preconditioning methods based on matrix properties and expertise to expedite the solution process. To show the generalizability of the SKR algorithm across different preconditioning methods, we tested all common ones, including None, Jacobi, BJacobi, SOR, ASM, ICC, and ILU, with detailed descriptions available in **Appendix D.3**.
>   - **Tolerance**: Different scenarios may require different computational precisions, and algorithms may converge at varying speeds depending on the precision. To affirm the SKR's generalizability, we tested each problem with over five different precision levels, covering most common scenarios.
>   - **Matrix Dimensionality (grid density).**: The choice of PDE discretization methods and grid density in various contexts leads to different matrix sizes in the linear systems. Different matrix sizes represent varying complexities of PDEs, and algorithms may perform differently at each size. To validate SKR's generalizability, we tested each problem with over five different matrix sizes, ranging from $10^3$ to $10^5$.
> - To thoroughly verify the algorithm's performance, **we conducted nearly 3000 experimental combinations based on the above aspects, each simulating the real process of generating a neural operator dataset, comparing the iteration counts and computation times of our SKR algorithm with the original method.** All experimental results are provided in **Appendix D.6**. In nearly all cases, our SKR algorithm demonstrated significantly better results than the baseline, with reduced computation time, fewer iterations, and greater computational stability. **These experiments robustly affirm the strong generalizability of our algorithm.**

---

> ### Author Response · Authors · 2023-11-19
> **Response to Reviewer ZpMg (4/5)**
>
> ## **Question 1**
>
> > Could you provide more details about the specific types of PDEs or problems where the SKR algorithm is expected to have the most significant impact in terms of efficiency improvement?
>
> - Thank you for your question. As shown by the experimental results in Section 6.2, **we have observed that our SKR algorithm is highly effective when generating datasets for the Helmholtz Equation, achieving up to a 13.9-fold increase in average computational speed and a reduction of nearly 30 times in the average number of iterations.**
> - The exceptional performance of our SKR algorithm in this context may be attributed to the following factors:
>   - Type of PDE and its generation method: The mapping between the parameters and solutions of the PDE in question is relatively smooth. Additionally, we employ Gaussian random fields to generate parameters that simulate real physical scenarios. These elements result in a strong correlation between the generated linear systems, allowing our SKR algorithm to significantly reduce redundant computations and, consequently, enhance efficiency.
>   - Discretization method and boundary conditions of the PDE: We utilize finite difference methods that maintain a fixed matrix size for this problem. Moreover, the boundary conditions we set are continuous and stable, which contributes to the consistency of the resulting matrix properties. This consistency is conducive to our SKR algorithm's ability to recycle key information during the solution process and thereby achieve an acceleration effect.
> - As mentioned in Appendix D.2, the Helmholtz Equation is a commonly encountered issue in physics, appearing extensively in fields such as Acoustics, Electromagnetism, and Quantum Mechanics. Neural operators are an efficient method for solving the Helmholtz Equation, and our SKR algorithm can significantly lower the barrier to using neural operators. **In many real-world scenarios, the complex physical environments lead to large matrix sizes for the Helmholtz Equation, making it challenging to efficiently generate the datasets required for training neural operators. Our SKR algorithm greatly reduces this overhead, enabling the wide application of neural operators in this domain.**

---

> ### Author Response · Authors · 2023-11-19
> **Response to Reviewer ZpMg (5/5)**
>
> ## **Question 2**
>
> > The paper mentions that SKR can achieve a remarkable speedup. Are there specific parameters or settings that are critical for achieving this speedup, and are there scenarios where the speedup might be less pronounced?
>
> - Thank you for your query, which we will address in order.
>
> 1. Regarding whether **specific parameters** influence the effectiveness of our algorithm, our SKR algorithm is a linear equation solver based on Krylov subspace iterations, and there are two critical parameters:
>
>    1. **The** **restart parameter**: This parameter is associated with the Krylov algorithm needing to restart after iterating a subspace of a certain dimensionality, to conserve memory and prevent the accumulation of floating-point errors. Typically, the restart parameter for Krylov methods is an empirical value, which has a minor effect on performance within a reasonable range. In our paper, to ensure fairness, both our SKR algorithm and the baseline GMRES use the same restart parameter, set to 40. To further illustrate its potential impact, we conducted the following supplementary experiments:
>
>       1. A simulation was performed to create a dataset for the Helmholtz equation, with each matrix sized at 10000 and SOR matrix preconditioning. All experiments were carried out in the Env1 environment with an error precision set to 1E-07, and the first row of the table reflects varying restart parameters. Specific details can be found in Appendix D.4.
>
>          - |         |           | 35   | 40   | 45   | 50   |
>            | ------- | --------- | ---- | ---- | ---- | ---- |
>            | time(s) | GMRES     | 4.60 | 4.66 | 4.73 | 4.74 |
>            | time(s) | SKR(ours) | 0.42 | 0.44 | 0.46 | 0.47 |
>            | iter    | GMRES     | 8005 | 7976 | 7949 | 7890 |
>            | iter    | SKR(ours) | 321  | 308  | 295  | 280  |
>
>          -    Our findings suggest that within a reasonable scope, **the restart parameter has a negligible effect on the algorithm's performance, and the appropriate value is typically selected based on hardware capabilities.**
>
>    2. **The recycle restart parameter**: In mathematical terms, for our SKR algorithm, this represents the number of harmonic Ritz vectors to compute during the recycling of the subspace. Generally, the Recycle restart parameter is set empirically and is observed to have a minor impact on the algorithm's performance within a reasonable range. In our paper, to maintain consistency across our experiments, we have used the same restart parameter for our SKR algorithm, set to 20. Further supplementary experiments were conducted to elucidate its potential effects:
>
>       1.  The experimental setup and related configurations were the same as the previous experiments, with the first row of the table indicating different Recycle restart parameters.
>
>          - |         |           | 15   | 20   | 25   |
>            | ------- | --------- | ---- | ---- | ---- |
>            | time(s) | SKR(ours) | 0.43 | 0.44 | 0.46 |
>            | iter    | SKR(ours) | 332  | 308  | 285  |
>
>          -    The experiments showed that within a reasonable range, **the restart parameter has a minimal impact on algorithm performance.**
>
> 2. In response to whether **specific settings** could affect the performance, assessing the algorithm's applicability across various scenarios and contrasting its effects under different settings is crucial. As detailed in the response to your **Weakness 3 & Question 5** query, we conducted nearly 3000 experiments from different perspectives, as seen in Appendix D.6, which thoroughly demonstrate our algorithm's adaptability to various scenarios. As analyzed in Section 6.2, through extensive experimentation, we have also identified some settings that could significantly influence performance:
>
>    1. **Related to PDE Generation Methods**: Our SKR algorithm performs better on problems generated using GRF compared to those formulated with truncated Chebyshev polynomials.
>    2. **Pertaining to** **Precision** **and Matrix Size**: The larger the matrix and the higher the precision required, the more effective our SKR algorithm becomes.
>
> 3. Regarding scenarios where the acceleration might be less pronounced, indeed, **there are situations where the acceleration effect could be suboptimal or less evident**. Our analysis, based on numerous experiments, suggests that the following scenarios may yield poorer results:
>
>    1. **Matrix precondition**: As mentioned in Section 6.2, certain matrix preconditioning methods, such as ILU and ICC, may disrupt the continuity of the invariant subspace between successive matrices, resulting in a mediocre acceleration effect of our SKR algorithm.
>    2. **PDE** **types**: For instance, in Thermal Problem discussed in the main text, some PDEs exhibit non-smooth mappings or discontinuities between parameters and PDE solutions. For dataset generation for these types of PDEs, the acceleration effect of our SKR algorithm is generally moderate.

---

> ### Author Response · Authors · 2023-11-22
> **We are looking forward to your further comments and/or questions.**
>
> Dear Reviewer ZpMg，
>
> Thanks again for your valuable comments and constructive suggestions, which are of great help to improve the quality of our work. We are looking forward to your further comments and/or questions.
>
> We sincerely hope that our rebuttal has properly addressed your concerns. If so, we would deeply appreciate it if you could raise your score or confidence. If not, please let us know your further concerns, and we will continue actively responding to your comments and/or questions.
>
> Best,
>
> Authors

---

### Official Review · Reviewer_BjLB · 2023-11-07

**Soundness:** 3 good
**Presentation:** 3 good
**Contribution:** 2 fair
**Rating:** 8
**Confidence:** 4

**Summary:**

In this paper, authors proposed an iterative method called Sorting Krylov Recycling (SKR) to boost the efficiency of solving systems of linear equations, thus accelerating data generation for neural operators training. The proposed method leverages the power of Krylov subspace recycling and a sorting algorithm to arrange interrelated systems in a sequence, where adjacent systems exhibit high similarities.
The effectiveness of the SKR method was demonstrated through both theoretical analysis and extensive experiments. The SKR performance was compared with other state-of-the-art methods, such as GMRES, and by considering the impact of different preconditioning techniques on the convergence rate and computational efficiency. The results indicate that SKR can deliver impressive acceleration, reducing the wall clock time by factor of up to 13.9 and requiring up to 30 times fewer iterations.

**Strengths:**

- Paper is well-written and overall easy to follow, with additional useful materials in appendices.
- Extensive experiments and nice ablation studies to illustrate the impact of different components of the proposed algorithm
- The code and the reproducibility details are useful for other researchers to leverage the developments for their data driven PDE solvers, or benchmark their ongoing/future studies on related topics.

**Weaknesses:**

- It would have been nice if authors provided more elaboration on the results for different types of PDEs, in particular for Poisson equation, for which the SKR and GMRES perform comparably to some extent.
- In addition to time and iteration count, what about the numerical accuracy of their solvers w.r.t the grid resolution? Also, how can one interpret the slopes of plots indicating tolerance vs time and tolerance vs interations?

**Questions:**

- One general question for me was whether this approach can be used beyond neural operators, and can be used for any data driven PDE solvers? I did not quite understand the necessity of using NO to generate PDE, and it would be great if authors could elaborate on this.
- For many problems such as Poisson equation the resulting matrix, A, is symmetric and using iterative solvers for symmetric case (such as CG solver) could significantly help speed up the process. with that being said, could authors modify the algorithm by using symmetry property to make it perform better for symmetric matrices?

**Details Of Ethics Concerns:**

No concern

---

> ### Author Response · Authors · 2023-11-19
> **Response to Reviewer BjLB (1/4)**
>
> We thank the reviewer for the insightful and valuable comments. We respond to your comments as follows and sincerely hope that our rebuttal could properly address your concerns. If so, we would deeply appreciate it if you could raise your score and your confidence. If not, please let us know your further concerns, and we will continue actively responding to your comments and improving our work.
>
> ## **Weakness 1**
>
> > It would have been nice if authors provided more elaboration on the results for different types of PDEs, in particular for Poisson equation, for which the SKR and GMRES perform comparably to some extent.
>
> - Thank you for your question. As demonstrated in Section 6.2 of our paper, the SKR algorithm generally performs moderately on the Poisson equation, achieving a speedup of only 1.0 to 2.1 times, which is significantly less than the speedup observed for other problems. The variance in performance across different types of PDEs may be attributed to several factors:
>   - **The method of PDE problem generation**: As mentioned in **Appendix D.2**, to assess performance across diverse scenarios, we selected different physical problems and methods for generating PDEs. Specifically:
>     1. For the Darcy flow problem and Helmholtz equation, we employed Gaussian random fields to generate simulations of real physical scenarios.
>     2. In the Thermal Problem, we used a range of random numbers to generate boundary temperatures for the heat conduction issue.
>     3. For the Poisson equation, we generated boundary condition functions using truncated Chebyshev polynomials.
>     4.   The continuity of the mapping between the parameters and solutions varies with the generation method, as does the relatedness of the resulting linear systems. **In the case of the Poisson equation, our use of coefficients from truncated Chebyshev polynomials may lead to weak inter-system relations, which in turn might result in the modest performance of our SKR algorithm.**
>   - **The type of PDE and** **discretization** **method**: As described in **Appendix D.2**, to test the algorithm's effectiveness in different settings, we chose various PDE boundaries and discretization methods. In particular:
>     1. For the Darcy flow problem, Helmholtz equation, and Poisson equation, we discuss PDEs with square boundaries and discretize using the finite difference method.
>     2. For the Thermal Problem, we discuss PDEs with irregular boundaries and discretize using the finite element method with triangular meshes.
>     3.   The nature of the PDEs' invariant subspaces varies with the type of PDE, and different discretization methods could alter the mathematical significance of the linear systems' subspaces, ultimately affecting the performance of the SKR algorithm.

---

> ### Author Response · Authors · 2023-11-19
> **Response to Reviewer BjLB (2/4)**
>
> ## **Weakness 2.1**
>
> > In addition to time and iteration count, what about the numerical accuracy of their solvers w.r.t the grid resolution?
>
> - Thank you for your question. In the process of numerically solving and generating PDE data, there are two primary sources of error:
>   - Error due to discretization methods: To numerically solve PDEs, we often employ methods such as finite difference (FDM) or finite element methods (FEM). These approaches discretize the continuous PDE problem into a linear system of equations based on a discrete grid, inherently introducing an error. This error is dependent on the type of PDE, the choice of discretization method, and the grid resolution.
>   - Error from solving the linear system of equations: As PDEs typically result in large systems of equations that are impractical to solve analytically, iterative methods must be used. Common iterative methods, like the GMRES algorithm involving Krylov subspace iterations, require a predefined stopping criterion, signaling the algorithm to halt when the error reduces below a specified threshold.
> - The actual error in PDE solutions is a combination of these two types. For a given discretization method, increasing grid resolution can reduce the discretization error but at the cost of substantially increasing the computation time and iterations required to solve the larger resulting linear system. If the grid resolution is increased without also allowing for more computation time, the error from solving the linear system can increase.
> - **Our SKR algorithm is focused on reducing the computational effort required to solve linear systems**; thus, the errors discussed in our paper pertain to those arising from solving linear systems. To demonstrate the effectiveness of our algorithm across various grid densities, we considered different grid resolutions in the supplementary experiments detailed in **Appendix D.6**. Here, grid density is reflected in the size of the matrices within the linear systems (larger grid density implies larger matrix size), and **our algorithm achieved excellent results across these varied resolutions**.
> - In practical applications, adjusting the relationship between grid density and the precision of solving linear systems depends on the specifics of the problem at hand, but this is beyond the scope of our current study.
>
> ## **Weakness 2.2**
>
> > How can one interpret the slopes of plots indicating tolerance vs time and tolerance vs interactions?
>
> - Thanks to your reminder, we have now included a thorough analysis in **Appendix D.5** of the paper. Here we elaborate on the purpose and significance of the slopes of plots analysis here.
> - The creation of these plots and the calculation of their slopes were intended to qualitatively demonstrate that, given the same time expenditure or number of iterations, **our SKR algorithm converges faster than GMRES**. However, to better measure and compare the convergence rates of the two algorithms, one must not only consider the overall convergence but also quantitatively compare their convergence during the high-precision phase.
> - GMRES and similar Krylov subspace convergence algorithms exhibit a phenomenon known as superlinear convergence phenomenon in **Appendix C.1**, which means in a log-error versus time or iterations graph, the GMRES convergence curve can be interpreted as a combination of two segments (the first with a smaller absolute slope value, followed by a larger one). Due to the presence of superlinear convergence, an equitable and objective assessment of our SKR and GMRES in the high-precision phase requires discarding the data from the low-precision phase. Therefore, we selected three high-precision points for each algorithm to perform a linear fit and obtain the convergence slopes for this phase. A comparative analysis revealed that our algorithm's slope has a greater absolute value, indicating that **our SKR also converges  faster than GMRES in the high-precision stage**.

---

> ### Author Response · Authors · 2023-11-19
> **Response to Reviewer BjLB (3/4)**
>
> ## **Questions 1**
>
> > One general question for me was whether this approach can be used beyond neural operators, and can be used for any data driven PDE solvers? I did not quite understand the necessity of using NO to generate PDE, and it would be great if authors could elaborate on this.
>
> - Thank you for your insightful question. **Neural operators is not a prerequisite for the application of our SKR algorithm**. Indeed, our initial use of neural operators revealed the time-consuming issue of dataset generation. After an extensive study of Krylov methods, we developed the SKR algorithm, which has proven to be highly successful. Neural operators are one of the most renowned and efficient data-driven methods for solving PDE problems, and the time-intensive nature of dataset generation has been a significant hurdle in the field, which is why our paper is set in this context. However, **the SKR algorithm's utility is not limited to neural operator problems and can be applied to other areas**, such as:
>   - **Data-Driven PDE Algorithms**: There are various data-driven PDE solving or parameter optimization algorithms in both traditional heuristics and AI for PDE domains. These algorithms often require the frequent generation of large volumes of PDE data for datasets, a process that our SKR algorithm can accelerate. For instance, [1] [2].
>   - **Specific Physical Problem Solving**: In certain scenarios, modeling the physical problems at hand translates into solving multiple related linear systems. Our SKR algorithm is adept at efficiently handling such issues. For instance, this is evident in the modeling of fatigue and fracture through finite element analysis [3], which employs dynamic loading across numerous steps, resulting in a substantial number of interrelated linear systems. Similarly, in the resolution process of lattice quantum field theory [4], the end goal involves solving a large collection of correlated linear systems.
>   - Reference
>     - [1] Daniel Greenfeld, Meirav Galun, Ron Kimmel, Irad Yavneh, Ronen Basri. "Learning to Optimize Multigrid PDE Solvers."  International Conference on Machine Learning (ICML). 2019.
>     - [2] Jun-Ting Hsieh, Shengjia Zhao, Stephan Eismann, Lucia Mirabella, Stefano Ermon. "Learning Neural PDE Solvers with Convergence Guarantees." International Conference on Learning Representation (ICLR). 2019.
>     - [3] A. Gullerud, R. H. Dodds. "MPI-based implementation of a PCG solver using an EBE architecture and preconditioner for implicit, 3-D finite element analyses." Comput. & Structures, 79 (2001), pp. 553–575.
>     - [4] Shin Muroya, Atsushi Nakamura, Chiho Nonaka, Tetsuya Takaishi. "Lattice QCD at Finite Density: An Introductory Review." Progress of Theoretical Physics, Volume 110, Issue 4, October 2003, Pages 615–668.

---

> ### Author Response · Authors · 2023-11-19
> **Response to Reviewer BjLB (4/4)**
>
> ## **Questions 2**
>
> > For many problems such as Poisson equation the resulting matrix, A, is symmetric and using iterative solvers for symmetric case (such as CG solver) could significantly help speed up the process. With that being said, could authors modify the algorithm by using symmetry property to make it perform better for symmetric matrices?
>
> - Thank you for your suggestion. In our paper, the SKR algorithm is designed to address the most general and common case, which is that of non-symmetric matrices, hence our baseline comparison with GMRES. As you mentioned, specific optimizations are indeed required for matrices with particular properties. In this regard, **we have already designed some specific versions of the SKR algorithm for particular matrix types.**
>
>   - **Block parallel version**: The block concept is an approach to parallelize large matrices and reduce memory overhead. Inspired by the block Krylov idea, we have redesigned a block version of our SKR algorithm and conducted preliminary experiments:
>
>     1. An experimental simulation was conducted to generate a dataset for the Helmholtz equation. Each corresponding matrix has a dimension of 10000, with Successive Over-Relaxation (SOR) used uniformly for matrix preconditioning, running on 72 parallel MPI threads.
>
>     2. The upper half of the table presents the average computation time taken by both algorithms to solve each linear system, measured in seconds. The lower half details the average number of iterations required to solve each linear system. The first row of the table specifies the precision requirements for solving linear systems.
>
>        - |         |                 | 1E-03 | 1E-05 | 1E-07 |
>          | ------- | --------------- | ----- | ----- | ----- |
>          | time(s) | GMRES           | 3.22  | 4.16  | 4.66  |
>          | time(s) | Block SKR(ours) | 0.015 | 0.025 | 0.030 |
>          | iter    | GMRES           | 5503  | 7074  | 7976  |
>          | iter    | Block SKR(ours) | 224   | 269   | 305   |
>
>     3. **The experimental results reveal that our SKR algorithm achieves a 150-200 fold acceleration in computation time and a 24-26 fold reduction in the number of iterations compared to the baseline. The significant improvement in computation time is mainly attributed to the effects of** **MPI** **parallelization.**
>
>   - **For symmetric matrices**: For matrices that are symmetric, the conventional algorithm is the Minimal Residual Method (MINRES). The primary difference between MINRES and GMRES lies in their Krylov subspace iteration process: GMRES is derived from the non-symmetric Arnoldi iteration, which produces an upper Hessenberg matrix, while MINRES is based on the symmetric Lanczos iteration, resulting in a tridiagonal matrix. Our SKR algorithm can be similarly modified to create a version tailored for symmetric matrices.
>
>   - **For symmetric positive definite matrices**: When the matrix is symmetric and positive definite, the traditional algorithm is the Conjugate Gradient (CG) method. Due to the symmetric positive definite property, the CG method essentially solves the quadratic form $\frac{1}{2}x^TAx - b^Tx$. The CG algorithm constructs a set of conjugate directions, and each iteration is akin to performing Krylov subspace iteration in two directions simultaneously. With similar modifications, we can adapt our SKR algorithm for symmetric positive definite matrices.
>
>   - **For any specific algorithm**: In fact, with certain modifications, our SKR algorithm can be configured as a special matrix preconditioner. As a preconditioner, it can be easily embedded into any algorithm designed for different types of matrices.
>
> - We have successfully implemented the block parallel version of our SKR algorithm. The designs for the symmetric and symmetric positive definite matrix versions have been completed. However, due to the limited time available for this rebuttal, we have not yet realized the specific code implementation. **In the future, we plan to actualize these specialized versions and integrate them into the FEniCSx library.**

---

### Author Response · Authors · 2023-11-21
**We are looking forward to your further comments and/or questions.**

Dear Reviewers,

Thanks again for your positive and constructive comments, which are of great help to improve the quality of our work. As the rebuttal phase is approaching (due on November 22), we are eagerly looking forward to your further comments and/or questions.

We sincerely hope that our rebuttal has properly addressed your questions. If possible, we would deeply appreciate it if you could further raise your scores. If not, please let us know your further comments, and we will continue actively responding to your comments and improving our submission.

Best,
Authors

---

### Meta-Review · Area_Chair_DWqr · 2023-12-10

**Metareview:**

The paper utilizes sorted Krylov subspace recycling technique to arrange the linear systems for the dataset creation in 'similarity order', and then recycle the subspaces -- classical idea of linear algebra. Results in 10x speedup.

Strengths:
1) Simple idea
2) Based on classical work in linear algebra
3) Well-written paper
4) Significant practical speedup

Weaknesses:
1) Needs linear systems

**Justification For Why Not Higher Score:**

The method is still quite specific.

**Justification For Why Not Lower Score:**

The reviews are positive, and the paper is pleasant to read, with nice practical results.

---

### Decision · Program_Chairs · 2024-01-16

Accept (spotlight)